# Evaluation of root-zone soil moisture products over the Huai river basin

En Liu[1,2], Yonghua Zhu[1], Jean-Christophe Calvet[2], Haishen Lü[1], Bertrand Bonan[2], Jingyao Zheng[1], Qiqi Gou[1], Xiaoyi Wang[1], Zhenzhou Ding[1], Haiting Xu[1], Ying Pan[1], Tingxing Chen[1]

[1]State Key Laboratory of Hydrology-Water Resources and Hydraulic Engineering, 14 College of Hydrology and Water Resources, Hohai University, Nanjing 210098, China
[2]CNRM, Université de Toulouse, Météo-France, CNRS, 31057, Toulouse, France

*Correspondence to*: Yonghua Zhu (zhuyonghua@hhu.edu.cn),

Jean-Christophe Calvet (jean-christophe.calvet@meteo.fr)

**Abstract:** Root zone soil moisture (RZSM) is critical for water resource management, drought monitoring and sub-seasonal flood climate prediction. While RZSM is not directly observable from space, several RZSM products are available and widely used at global and continental scales. This study conducts a comprehensive and quantitative evaluation of eight RZSM products using observations from 58 *in situ* soil moisture stations over the Huai River Basin (HRB) in China. Attention is drawn to the potential factors that contribute to the uncertainties of model-based RZSM, including the errors in atmospheric forcing, vegetation parameterizations, soil properties, and spatial scale mismatch. The results show that the Global Land Data Assimilation System Catchment Land Surface Model (GLDAS_CLSM) outperforms the other RZSM products with the highest correlation coefficient ($R = 0.69$) and the lowest unbiased root mean square error (ubRMSE = 0.018 $m^3\,m^{-3}$). While SMOS Level 4 (L4) RZSM shows the worst performance among eight RZSM products. The RZSM products based on land surface models generally perform better in the wet season than in the dry season due to the enhanced ability to capture of the temporal dynamics of *in situ* observations in the wet season and the inertia of remaining high soil moisture values even in the dry season. While SMOS L4 RZSM product, derived from SMOS L3 surface moisture (SSM) combined with exponential filter method, performs better in the dry season due to the attenuated ground microwave radiation signal caused by the increased water vapor absorption and scattering in the wet season. The underestimated SMOS L3 SSM triggers the underestimation of RZSM in SMOS L4. The overestimated RZSM products based on land surface models could be associated with the overestimated precipitation amounts and frequency, the underestimated air temperature and ratio of transpiration to the total terrestrial evapotranspiration. In addition, the biased soil properties and flawed vegetation parameterizations affect the hydrothermal transport processes represented in different LSMs and lead to inaccurate soil moisture simulation. The scale mismatch between point and footprint also introduces representative errors. The comparison of frequency of normalized soil moisture between RZSM products and *in situ* observations indicates that the LSMs should focus on reducing the frequency of wet soil moisture, increasing the frequency of dry soil moisture and the ability to capture the frequency peak of soil moisture. The study provides some insights into how to improve the ability of land surface models to simulate the land surface states and fluxes by taking into account the issues mentioned above. Finally, these results can be extrapolated to other regions located in the similar climate zone, as they share the similar precipitation patterns that dominate the terrestrial water cycle.

## 1 Introduction

Soil moisture plays a key role in the hydrological cycle and land-atmosphere interactions. It controls water and
energy balances (Calvet, 2000; Brocca et al., 2010; Xing et al., 2021) and has been recognised by the World
Meteorological Organization (WMO) as one of the 50 essential climate variables (Cho et al., 2015). In particular,
root zone soil moisture (RZSM) has important applications in agricultural drought monitoring, water resource
management, flood forecasting and seasonal climate prediction (Reichle et al., 2017a; Zhou et al., 2020; Beck et
al., 2021; Xing et al., 2021; Xu et al., 2021; Fan et al., 2022). RZSM is the amount of water held in the top 1 m of
the soil column that is available for plant transpiration and biomass production, which is crucial for agricultural
drought monitoring. Different ecosystems in different climate and topography conditions have different rooting
depth, and root zone water storage capacity (Gao et al., 2014; Kleidon, 2014; Fan et al., 2017; Gao et al., 2019a).
The depth of root tissue can vary from a few centimeters to about two meters. However, in large-scale modelling
studies, the term "root zone" commonly refers to the 0-100 cm soil layer. This assumption is based on the fact that
the vegetation root tissue is mostly densely distributed in this area (Baldwin et al., 2017). In the context of climate
change, extreme events such as floods, droughts and heat waves are becoming more frequent around the world,
with significant impacts on RZSM (Lorenz et al., 2010; Hauser et al., 2016; Al Bitar et al., 2021). For example,
flash droughts are severely affecting RZSM and agricultural production in the Huaibei Plain, China (Gou et al.,
2022).

Recently, microwave-based satellite missions provide global soil moisture retrievals with approximately 3-
day temporal resolution, but are limited to the top few centimetres (0-5 cm for L-band) due to the limitations of
microwave penetration depth (Kerr et al., 2001; Reichle et al., 2017b). Therefore, various approaches have been
developed to estimate the RZSM and are roughly divided into three categories (Liu et al., 2023), including (1)
statistics-based methods, such as linear regression (Zhang et al., 2017) and cumulative distribution function (Gao
et al., 2019b), (2) data-driven machine learning methods, such as random forest (Carranza et al., 2021) and artificial
neural network (Kornelsen and Coulibaly, 2014), (3) physically based methods, such as data assimilation of
satellite-derived observations into LSMs (Albergel et al., 2017; Bonan et al., 2020). Among them, the assimilation
of satellite-derived observations into LSMs is considered as the most accurate method to estimate RZM due to the
explicit physical mechanism, while requiring large amounts of input data (precipitation, air temperature, radiation,
etc.). To date, several RZSM products have been developed for broader global-scale applications, such as the
Global Land Data Assimilation System (GLDAS_NOAH and GLDAS_CLSM) (Rodell et al., 2004), the China
Land Data Assimilation System (CLDAS) (Shi et al., 2014) and the Soil Moisture Active Passive (SMAP) Level
4 (L4) (Reichle et al., 2012; Reichle et al., 2017a), the European Centre for Medium-Range Weather Forecasts
(ECMWF) fifth generation reanalysis (ERA5) (Hersbach et al., 2020), the Modern-Era Retrospective Analysis for
Research and Applications version 2 (MERRA-2) (Gelaro et al., 2017), and the National Centers for
Environmental Prediction Climate Forecast System version 2 (NCEP CFSv2) (Saha et al., 2014). These RZSM
products are generated by combining LSMs driven by meteorological forcing fields from atmospheric general
circulation model (AGCM) and satellite-derived data using different data assimilation techniques (Calvet and
Noilhan, 2000; Rodell et al., 2004). In addition, the Soil Moisture and Ocean Salinity (SMOS) Centre Aval de
Traitement des Données (CATDS) provides SMOS L4 RZSM products, which are derived from SMOS Level 3
(L3) 3-day SSM retrievals using a statistical exponential filter model (Albergel et al., 2008; Al Bitar and Mahmoodi,
2020).

The accuracy of RZSM products is strongly influenced by the quality of meteorological forcing data, especially precipitation and air temperature (Zeng et al., 2021). Numerous studies have shown large uncertainties in global climate atmospheric forcing data, particularly for precipitation frequency, intensity and heavy precipitation events (Sun et al., 2005; Piani et al., 2010; Velasquez et al., 2020; Jiao et al., 2021). Accurate representation of soil properties is also critical. Many global LSMs rely on the FAO/UNESCO (Food and Agriculture Organization, United Nations Educational, Scientific and Cultural Organization) World Soil Map (Reynolds et al., 2000), including GLDAS products (Bi et al., 2016; Yang et al., 2020), NCEP CFSv2 (Yang et al., 2020), ERA5 (Qin et al., 2017; Yang et al., 2020), SMOS L4 (Al Bitar et al., 2021), MERRA-2 (McCarty et al., 2016; Gelaro et al., 2017). However, this soil map contains limited soil information in many regions, including China (Shangguan et al., 2013), leading to increased uncertainty in soil moisture simulations. In addition, the lack of representation of soil stratification can significantly affect the simulation of RZSM by LSMs. In the Huaibei Plain, the stratification of plough, black soil and lime concretion layers can hinder the vertical movement of water from the surface layer to the root zone layer (Li et al., 2011; Zha et al., 2015; Gu et al., 2021). Finally, the accuracy of soil moisture simulations is also affected by inadequate model structures and imperfect parameterization schemes, especially for representation of vegetation in LSMs, such as the land cover and vegetation canopy and root tissue parameterizations (Nogueira et al., 2020; Stevens et al., 2020; van Oorschot et al., 2021), soil evaporation and transpiration model representation (Lian et al., 2018; Dong et al., 2022; Feng et al., 2023). Vegetation is usually represented by land cover maps (that are usually prescribed similar to soil maps), which can be very different for the different models and exhibits large uncertainties in simulating the water and energy exchange between land surface and atmosphere. For example, Nogueira et al. (2020) found that the misrepresentation of the vegetation coverage results in a cold bias in land surface temperature during summer, they proposed an improved representation of vegetation with an update of the LAI and high- and low- vegetation fractions, types and density, which effectively reduces the cold bias. van Oorschot et al. (2021) proposed a climate-controlled root zone storage capacity by calculating a time-varying total soil depth based on a moisture depth model instead of using a constant of 2.89 m in the original HTESSEL LSM, which improved the water flux simulations. Dong et al. (2022) demonstrated that the inaccurate partitioning of evapotranspiration into soil evaporation and vegetation canopy transpiration results in warm bias in air temperature due to the inadequate utilization of RZSM for transpiration, which results in the underestimated ration of transpiration to evapotranspiration. Different LSMs are used in LDAS or reanalysis products, such as the Noah LSM in GLDAS_NOAH and NCEP CFSv2 (Rodell et al., 2004; Saha et al., 2014), HTESSEL in ERA5 (Hersbach et al., 2020), CLSM in GLDAS_CLSM, MERRA-2 and SMAP L4 (Koster et al., 2000; Reichle et al., 2017d; Reichle et al., 2021), the Common Land Model (CoLM) and the Community Noah LSM with multi-parameterisation options (Noah-MP) in CLDAS products (Wang et al., 2021a). The exponential filter technique is used in SMOS L4 (Al Bitar et al., 2021).

Numerous studies have been conducted to validate and assess the utility of SSM using *in situ* observations in the topsoil layer (Collow et al., 2012; Cui et al., 2017; Beck et al., 2021; Zheng et al., 2022). On the other hand, validation studies for RZSM are relatively rare, especially over China (Xing et al., 2021; Xu et al., 2021; Fan et al., 2022). Given the importance of the Huai River Basin (HRB) as an agricultural grain production area in China, it is crucial to evaluate the performance of different RZSM products in this region. RZSM products can be validated against *in situ* observations, which serve as a reference dataset. Differences between *in situ* RZSM observations

and RZSM products can be attributed to errors in meteorological forcing data, soil properties, parameterisation and scale mismatch.

The objectives of this study are to (1) compare eight global RZSM products (ERA5, MERRA-2, NCEP CFSv2, GLDAS_CLSM v2.2, GLDAS_NOAH v2.1, CLDAS v2. 0, SMAP L4 and SMOS L4) with *in situ* soil moisture observations over the HRB from 1 April 2015 to 31 March 2020, (2) compare the RZSM products with each other over the HRB, (3) investigate the potential sources of errors on the performance of the RZSM products, including meteorological forcing data, soil properties, soil stratification, vegetation parameterization and scale

mismatch. The paper is organized as follows. The gridded RZSM products and *in situ* validation datasets (precipitation, air temperature, soil texture) are presented in Sect. 2. Section 3 describes the RZSM pre-processing methods and the statistical metrics used to evaluate the different datasets. The validation and the intercomparison of the RZSM products are presented in Sect. 4. Section 5 discusses the potential sources of error in various RZSM products. Section 6 provides the main conclusions.


**2 Datasets**

**2.1 The Huai River Basin study area**

The HRB is the transitional zone between the northern subtropical and warm temperate climates, and it is one of the most important commodity grain production areas in China. It is located in eastern China, 111°55′-121°25′
E, 30°55′-36°36′N, and covers an area of 270000 km$^2$ (Fig. 1). The HRB has a typical humid and sub-humid monsoon climate. The average annual precipitation is 888 mm and increases from north to south. More than 60 % of the annual precipitation falls between June and September (Zhang et al., 2009). The HRB suffers from frequent floods and droughts due to the spatial and temporal variability of precipitation and evaporation. The main land cover types in the HRB are rainfed croplands, followed by irrigated croplands, forests and grasslands. Overall, the
terrain of the HRB is relatively flat, with a large plain covering 90% of the area. The cultivated area in the HRB is approximately 127200 km$^2$, of which 76 % is irrigated according to the Manual of the Huai River Basin Irrigation Area (Chapter 2.1) and Summary of Flood Control Planning for the Huai River Basin (http://www.hrc.gov.cn). The water resource infrastructures include reservoirs, electromechanical wells, diversion locks and pumping stations built along lakes and rivers. Most cropland fields are irrigated by irrigation canals or a combination of
wells and canals (Wang et al., 2021a). Annual evaporation can exceed precipitation. It ranges from 900 to 1500 mm and decreases from north to south (Wang et al., 2021a). Heavy irrigation in the HRB can explain the extra water available for evaporation.

**2.2 HRB *in situ* measurements**

The HRB soil moisture network was established by the Ministry of Water Resources of the People's Republic
of China. It consists of 58 *in situ* stations (see Fig .1) and provides soil moisture measurements at four depths of 10, 20, 40 and 100 cm (Liu et al., 2023). At each station, volumetric soil moisture measurements in units of m$^3$ m$^{-3}$ are collected at 08:00 AM local solar. These probes are calibrated using gravimetric measurements taken at each soil depth. The deployment of the soil moisture stations and the collection of soil moisture measurements follows the specifications for soil moisture monitoring (MWR, 2015). Since the study aims to evaluate the accuracy of
eight RZSM products (0-100 cm) which are summarized in Table 1, the *in situ* soil moisture measurements at the four depths are depth-weighted averaged to obtain the 0-100 cm soil moisture data.

The China Daily Gridded Ground Precipitation and Air Temperature dataset V2.0, provided by the China Meteorological Administration (CMA) (http://data.cma.cn) with a spatial resolution of 0.5°×0.5° (approximately 55.6 km), serves as a reference dataset for validating the meteorological forcing fields used in reanalyses and
LDAS. The CMA dataset is derived by spatial interpolation using the partial thin-plate smoothing spline method from 2474 ground-based meteorological station observations across China, following stringent quality controls and necessary corrections. At the national level, the average coverage of gauging stations in a grid cell is 38 %. However, in the eastern part of China, where the HRB is located, the coverage reaches up to 77 %. The dataset has been extensively validated against ground observations and is of high quality. For example, the precipitation
data has a root mean square error (RMSE) of 0.49 mm/month and a correlation coefficient (*R*) of 0.93 with a significance level p smaller than 0.01 (CMA, 2012b). The annual air temperature data have a mean bias and RMSE ranging from -0.2 to 0.2°C and from 0.2 to 0.3°C, respectively (CMA, 2012a).

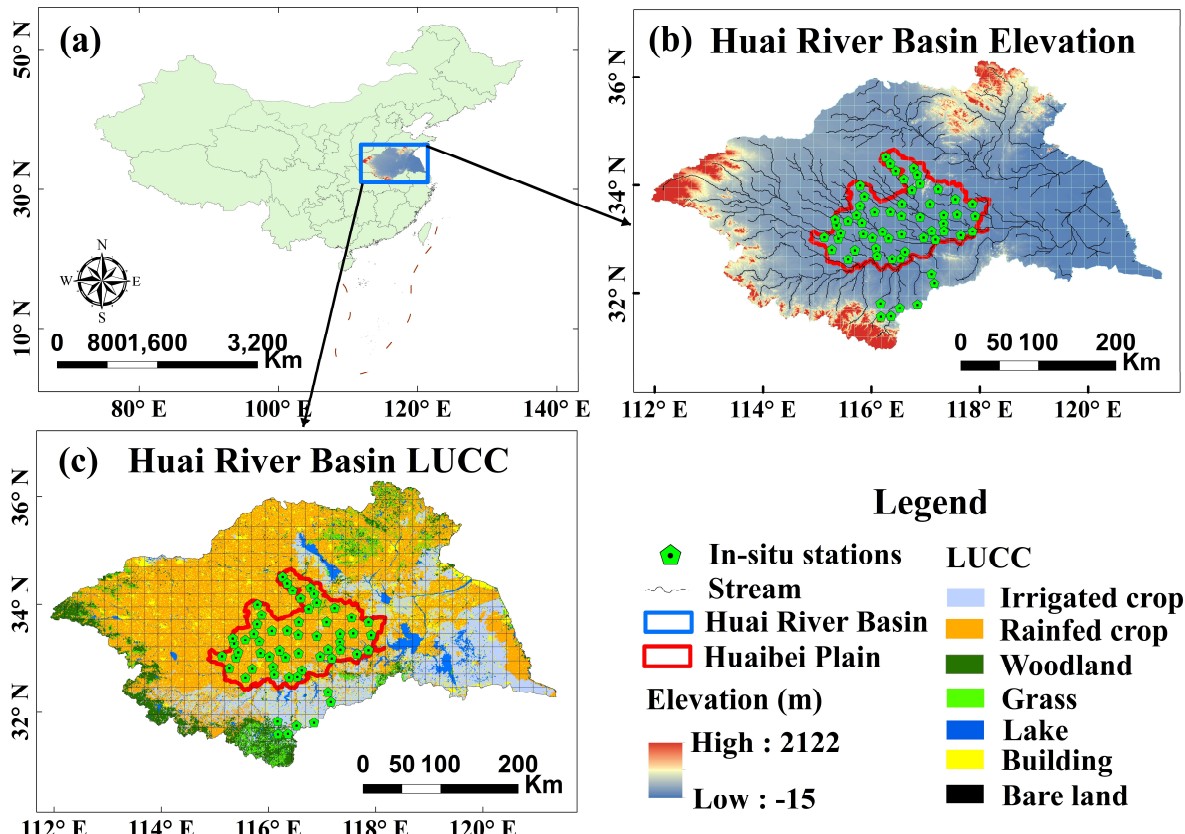

**Fig. 1 Overview of the study area (including elevation, stream and land cover) and distribution of *in situ* soil moisture**
**stations (green pentagon). The squares in Fig.1b and c represent 0.25° grid.**

### 2.3 Soil map

Soil databases used in many global LSMs have traditionally relied on the FAO/UNESCO 1:5 million scale World
Soil Map with a spatial resolution of 5 arc minutes (approximately 10 km). However, this FAO/UNESCO soil map
contained limited soil information in different regions, including China. Consequently, the uncertainties in soil
properties contributed to larger errors in the land surface variables simulated by the LSMs (e.g. RZSM), especially
over China (Nachtergaele et al., 2009; Shangguan et al., 2013). To address these uncertainties, the Harmonised
World Soil Database (HWSD) was developed by FAO and the International Institute for Applied Systems Analysis
(IIASA) with a resolution of 30 arcseconds (approximately 1 km). The HWSD combines recently collected
regional and national updates of soil information with the FAO/UNESCO 1:5 million scale World Soil Map (FAO
et al., 2012). HWSD also incorporates the 1:1 million scale soil map of China provided by the Institute of Soil
Science, Chinese Academy of Sciences (ISSCAS).

A China dataset of soil properties was developed by Shangguan et al. (2013), which integrates the physical
and chemical properties of 8979 soil profiles along with the soil map of China and is employed in the CLDAS
product (Qin et al., 2017). The dataset provides information on soil properties for eight layers (0-2.3 m) at a spatial
resolution of 30×30 arcseconds (approximately 1 km). The FAO/UNESCO and HWSD V1.2 soil datasets are
employed in different LSMs, respectively. The China dataset of soil properties developed by (Shangguan et al.,

2013) is used as a reference to evaluate the soil properties (i.e. sand and clay content, bulk density and soil organic matter) of FAO/UNESCO and HWSD V1.2 datasets in section 5.2.

### 2.4. Gridded RZSM products

The eight products considered in this study (Table 1) are presented below.

### 2.4.1 ERA5

ERA5 is the fifth generation global atmospheric reanalysis produced by ECMWF (Hersbach et al., 2023). ERA5 is developed using the 4-Dimensional Variational (4D-Var) data assimilation method, which incorporates a 10-member ensemble and model forecasts from the ECMWF Integrated Forecast System (IFS) in CY41R2 with 137

hybrid sigma/pressure model levels in the vertical and the top level at 0.01 hPa (Hersbach et al., 2020). ECMWF IFS mainly assimilates satellite-derived precipitation data, such as the Advanced Microwave Scanning Radiometer 2 (AMSR-2), the Global Precipitation Measurement (GPM), FengYun-3-C (FY-3-C), the Tropical Rainfall Measuring Mission (TRMM), and ground-based radar precipitation composites provided by the World Meteorological Organization Information System, to obtain the best precipitation estimates. The near-surface

atmospheric forcing field at the lowest level of the atmospheric model (about 10 m a.g.l.) is used to force the HTESSEL LSM, which serves as the land surface component of the ECMWF IFS to model the land surface variables. HTESSEL uses the FAO/UNESCO World Soil Map and the Global Land Cover Characteristics (GLCC) database (Nogueira et al., 2020). The diffusivity form of the Richard's equation is used to describe the vertical water flow within the soil column that is discretized into four layers in the HTESSEL. Besides, HTESSEL ignores

the exchange of lateral water fluxes between adjacent grid cells. The screen-level parameters analysis (2 m temperature and relative humidity) is carried out first, then its increments are incorporated into the soil moisture analysis.

### 2.4.2 MERRA-2

MERRA-2 is the latest version of a global atmospheric reanalysis product produced by NASA Global Modelling

and Assimilation Office (GMAO, 2015). It uses the Goddard Earth Observing System Model (GEOS-5.12.4) atmospheric data assimilation system, which consists of (1) the GEOS atmospheric model and (2) the Gridpoint Statistical Interpolation assimilation system. The precipitation forcing is the weighted average of model background precipitation generated by GEOS-5 FP-IT (Forward Processing system for Instrument Teams) after 31 December 2014 and precipitation generated by AGCM, with weights dependent on latitude. The National

Oceanic and Atmospheric Administration (NOAA) Climate Prediction Center (CPC) Unified Gauge-Based Analysis of Global Daily Precipitation (CPCU) product is used to correct the model background precipitation. The CPC Merged Analysis of Precipitation (CMAP) product is rescaled to match the climatology of the Global Precipitation Climatology Project product, version 2.1 (GPCPv2.1) and is fully used in Africa, which allows the observed precipitation to impact, via evapotranspiration, the near-surface air temperature and humidity, thereby

yielding a more self-consistent near-surface meteorological dataset (Reichle et al., 2017d). CLSM uses the FAO/UNESCO World Soil Map and the Global Land Cover Characteristics (GLCC), version 2.0 (Reichle et al., 2017c), and is used as the land surface component of MERRA-2 to perform the land surface analysis. The CLSM used in MERRA-2 simulates the average soil moisture in the surface layer (0-5 cm), the root zone (0-100 cm) and

the varying profile (from the land surface to the bedrock), and does not take into account lateral water fluxes
(groundwater or river flow) between catchments which is used as the basic computational unit (Reichle and Koster, 2003).

### 2.4.3 NCEP CFSv2

NCEP CFSv2 is the third generation global atmospheric reanalysis product developed by the Environmental Modelling Center at NCEP. It is a global, high-resolution, coupled atmosphere-ocean-land surface-sea ice system designed to provide the best estimate of the state of these coupled domains (Saha et al., 2011). The global atmospheric data assimilation system (GDAS) employed in the climate forecast system simulates 64 sigma-pressure hybrid layers vertically. The Noah LSM is forced by the atmospheric forcing variables at the lowest level from the Climate Forecast System Reanalysis (CFSR) GDAS and the blended precipitation forcing. The global precipitation analysis from CMAP and CPCU and the model background precipitation from GDAS are integrated based on a latitude-dependent weighting method to provide the optimal global precipitation forcing for a reliable land surface simulation (Meng et al., 2012). The Noah LSM is first used in the coupled land-atmosphere-ocean model to provide the initial conditions of the land surface states and fluxes, and then in the semi-coupled CFSR Global Land Data Assimilation System (GLDAS) to perform the land surface analysis and provide the evolving land surface states and fluxes (Saha et al., 2010; Saha et al., 2014). The Noah LSM employed in NCEP CFSv2 uses the FAO/UNESCO World Soil Map and the land cover classification based on Advanced Very High Resolution Radiometer (AVHRR) 1 km data set.

### 2.4.4 GLDAS_NOAH

GLDAS_NOAH version 2.1 is the mainstream land surface analysis product developed by NASA Goddard Earth Sciences Data and Information Services Center (GES DISC) and aims to provide the optimal fields of land surface states and fluxes by incorporating large amounts of satellite- and ground-based observations (Rodell et al., 2004; Beaudoing et al., 2020). No data assimilation procedure was implemented in the GLDAS_NOAH version 2.1 product. The offline (not coupled to the atmosphere) Noah LSM is forced with combination of model- (NOAA/Global Data Assimilation System (GDAS) atmospheric analysis fields) and observation-based precipitation (the disaggregated Global Precipitation Climatology Project (GPCP) V1.3 Daily Analysis precipitation fields) and radiation data (the Air Force Weather Agency's AGRicultural METeorological modeling system (AGRMET) radiation fields) to provide optimal fields of land surface analysis. The soil column is discretized into four layers for describing the movement of soil moisture based on the diffusive form of Richard's equation in NOAH LSM, which is same with NCEP CFSv2. GLDAS_NOAH uses the hybrid STATSGO/FAO World Soil Map and the modified IGBP MODIS (Moderate Resolution Imaging Spectroradiometer) 20-category vegetation classification (Rui et al., 2021).

### 2.4.5 GLDAS_CLSM

GLDAS_CLSM version 2.2 is one of the most popular analysis dataset of land surface states and fluxes developed by NASA Goddard Earth Sciences Data and Information Services Center (GES DISC). The CLSM embedded in the GLDAS is forced by the meteorological analysis fields from the operational ECMWF IFS. These meteorological forcing fields are obtained by assimilating large amounts of atmospheric observations to update

the model background predictions (e.g. precipitation) derived in the forecast step and are available at 0.25-degree, 3-hourly interval (Li et al., 2019; Li et al., 2020). The CLSM used in GLDAS does not have explicit vertical levels for soil moisture and only simulate the soil moisture represented by the surface layer (0-2 cm), root zone (0-100 cm) and varying profile. The lateral water fluxes between catchments are also not taken into account in the current CLSM (Reichle and Koster, 2003). The FAO/UNESCO World Soil Map and the University of Maryland (UMD) land cover classification based on AVHRR land cover map are used in the GLDAS_CLSM (Rui et al., 2021). Unlike the open-loop GLDAS version 2.1 product, GLDAS version 2.2 product assimilates observations of the total terrestrial water (TWS) anomaly from Gravity Recovery and Climate Experiment (GRACE). Temporal changes of TWS are influenced by changes in soil moisture, snow and ice, surface water and biomass, and groundwater storage.

### 2.4.6 CLDAS

The CLDAS-2.0 is the Asian atmospheric and land surface analysis product with high temporal and spatial resolution developed and released by CMA. It is produced based on a multi-LSMs operational system consisting of CLM, CoLM, and Noah-MP, with a spatial coverage of 0-60° N and 70-150° E and temporal coverage from January 2008 to present (CMA, 2015). The production of CLDAS-V2.0 involves the following three processes. Firstly, nearly 40000 automated meteorological station measurements, ECMWF and NCEP numerical analysis/forecast products, satellite-derived precipitation (FY2) and digital elevation model (DEM) are used to produce 0.0625°, hourly estimates of meteorological forcing data by operating the Space-Time Multi-Scale Analysis System (STMAS) (Shi et al., 2014; Wang et al., 2021a). Meanwhile, the meteorological forcing is validated using national automatic station observations (more than 2400 stations). Second, the meteorological forcing is used to drive the multi-LSMs system to obtain a multi-layer ensemble of soil moisture estimates. Finally, the ensemble mean is applied to each soil layer to produce a soil moisture ensemble analysis product. CLDAS utilizes the soil property dataset developed by Shangguan et al. (2013) and simulates five soil layers for the diffusion for water flux and the transmission for heat flux vertically.

### 2.4.7 SMAP L4

The SMAP Level-4 soil moisture (L4-SM) is produced by assimilating SMAP radiometer Level 1C brightness temperature observations into CLSM and provides global, 3-hourly, 9-km resolution estimates of SSM (0-5 cm) and RZSM (0-100 cm) from March 2015 to present (Reichle et al., 2020; Reichle et al., 2021). The Goddard Earth Observation System, version 5, LDAS (GEOS-5 LDAS) uses a spatially distributed ensemble Kalman filter (EnKF) to assimilate the observations into CLSM (Rienecker et al., 2008). The EnKF has a 3-hourly update time step and is used to interpolate and extrapolate the brightness temperature and model estimates in time and space (Reichle et al., 2017a). The GEOS-5 CLSM is driven by surface meteorological data (precipitation, radiation, etc.) from the GEOS-5 Forward Processing (FP) system where large amounts of observations are assimilated into a global atmospheric model. The CPCU, 0.5-degree, daily precipitation observations are used to correct the GEOS-5 FP model background precipitation. Prior to the GEOS-5 FP precipitation correction, both the CPCU precipitation data and the hourly background precipitation are scaled to the climatology of the GPCPv2.2 pentad precipitation product. SMAP L4 product uses the updated HWSD V1.2 soil property dataset and the MODIS land cover product based on the UMD classification (Reichle et al., 2012).

### 2.4.8 SMOS L4

The SMOS L4 soil moisture product is disseminated by SMOS CATDS and provides global, daily estimates of RZSM (0–100 cm) over a 25-km EASE-2 grid from January 2010 to present (Al Bitar and Mahmoodi, 2020; CATDS, 2021). The SMOS L4 RZSM is derived from the SMOS L3 3-day SSM product using a modified exponential filter linking the characteristic time length T (the transfer time of water from the surface layer to the root zone layer) to the soil properties (Pablos et al., 2018). The soil parameters (i.e. saturated water content, the

soil moisture at wilting point and the soil moisture at field capacity) are calculated based on the soil texture from FAO soil texture map (Al Bitar et al., 2021). The product is based on SMOS descending orbit (18:00) observations and other ancillary datasets such as MODIS observations, NCEP climate data and an updated FAO/UNESCO soil properties map. The soil column is divided into three layers (layer 1: 0-5 cm, layer 2: 5-40 cm, layer 3: 40-100 cm) in a water bucket model. The scaled 0-5 cm soil moisture is modified using a

logarithmic function and filtered to obtain the layer 2 soil moisture. The scaled layer 2 soil moisture is then filtered using a different value of T to give the layer 3 soil moisture. Finally, the RZSM (0-100 cm) is calculated as a depth-weighted average of the soil moisture of the three layers (Al Bitar et al., 2021).

**Table 1.** Description of global and regional RZSM gridded products used in this study.

| Dataset | Land surface model | Time period | resolution | Soil map | Soil layers | References |
|---|---|---|---|---|---|---|
| ERA5 (Global) | HTESSEL | January 1979-present | Hourly /0.25° | FAO | 0-7 cm, 7-28 cm, 28-100 cm, 100-289 cm | Hersbach et al. (2020); Xu et al. (2021) |
| MERRA-2 V2.0 (Global) | CLSM | January 1980-present | Hourly /0.25° | FAO | 0-5 cm, 0-100 cm | Gelaro et al. (2017); Reichle et al. (2017d) |
| NCEP CFSv2 V2.0 (Global) | Noah | January 2011-present | 6-Hourly /0.20° | FAO | 0-10 cm, 10-40 cm, 40-100 cm, 100-200 cm | Qin et al. (2017) |
| GLDAS_NOAH V2.1 (Global) | Noah | January 2000-present | 3-Hourly /0.25° | FAO | 0-10 cm, 10-40 cm, 40-100 cm, 100-200 cm | Bi et al. (2016); Xing et al. (2021) |
| GLDAS_CLSM V2.2 (Global) | CLSM | February 2003-present | Daily /0.25° | FAO | 0-2 cm, 0-100 cm | Li et al. (2019) |
| CLDAS V2.0 (Asia) | CLM CoLM Noah-MP | January 2008-present | Hourly /0.0625° | Shuang guan et al. (2013) | 0-5 cm, 0-10 cm, 10-40 cm, 40-100 cm, 100-200 cm | Chen andYuan (2020); Wang et al. (2021a) |
| SMAP Level 4 V5 (Global) | CLSM | March 2015-present | 3-Hourly /9 km | HWSD | 0-5 cm, 0-100 cm | Reichle et al. (2017a); Ma et al. (2019) |
| SMOS Level 4 V301 (Global) | Exponential filter (no LSM) | January 2010-present | Daily /0.25° | FAO | 0-100 cm | Tangdamrongsub et al. (2020); Al Bitar et al. (2021) |

Note that precipitation, air temperature and soil texture have the same resolution as soil moisture.

## 3 Methods

### 3.1 Statistical metrics

Four widely used statistical metrics were used to quantitatively assess the performance of RZSM products against *in situ* measurements. The Pearson correlation coefficient (*R*) measures the linear correlation between the *in situ* measurements and the RZSM products. Mean Bias Error (MBE) reflects the mean systematic deviation of the model simulations relative to the measurements. Accuracy is assessed using the Root Mean Square Error (RMSE). The unbiased RMSE (ubRMSE) measures the standard deviation of the differences. In addition, Probability of Detection (POD), False Alarm Ratio (FAR) and Critical Success Index (CSI) are used to assess the ability of the global gridded rainfall to reproduce the measured rainfall (Su et al., 2019). POD is the proportion of real precipitation events simulated by AGCM relative to the actual precipitation events, reflecting the ability of AGCM to detect precipitation. FAR is the fraction of unreal precipitation events out of the total precipitation events simulated by AGCM. CSI is a more balanced score that combines the characteristics of false alarms and missed events, representing the probability of successful simulation of AGCM precipitation. In this study, these metrics

are calculated at daily time steps after aggregating all sub-daily products to daily time steps. Note that the number of observations at each *in situ* station used to calculate the scores is 1827.

## 3.2 Calculation and validation of RZSM

As the *in situ* measurements are available at several specific depths (10, 20, 40 and 100 cm), the RZSM is calculated using a depth-weighted average of the four layers soil moisture layers (Xing et al., 2021). The equation is as follows:

$$\theta_{RZSM} = \frac{2\theta_1 L_1 + (\theta_1 + \theta_2)L_2 + \cdots (\theta_{n-1} + \theta_n)L_n}{2(L_1 + L_2 + L_3 + \cdots L_n)} \tag{1}$$

where $\theta_{RZSM}$ refers to the 0-100 cm RZSM ($m^3\,m^{-3}$), $\theta_n$ is the volumetric soil moisture at the $n_{th}$ observation depth ($m^3\,m^{-3}$), and $L_n$ is the soil layer thickness between adjacent observation depths (m).

For the RZSM products, in addition to the GLDAS_CLSM, MERRA-2, SMAP L4 and SMOS L4, which directly provide the 0-100 cm RZSM, other RZSM products are provided in different soil layers, NCEP CFSv2, CLDAS and GLDAS_NOAH ($\theta_{0-10c}$, $\theta_{10-40cm}$, $\theta_{40-100cm}$), ERA5 ($\theta_{0-7cm}$, $\theta_{7-28c}$, $\theta_{28-100cm}$). For example, the GLDAS_NOAH RZSM can be calculated as:

$$\theta_{RZS} = 0.1 \times \theta_{0-10cm} + 0.3 \times \theta_{10-40cm} + 0.6 \times \theta_{40-100cm} \tag{2}$$

where $\theta_{RZSM}$ denotes 0-100 cm RZSM ($m^3\,m^{-3}$), $\theta_{0-10c}$, $\theta_{10-40cm}$ and $\theta_{40-100cm}$ denote the soil moisture estimates at 0-10 cm, 10-40 cm and 40-100 cm, respectively.

## 3.3 RZSM products aggregation and validation strategies

In terms of the temporal resolution, GLDAS_CLSM and SMOS L4 products provide RZSM data at daily time intervals. NCEP CFSv2 and GLDAS_NOAH products provide RZSM data at 3-hourly and 6-hourly time interval, respectively, which don't have consistent hour of soil moisture data with *in situ* observations only available at 08:00 AM. To keep consistent, thus the other sub-daily RZSM datasets (hourly/3-hourly/6-hourly time steps, shown in Table 1) are aggregated to daily average values to match the daily sampling frequency of the *in situ* observations. In terms of spatial resolution, we did not change the spatial resolution of any RZSM products and used the RZSM time series for each grid where the *in situ* stations are located. Two validation strategies were used in the study. The first is to compare the RZSM time series averaged over all in situ stations with the RZSM time series averaged over all model grids where the *in situ* stations are located (Fig.2 and 3 shown in this study). The second one is the point-grid validation, the RZSM measurements at each in situ station are compared directly with the RZSM values for the grid where the *in situ* station is located, if there is more than one in situ station within a grid, the RZSM measurements at each station are compared to the grid values separately. The point-grid validation is provided in Fig.4 and S1.

The global precipitation and air temperature forcing data are used in the production of model-based RZSM products except for SMOS L4, which are validated against the China daily gridded ground precipitation and air temperature dataset V2.0 described in section 2.2. The soil properties data used in the eight RZSM products were all derived from the FAO/UNESCO soil map of World except for CLDAS, which used the soil data developed by Shangguan et al. (2013), and SMAP L4, which used the HWSD V1.2 soil properties over China. The China soil

dataset developed by Shangguan et al. (2013) is used as a reference to evaluate the accuracy of FAO/UNESCO and HWSD V1.2 soil properties (clay and sand content, organic carbon content and bulk density).

**3.4 Calculation of seasonal anomaly**

Soil moisture products can show large differences at different timescales (e.g. subseasonal, mean seasonal and interannual) (Draper and Reichle, 2015; Gruber et al., 2020). To avoid seasonal effects, the soil moisture products are typically decomposed into different frequency components (e.g., the raw soil moisture and monthly soil moisture anomaly). In this study, the monthly anomaly time series of the RZSM are calculated based on the moving average decomposition method. The difference from the mean is divided by the standard deviation for a moving average window of five weeks (Rüdiger et al., 2009; Albergel et al., 2012). The moving window F is defined as follows for each RZSM estimate or observation on day (t), F=[t-17:t+17]. If at least five measurements are available in this period, the moving average and the standard deviation of the root zone soil moisture are calculated. The anomaly is given by the following equation:

$$RZSM_{anomaly}(t) = \frac{RZSM(t) - \overline{RZSM(F)}}{stdev(RZSM(F))} \qquad (3)$$

where $RZSM(t)$, $RZSM_{anomaly}(t)$ and stdev, denote the raw RZSM, the seasonal anomaly of RZSM at day *t*, and standard deviation, respectively. Equation (3) is applied to gridded and *in situ* RZSM for comparison.

**Table 2.** List of the statistical metrics for evaluation of RZSM products and corresponding precipitation forcing data using *in situ* measurements.

| Statistic metrics | Unit | Equation | Optimal value |
|---|---|---|---|
| Pearson correlation coefficient ($R$) | - | $\dfrac{\sum_{i=1}^{n}\left(\theta_{est,i}-\overline{\theta_{est,i}}\right)\left(\theta_{obs,i}-\overline{\theta_{obs,i}}\right)}{\sqrt{\sum_{i=1}^{n}\left(\theta_{est,i}-\overline{\theta_{est,i}}\right)^2}\sqrt{\sum_{i=1}^{n}\left(\theta_{obs,i}-\overline{\theta_{obs,i}}\right)^2}}$ | 1 |
| Mean Bias Error (MBE) | $m^3\,m^{-3}$ | $\dfrac{\sum_{i=1}^{n}\left(\theta_{est,i}-\theta_{obs,i}\right)}{n}$ | 0 |
| Root Mean Square Error (RMSE) | $m^3\,m^{-3}$ | $\sqrt{\dfrac{\sum_{i=1}^{n}\left(\theta_{est,i}-\theta_{obs,i}\right)^2}{n}}$ | 0 |
| unbiased Root Mean Square Error (ubRMSE) | $m^3\,m^{-3}$ | $\sqrt{\dfrac{\sum_{i=1}^{n}\left(\left(\theta_{est,i}-\overline{\theta_{est,i}}\right)-\left(\theta_{obs,i}-\overline{\theta_{obs,i}}\right)\right)^2}{n}}$ | 0 |
| Probability of Detection (POD) | - | $\dfrac{H}{H+M}$ | 1 |
| False Alarm Ratio (FAR) | - | $\dfrac{F}{H+F}$ | 0 |
| Critical Success Index (CSI) | - | $\dfrac{H}{H+M+F}$ | 1 |
| Normalized RZSM ($RZSM_{nor}$) | - | $\dfrac{RZSM-RZSM_{min}}{RZSM_{max}-RZSM_{min}}$ | - |

Note: n is the number of gap-filled daily observations (1827) used at each of the 58 *in situ* stations (see Table S1). $\theta_{est,i}$ and $\theta_{obs,i}$ are RZSM products and *in situ* measurements ($m^3\,m^{-3}$), respectively ; $\overline{\theta_{est,i}}$ and $\overline{\theta_{obs,i}}$ are the means of $\theta_{est,i}$ and $\theta_{obs,i}$ over the entire research period; H is the number of precipitation events detected by model and *in situ* measurements; M is the number of measured precipitation events not recognized by the model product; F is the number of model-based precipitation events not detected by *in situ* measurements. $RZSM_{nor}$ represents the normalized RZSM, $RZSM_{min}$ and $RZSM_{max}$ represent the maximum and minimum of RZSM, respectively.

## 4 Results

### 4.1 Comparison between gridded and *in situ* RZSM

Figure 2 shows scatterplots of RZSM products against the *in situ* measurements averaged across all in situ stations over the HRB, from 1 April 2015 to 31 March 2020. The statistical metrics are shown in Table 3. Regarding the bias, except for the underestimation by SMOS L4 (-0.047 $m^3 m^{-3}$), all the other products overestimate the RZSM observations by 0.030 $m^3 m^{-3}$ to 0.117 $m^3 m^{-3}$ (SMAP L4 and ERA5, respectively). ERA5 and CLDAS have the largest RMSE values among all the RZMS products due to the large bias. Regarding correlation and ubRMSE,

GLDAS_CLSM ($R = 0.69$, ubRMSE $= 0.018$ $m^3 m^{-3}$) outperforms the other RZSM products, followed by MERRA-2, ERA5, CLDAS, SMAP L4 and GLDAS_NOAH, NCEP CFSv2 and SMOS L4. Overall, GLDAS_CLSM performs best among the eight RZSM products in terms of $R$, ubRMSE and bias values, while SMAP L4 presents the lowest RMSE and the lowest bias. SMOS L4 presents the worst performance with the lowest $R$ value. The detailed statistics are shown in Table 3.

Figure 3 shows the time series of observation- and model-based RZSM averaged over all *in situ* stations and the grids where the *in situ* stations are located. ERA5, SMOS L4 and GLDAS_CLSM show the highest overestimation, the lowest underestimation, and the best overall agreement with *in situ* observations, respectively. In general, all RZSM products capture the rapid temporal variations of the *in situ* soil moisture observations and respond well to precipitation events, except for SMOS L4, which shows less rapid changes and smoother time

series. The model-based RZSM products generally perform better in the wet season than in the dry season. While SMOS L4 performs better in the dry season than in the wet season (Fig. 4 and S1). The *in situ* RZSM observations show a variation in the range of 0.1 to 0.4 $m^3 m^{-3}$. The range of NCEP CFSv2 and SMAP L4 RZSM is similar to the observed RZSM range. ERA5 and CLDAS present larger RZSM values, ranging from 0.2 to 0.5 $m^3 m^{-3}$. MERRA-2, GLDAS_CLSM and GLDAS_NOAH RZSM values range from 0.2 to 0.4 $m^3 m^{-3}$, which is a narrower

interval compared to the other products. SMOS L4 displays the smallest RZSM values, ranging from 0.1 to 0.3 $m^3$ $m^{-3}$.

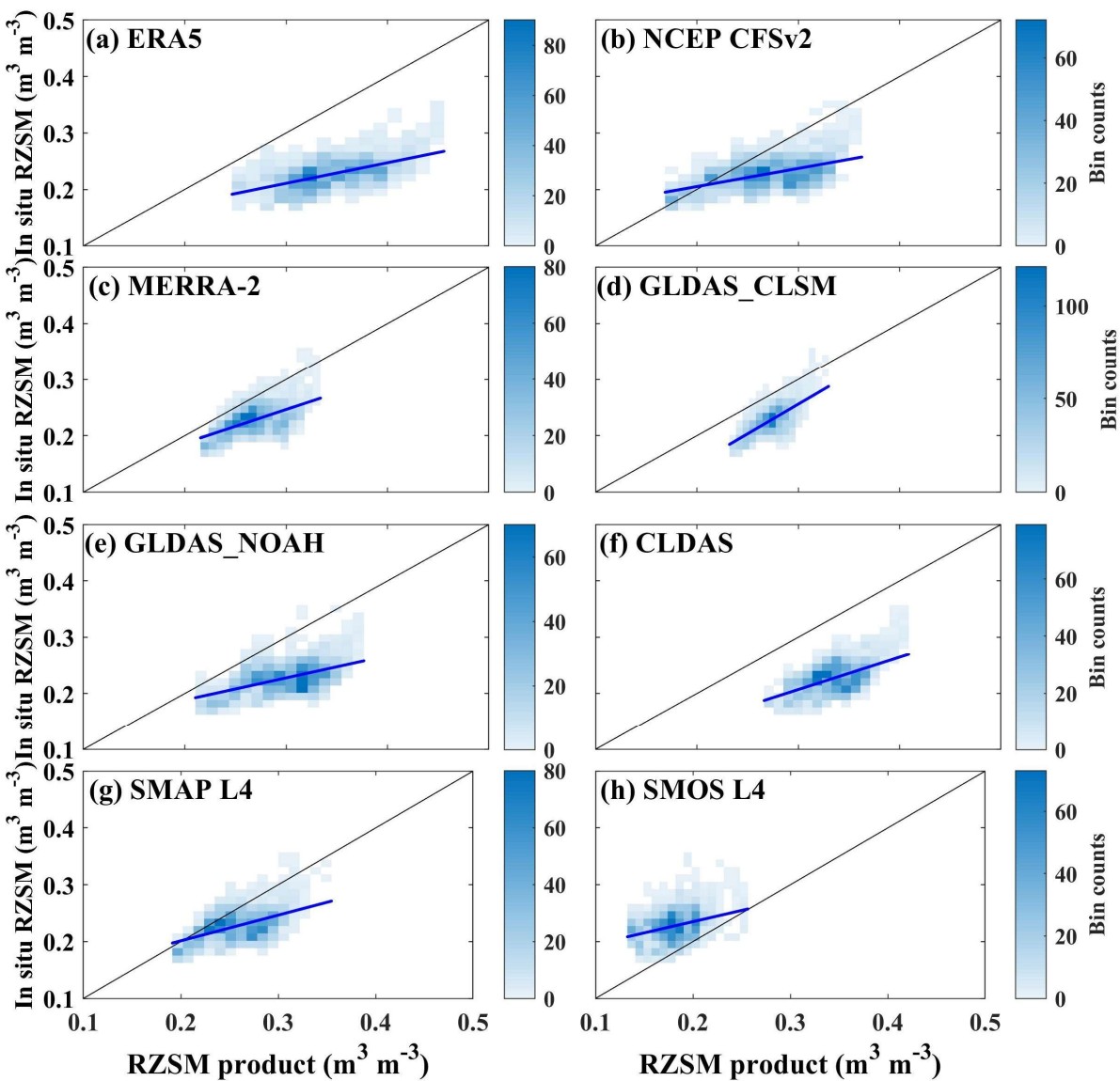

**Fig. 2** Scatterplots of RZSM products vs. *in situ* RZSM observations averaged across all in situ stations from 1 April 2015 to 31 March 31 2020. Scores are given in Table 3. Darker regions show a higher density of data point and the blue line in each subplot represents the fitted trend for the data points.

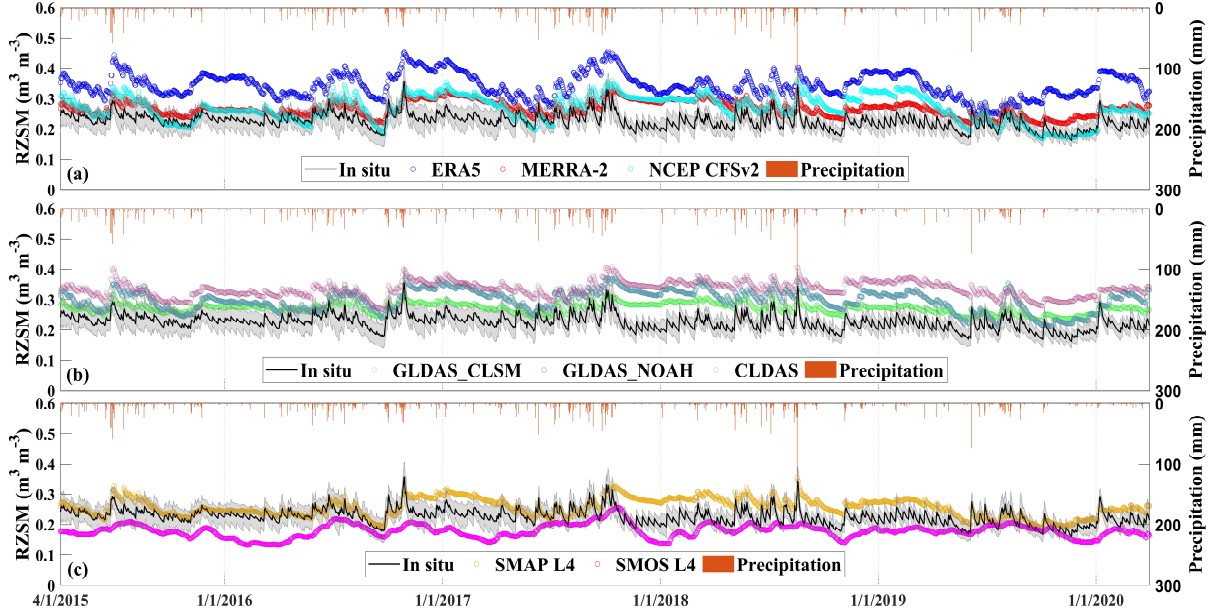

**Fig. 3** Time series of RZSM (0-100 cm) products and *in situ* soil moisture observations averaged across all in situ stations from 1 April 2015 to 31 March 31 2020. The dark line and the gray-shaded areas represent the mean and standard deviation of *in situ* stations observations. Colored lines represent different RZSM products. Daily precipitation is represented by the orange vertical bars.

**Table 3.** Statistical metrics of eight RZSM products validated by *in situ* measurements (0-100 cm) averaged over all stations from 1 April 2015 to 31 March 2020 (Fig. 2). Mean score values are given. Best score values are in bold. The number of observations used to calculate the scores is 1827 for each product.

| Dataset | *In situ* validation | | | |
| --- | --- | --- | --- | --- |
| | $R$ | Bias (m³ m⁻³) | RMSE (m³ m⁻³) | ubRMSE (m³ m⁻³) |
| ERA-5 | 0.58 | 0.117 | 0.122 | 0.033 |
| MERRA-2 | 0.58 | 0.040 | 0.046 | 0.023 |
| NCEP CFSv2 | 0.54 | 0.041 | 0.055 | 0.036 |
| GLDAS_NOAH | 0.54 | 0.071 | 0.077 | 0.030 |
| GLDAS_CLSM | **0.69** | 0.046 | 0.049 | **0.018** |
| CLDAS | 0.56 | 0.107 | 0.114 | 0.023 |
| SMAP L4 | 0.53 | **0.030** | **0.040** | 0.027 |
| SMOS L4 | 0.35 | -0.047 | 0.055 | 0.027 |

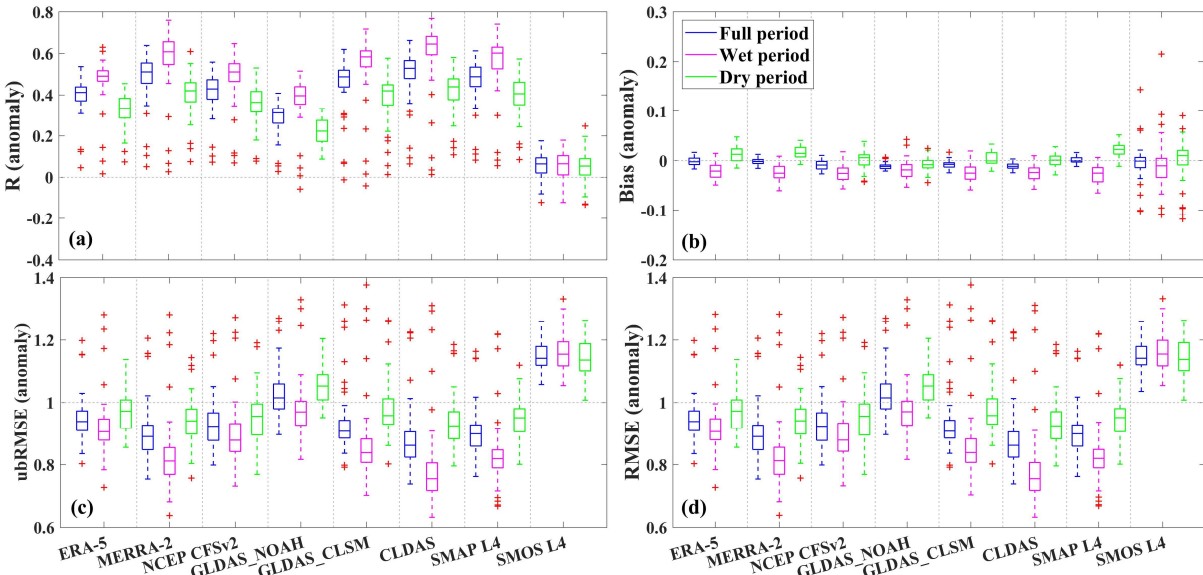

**Fig. 4 Single-station RZSM anomalies comparison between model-derived RZSM and *in situ* soil moisture observations for different periods, including the Full period (from 1 April 2015 to 31 March 2020), Wet period (from June to September) and Dry period (from October to May). Each outlier "+" represents an *in situ* station. The five horizontal lines of the box plot represent the minimum, 25th percentile, 50th percentile, 75th percentile and maximum from bottom to top, respectively.**


## 4.2 Intercomparison of gridded RZSM products

Figure 5 displays the pairwise comparison of the eight RZSM products for grid cells located above the *in situ*
stations. Overall, there is good agreement between all RZSM products, except for SMOS L4. The correlation
coefficient $R$ between each of the other seven RZSM products varies from 0.30 (MERRA-2 versus SMOS L4) to
0.95 (SMAP L4 versus MERRA-2), with an average value of 0.71. The mean bias varies from -0.067 $m^3\,m^{-3}$
(MERRA-2 minus CLDAS) to 0.165 $m^3\,m^{-3}$ (ERA5 minus SMOS L4), with an average value of 0.037 $m^3\,m^{-3}$. The
ubRMSE varies from 0.010 $m^3\,m^{-3}$ (MERRA-2 versus SMAP L4) to 0.040 $m^3\,m^{-3}$ (NCEP CFSv2 versus SMOS
L4), with an average value of 0.024 $m^3\,m^{-3}$. SMOS L4 differs most from the other products. The correlation
coefficient $R$ between SMOS L4 and the other seven RZSM products varies from 0.30 (MERRA-2 vs. SMOS L4)
to 0.41 (GLDAS_NOAH versus SMOS L4), with an average value of 0.35, and the mean bias varies from 0.077
$m^3\,m^{-3}$ (SMAP L4 minus SMOS L4) to 0.165 $m^3\,m^{-3}$ (ERA5 minus SMOS L4), with an average value of 0.112 $m^3$
$m^{-3}$. The ubRMSE varies from 0.023 $m^3\,m^{-3}$ (GLDAS_CLSM versus SMOS L4) to 0.400 $m^3\,m^{-3}$ (NCEP CFSv2
versus SMOS L4), with an average value of 0.031 $m^3\,m^{-3}$.

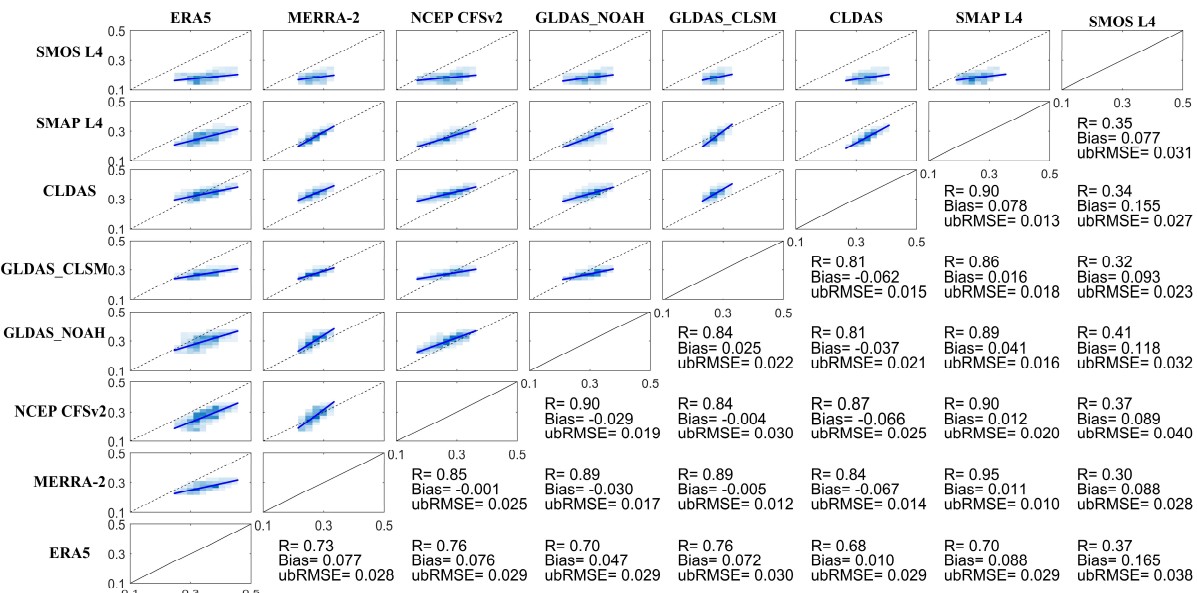

**Fig. 5 Comparison of different RZSM products (volumetric water content, $m^3\,m^{-3}$) with each other. The scatterplots
and their corresponding statistics are located on opposite sides of each other, that is, the scatterplot of the data pair
SMOS L4-ERA5 is in the top left-hand corner, while the respective statistical values are found in the bottom right-**
**hand corner. Darker regions show a higher density of data point and the blue line in each subplot represents the fitted
trend for the data points.**

Figure 6 shows the histograms of the normalised RZSM of the eight products and the *in situ* observations.
The relative frequency distribution corresponding to the normalized soil moisture interval varies considerably
between the different RZSM datasets. All soil moisture datasets are almost normally distributed with a clear peak.
The observed RZSM distribution is skewed towards low values and has a peak frequency around 0.3. The
MERRA-2, GLDAS_CLSM and SMAP L4 products exhibit the similar distribution patterns with a peak frequency
around 0.4. In contrast, the frequency distribution of the other RZSM products show an obvious offset towards
wet soil moisture compared to the in situ observations, with a peak frequency in the range of 0.4 to 0.5. In particular,
GLDAS_NOAH shows a peak frequency in the range of 0.6 to 0.7, and is clearly skewed towards the wetter end

of the distribution. It is obvious that the histograms of MERRA-2, GLDAS_CLSM and SMAP L4 show better agreement with the *in situ* observations than the other RZSM products, although they slightly overestimate the frequency of wet soil moisture. However, they all don't capture the peak frequency and underestimate the peak frequency of normalized soil moisture ranging from 0.2 to 0.4. The other RZSM products show significant overestimation of frequency of wet soil moisture, underestimation of dry soil moisture and of peak frequency.

Therefore, the Richard's equation used to simulate the water content in different soil layers in LSMs should focus on producing less wet soil moisture and more dry soil moisture to obtain a more accurate frequency distribution of modelled soil moisture by modifying the soil water retention curve or changing the initial and boundary conditions.

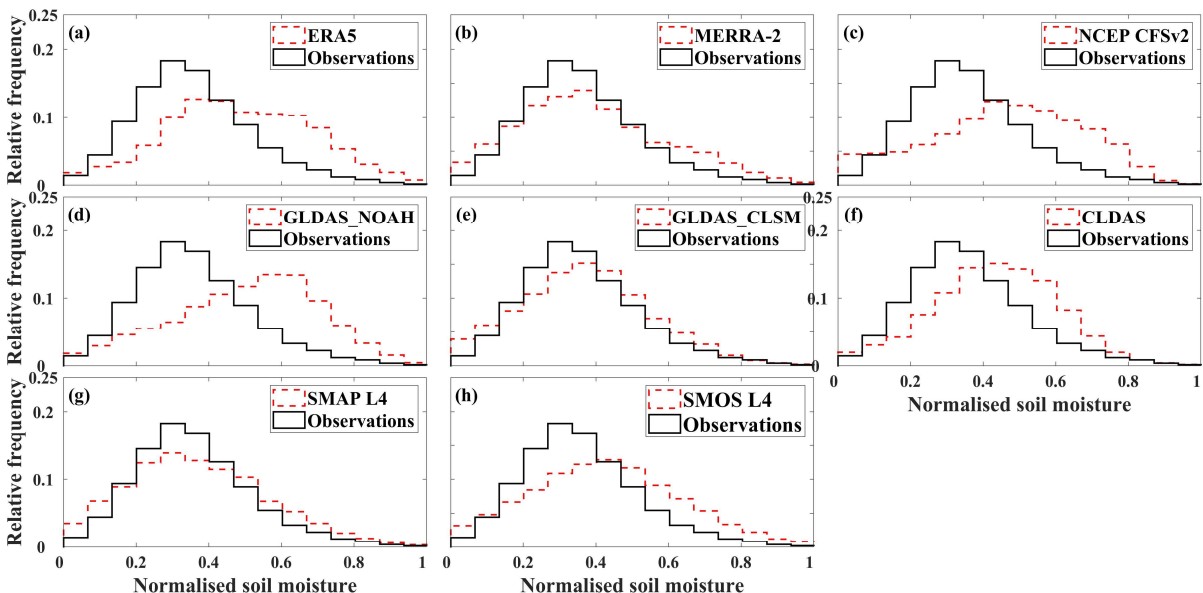


**Fig. 6 Histograms of the relative frequency (vertical axis) of the various normalized RZSM datasets and *in situ* observations.**

**4.3 Validation of atmospheric forcing and soil properties**

4.3.1 Precipitation and air temperature

Figure 7 shows the differences between the model and ground-based precipitation. A daily precipitation amount of less than 1 mm is considered as a no-rain criterion. During the period from 1 April 2015 to 31 March 2020, the annual mean precipitation amount from global products (SMAP: 1024 mm yr$^{-1}$, GLDAS_NOAH: 988 mm yr$^{-1}$, MERRA-2: 974 mm yr$^{-1}$, NCEP CFSv2: 951 mm yr$^{-1}$, GLDAS_CLSM: 912 mm yr$^{-1}$, ERA5: 880 mm yr$^{-1}$) overestimate the ground-based observations (840 mm yr$^{-1}$) by 22, 17, 16, 13, 9 and 5 %, respectively. In addition,

the mean frequency of rainy days (131, 114, 105, 113, 114, 126 d yr$^{-1}$) is larger than observed (97 d yr$^{-1}$) due to the drizzle effect often produced by AGCM (Piani et al., 2010; Velasquez et al., 2020). In contrast to the global products mentioned above, CLDAS (806 mm yr$^{-1}$) slightly underestimates the mean annual precipitation amount by 4 %, and the precipitation frequency (99 days yr$^{-1}$) is close to the ground-based observation. Furthermore, the global precipitation products tend to underestimate the *in situ* precipitation observations for precipitation events

above 50 mm d$^{-1}$ (Fig. 7). Overall, the *R* values between precipitation products and the observed precipitation are

higher than 0.4 (left panel of Fig. 8). MERRA-2, ERA5, GLDAS_CLSM, and SMAP L4 show strong ability to detect precipitation with POD value above 0.6 (right panel of Fig. 8). The *R* value between modelled and ground-based precipitation is directly related to the CSI value except for GLDAS_NOAH.

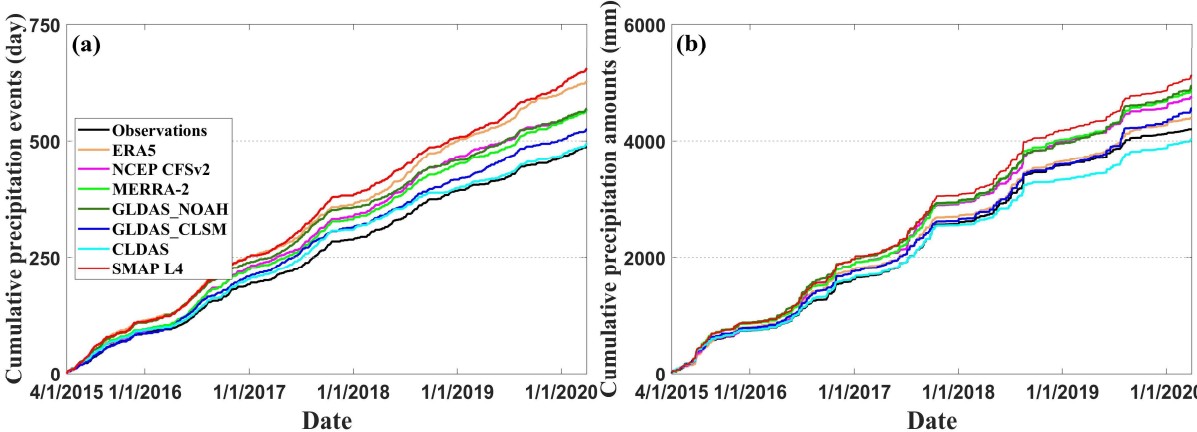

**490    Fig. 7 Comparison of cumulative precipitation events and cumulative precipitation amounts between model-derived precipitation and *in situ* precipitation observations averaged over all *in situ* stations from 1 April 2015 to 31 March 2020.**

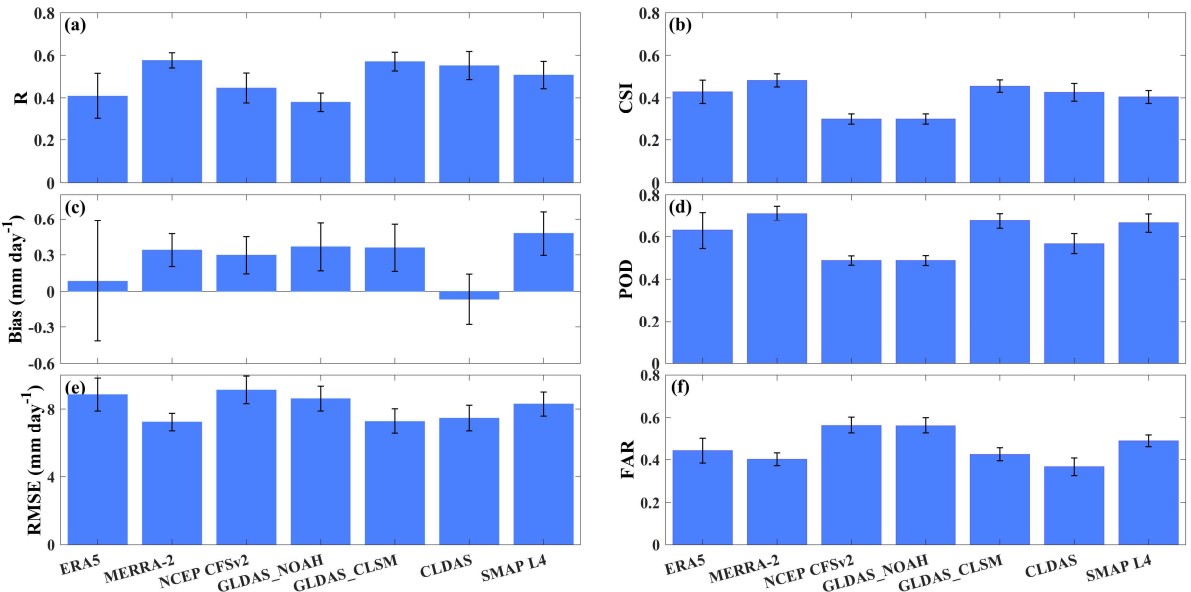

**Fig. 8 Summary of error metrics of gridded precipitation data against *in situ* precipitation observations (left panel),**
**495    right panel shows the detection ability of gridded precipitation to reproduce the observed precipitation. The blue histogram represents the median and black error bar represents the standard deviation.**

The daily air temperature data derived from ERA5, MERRA-2, NCEP CFSv2, GLDAS_CLSM, CLDAS, GLDAS_NOAH and SMAP L4 are validated against *in situ* observations of daily air temperature after aggregating all sub-daily products to daily time steps. Figures 9 and S2 shows that the modelled air temperature captures the
observed temporal variation well, with *R* values above 0.96. However, all of them show underestimation, indicated by negative bias values ranging from -4.0 to -5.2 K. In terms of the comprehensive scores of the four statistical metrics, GLDAS_NOAH air temperature outperforms the other datasets and SMAP L4 shows the worst scores. Detailed statistics are shown in Table 4.

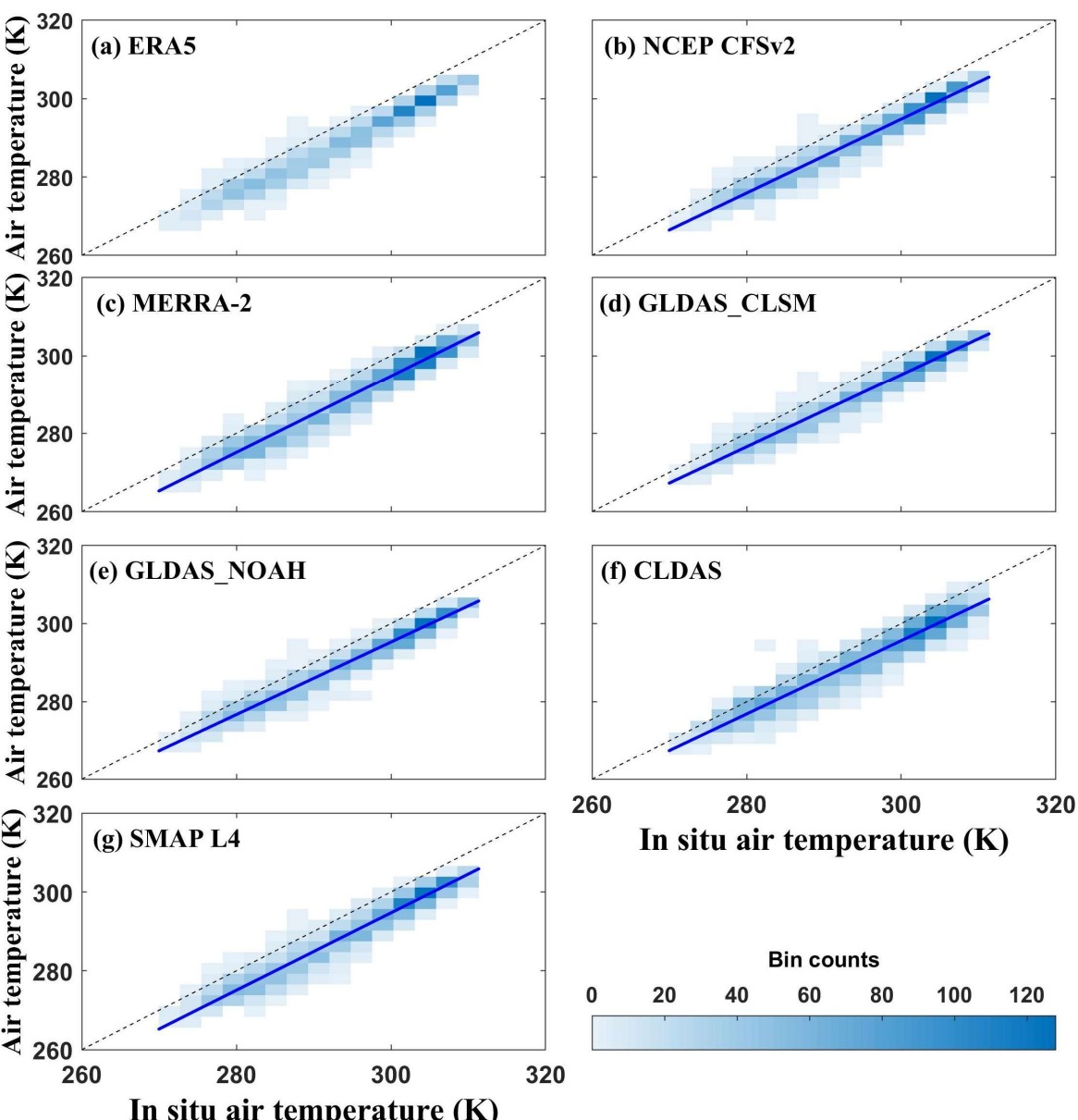

Fig. 9 Scatterplots of model- and observation-based air temperature averaged over all stations, from 1 April 2015 to 31 March 2020. ERA5, MERRA-2, NCEP CFSv2, GLDAS_CLSM, GLDAS_NOAH, CLDAS and CMA products provide the air temperature datasets at the 2-m screen level. SMAP L4 product provides the air temperature at center height of the lowest atmospheric model layer. Darker regions show a higher density of data point and the blue line in each subplot represents the fitted trend for the data points.

**Table 4.** Statistical metrics of air temperature products validated by *in situ* measurements averaged over all stations from 1 April 2015 to 31 March 2020. Mean score values are given. Best score values are in bold. The number of observations used to calculate the scores is 1827 for each product.

| Dataset | | *In situ* validation | | |
| | R | Bias (K) | RMSE (K) | ubRMSE (K) |
| --- | --- | --- | --- | --- |
| ERA-5 | 0.98 | -4.8 | 5.2 | 2.1 |
| MERRA-2 | 0.98 | -5.1 | 5.7 | 2.4 |
| NCEP CFSv2 | 0.98 | -4.9 | 5.3 | 2.1 |
| GLDAS_NOAH | 0.98 | -4.3 | 4.8 | 2.1 |
| GLDAS_CLSM | 0.98 | -4.5 | 4.9 | 2.1 |
| CLDAS | 0.96 | -4.0 | 4.9 | 2.8 |
| SMAP L4 | 0.97 | -5.2 | 5.7 | 2.4 |

4.3.2 Soil properties

In this study, four soil properties indicators, including clay and sand content, organic carbon content and bulk density were selected to investigate the differences among the FAO/UNESCO soil map of World, HWSD, and the reference soil dataset developed by Shangguan et al. (2013). Figure 10 shows the reference dataset and HWSD generally exhibit similar properties, although the reference dataset has slightly higher organic carbon content and lower sand content. Both of them differ from the FAO/UNESCO soil properties data obviously. FAO/UNESCO overestimates the clay content for the upper (0-30 cm) and subsurface (30-100 cm) soil layers. Sand content is also overestimated for the subsurface layer but it is underestimated for the surface layer. Besides, FAO/UNESCO overestimates significantly the organic carbon content for both layers, resulting in the underestimated bulk density.

4.3.3 The mismatch of spatial and temporal scales

In addition to the model- and the observation-based soil moisture errors, the mismatch of spatial scales between grid-scale soil moisture simulations and point-scale observations also introduces additional errors. The eight RZSM products are evaluated against *in situ* observations using two validation strategies described in section 3.3. The statistical scores for spatial-average validation are generally better than that for point-grid validation, which are shown in Tables 3 and S1, respectively. For the point-grid validation, the spatial representativeness of *in situ* soil moisture observations at the grid scale is insufficient due to the heterogeneity of the underlying surface and precipitation forcing. This leads to an error in representativeness (Xia et al., 2014). In contrast, the spatial-average validation improves the representativeness of the grid-based RZSM and reduces the spatial noise (Wang and Zeng, 2012; Xia et al., 2014; Bi et al., 2016; Zheng et al., 2022). In addition, upscaling the sparse ground-based observations to the footprint-scale satellite soil moisture retrieval or model grid scale through the temporal stability concept, block kriging, field campaign data, or LSM, reduces the uncertainty of spatial resampling and further improves the reliability of soil moisture validation (Crow et al., 2012). Finally, the temporal mismatch between model-based RZSM values which are aggregated to daily average values and *in situ* observations available at 08:00 AM could also induce partial bias, but this type of bias is generally small due to the low variability of soil moisture during the day.

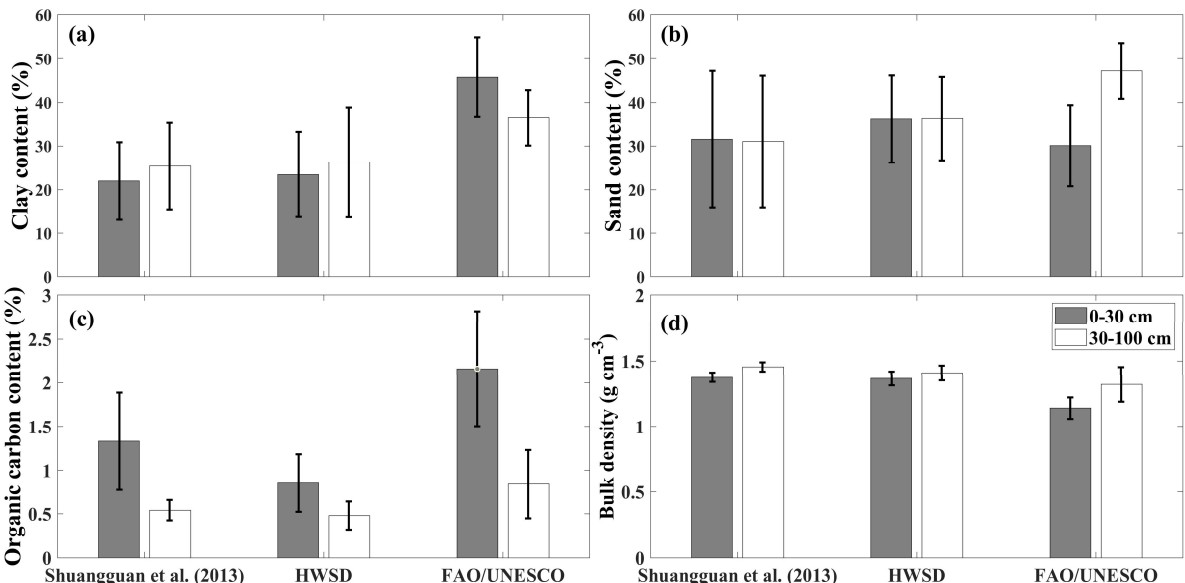

**Fig. 10 Comparison of three sets of soil properties data (FAO used in ERA5, MERRA2, NCEP CFSv2, GLDAS_NOAH, GLDAS_CLSM and SMOS), HWSD used in SMAP L4 and reference soil properties data Shangguan et al. (2013) used in CLDAS. The histogram (gray bar: 0-30 cm; white bar: 30-100 cm) represents the median and black error bar represents the standard deviation.**

## 5 Discussion

### 5.1 What is the impact of uncertainties of meteorological forcing data?

The accuracy of LSM simulations is influenced by the quality of the meteorological forcing, which is considered to be one of the most important and direct factors, especially precipitation and air temperature (Reichle et al., 2012; Yang et al., 2020; Zeng et al., 2021). In different LSMs, the diffusive form of Richard's equation is used to describe the vertical movement of water in the soil column. Precipitation serves as the upper boundary condition to regulate
the temporal dynamics of soil moisture. Therefore, the overestimation of precipitation amounts and the frequency of precipitation events (the wet bias excluding CLDAS) could be a reason for the overestimation of soil water simulated by the model-based RZSM products. We also investigate the effect of precipitation accuracy on the performance of RZSM products (Fig. 8). In terms of $R$, RMSE, CSI, POD and FAR, MERRA-2 and GLDAS_CLSM precipitation are the best performing products. This may explain the relatively better agreement
of MERRA-2 and GLDAS_CLSM RZSM with *in situ* data in terms of correlation (Table 3), as precipitation dominates the dynamics change of soil moisture. The low CSI and high FAR and the overestimated precipitation frequency indicate that the precipitation for each grid derived from AGCM has more rainy days and less dry days and struggles to reproduce the temporal pattern of the precipitation observed at each rain gauge, resulting in the relatively large RMSE values in precipitation generally above 7 mm day$^{-1}$. This could also explain the low
correlation R ranging from 0.4 to 0.6, although the daily average bias in model-based precipitation is less than 0.5 mm day$^{-1}$. For most reanalysis products, the precipitation used to drive the different LSMs was generated by the AGCM through the assimilation of atmospheric temperature, humidity and wind observations (Reichle et al., 2017d). Before driving the land surface water budget, the MERRA-2 model background precipitation was

corrected using CPCU gauge-based precipitation analysis in the coupled land-atmosphere reanalysis system. The correction leads to more accurate precipitation fields for MERRA-2, and then to more realistic RZSM simulations. Being driven by *in situ* precipitation observations, the CLDAS multi-LSMs should have produced RZSM values close to the observations. However, the CLDAS RZSM product overestimates the *in situ* observations by 0.107 $m^3 m^{-3}$ (Table 3). Therefore, precipitation may not be the dominant factor contributing to the overestimation of RZSM for the CLDAS RZSM (Bi et al., 2016; Qin et al., 2017).

Air temperature is another key factor in determining the accuracy of RZSM simulations, as it controls soil evaporation and plant transpiration. The agreement between model- and observation-based air temperature is much better than for precipitation due to the high spatial heterogeneity in precipitation. The underestimation of air temperature by reanalyses has been illustrated in previous studies (Wang and Zeng, 2012; Yang et al., 2020). In general, the lower air temperature results in less evapotranspiration, and more soil water storage. Compared to precipitation, air temperature has an overall better correlation with *in situ* observations. Note that ERA5 includes an analysis of soil moisture and screen-level (2 m) air temperature and air humidity. Studies have indicated that the assimilation of screen-level variables improves root zone soil moisture estimates relative to *in situ* observations providing more realistic lower boundary conditions for numerical prediction models (Douville et al., 2000; Seuffert et al., 2003; de Rosnay et al., 2012).

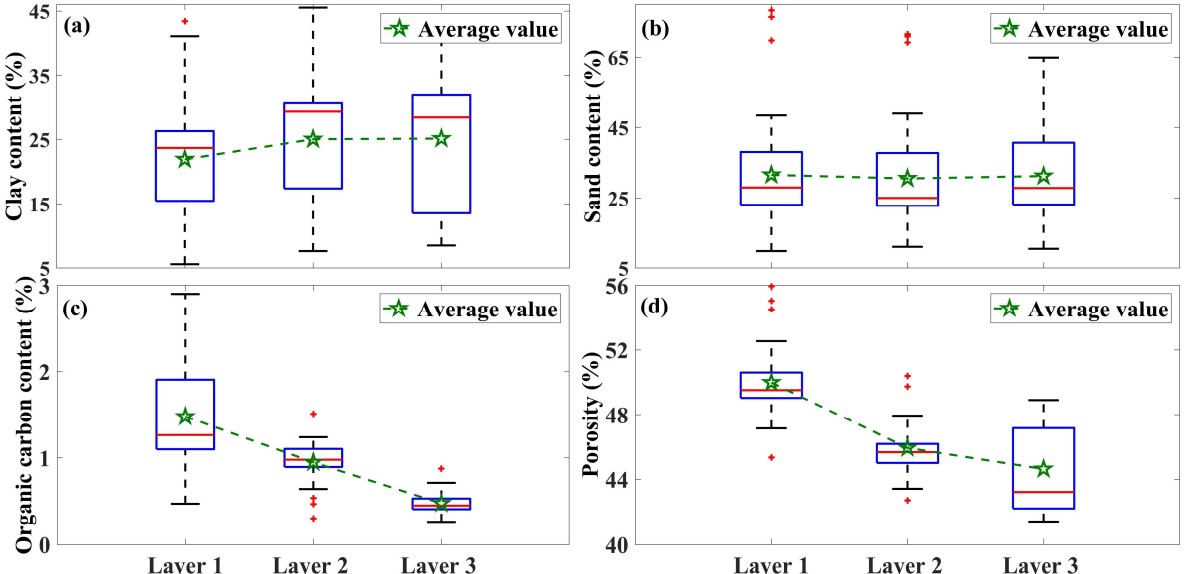

**Fig. 11 Boxplot of soil properties for three soil layers at all *in situ* stations (Layer1 (0-16.6 cm): plough layer; Layer 2 (16.6-49.3 cm): black soil layer; Layer3 (49.3-138.3 cm): lime concretion layer).**

**5.2 Are soil properties correctly represented?**

Time-invariant soil property data (e.g. porosity) are key model parameters for LSMs because they determine the physical structure of the soil in the vadose zone, which controls the partitioning of precipitation into surface runoff and infiltration. In general, soil texture is closely related to the ability of the soil to retain water, as water molecules adhere more tightly to fine-textured clay particles than coarse-textured sand particles. Consequently, clay exhibits stronger water retention capacity and higher water content stored in the soil compared to sand at the same matric potential. Meanwhile, the sandy soil shows the better drainage capacity and higher hydraulic conductivity than clay soil. In addition, the overestimated FAO/UNESCO soil organic carbon content (Fig. 10) leads to higher soil

porosity and lower bulk density. As a result, water can infiltrate more quickly and more water can flow through the soil and can be retained in the soil (Bot and Benites, 2005; Reichle et al., 2017b). Therefore, the use of inaccurate FAO/UNESCO soil property data used in LSMs may explain the overestimation of soil moisture by the various RZSM products compared to the ground-based observations. It is promising to improve the accuracy of LSM-based RZSM using HWSD instead of FAO/UNESCO soil property data. The soil hydraulic parameters (SHPs), such as the hydraulic conductivity and matric potential, are crucial parameters to describe the vertical transport of water in the soil column through the Richard's equation employed in the LSMs. Generally speaking, the SHPs are derived from a combination of soil properties (clay, sand, silt fractions and organic content, etc.) with pedotransfer functions (PTFs), which can be constructed by multivariate regression models, nonlinear regression models or artificial neural networks (Harrison et al., 2012). Therefore, different input variables and functional forms of the continuous PTFs are used to derive SHPs in the LSMs. The Richard's equation relying on the SHPs shows great uncertainty in the simulated soil moisture. For example, the HWSD soil properties used in SMAP L4 are more consistent with the reference dataset than FAO soil properties used in MERRA-2 by revising the underestimated sand content and the overestimated clay content in FAO. In addition, SMAP L4 adopts PTFs from Wösten et al. (2001) which take into account the organic carbon affecting soil hydraulic and thermal properties. MERRA-2 adopts PTFs adapted from Cosby et al. (1984) without considering organic carbon (De Lannoy et al., 2014). The revised soil parameters and new PTFs employed in SMAP L4 yield smaller shape parameter of water retention curve and result in less water retention than in MERRA-2, and increase the hydraulic conductivity. Thus, SMAP L4 has the smaller soil moisture estimates and less RZSM bias against in situ measurements than MERRA-2, which is consistent with the result of this study. Therefore, the soil properties and PTFs could also explain part of the uncertainty.

Soil stratification can affect the accuracy of LSM-based RZSM by impeding the water transfer from the surface layer to the root zone layer. In the Huaibei plain, the soil column can basically be divided into three layers, including the plough layer (0–16.6 cm), the black soil layer (16.6–49.3 cm) and the lime concretion layer (49.3–138.3 cm) due to the long-term human activities (e.g. fertilisation and ploughing), which significantly increases the soil organic carbon content and porosity in the plough layer compared to the deeper soil layer (Zhang et al., 2001; Li et al., 2011; Zha et al., 2015; Gu et al., 2021). There is a noticeable difference in soil properties between the plough layer and the black soil layer, while the difference between the black soil layer and the lime concretion layer is relatively small (see Fig. 11). High porosity results in high hydraulic conductivity and infiltration capacity (Zha et al., 2015). Therefore, interflow can occur due to the difference of infiltration rate between adjacent soil layers. The interflow may either flow horizontally due to good lateral drainage conditions or accumulate vertically and evaporate. These processes may not be well represented by LSMs.

In the study by Fan et al. (2022), RZSM products (SMAP-L4 V6, ERA5-land V2, GLDAS-Noah V2.1) were evaluated over croplands in Jiangsu province, which is close to the Huaibei Plain. A fourth RZSM dataset was derived from the ESA CCI SSM using an exponential filter. In this study, SMAP L4, ERA5 and GLDAS_NOAH overestimate the *in situ* RZSM. Overall, both studies show similar *R* values of RZSM products against the *in situ* observations, but with opposite biases. The changes in the sign of the bias could be attributed to differences in soil properties (see Fig. 11). In the Huaibei plain, the main soil type is lime concretion black soil, whose main characteristic are (1) soil stratification, (2) poor soil permeability and water retention capacity due to high clay content, (3) clay swelling during wet periods and shrinking during dry periods. For a given soil profile, porosity

decreases with depth and clay content increases with depth, resulting in a decrease in hydraulic conductivity. Expansive montmorillonite clay minerals are the main constituents of the lower black soil layer, giving the soil strong expansion and contraction and a high dry bulk density. During drought, cracks in the soil column widen and deepen, resulting in capillary breakage. This makes it difficult for groundwater and RZSM to recharge crops, even though the groundwater is shallow. In addition, the increased cracks in the soil column exacerbates the evaporation of soil moisture in the root zone, ultimately leading to frequent droughts. During wet periods, when precipitation or irrigation occurs, the soil absorbs water and swells, closing the cracks and preventing water infiltration. Water is then lost mainly through surface runoff. The crops are prone to waterlogging disasters. This could explain the lower RZSM values ranging from 0.2 to 0.3 $m^3\,m^{-3}$ observed in the Huaibei plain and the higher RZSM values ranging from 0.3 to 0.4 $m^3\,m^{-3}$ observed in Jiangsu. The larger amount of precipitation in Jiangsu could be another possible reason.

**5.3 What is the impact of vegetation representation in LSMs?**

Vegetation also plays a crucial role in the exchange of water, energy and carbon between the land surface and the atmosphere, which has significant effect on the simulation of soil moisture by LSMs. First, the land cover map describes the distribution and fractions of different land use types, which have different impact on the partitioning of net solar radiation into ground heat, sensible and latent heat fluxes, and the partitioning of precipitation into canopy interception, runoff and infiltration. The land cover maps employed in the LSMs are different. For example, GLDAS_NOAH uses the modified IGBP MODIS (Moderate Resolution Imaging Spectroradiometer) 20-category vegetation classification, and GLDAS_CLSM uses the University of Maryland (UMD) land cover classification based on AVHRR (Advanced Very High Resolution Radiometer) land cover map  and MERRA-2 and HTESSEL both use the global land cover characteristics database, version 2.0 (Reichle et al., 2017c; Rui et al., 2021). Second, the parameterization for vegetation canopy (e.g., leaf area index, bare soil fraction, high- and low-vegetation fraction, type and density, Nogueira et al. (2020)) and root tissue (root distribution, rooting depth, root density and root zone water storage, Gao et al. (2014), Stevens et al. (2020) and van Oorschot et al. (2021)) varies considerably across different LSMs. The discrepancy in land cover types, vegetation canopy and root parameterizations between different land cover maps not only affects the exchange of water, carbon and energy between land surface and atmosphere at the local scale, but also affects the water and carbon cycle, and energy balance at the terrestrial and global scales. Moreover, the inaccurate partitioning of the total terrestrial evapotranspiration into soil evaporation, canopy interception and vegetation transpiration also affects the exchange of water and energy between the land surface and the atmosphere. Generally speaking, the ratio of transpiration to the total terrestrial evapotranspiration is underestimated compared to the observations in most earth system models (ESMs) (Feng et al., 2023). This phenomenon could be related to the excessive reliance on the surface soil moisture and canopy-intercepted water storage rather than the adequate utilization of RZSM for transpiration, which leads to the overestimated RZSM (Dong et al., 2022), or the unreliable representation of canopy light use, interception loss and root water uptake processes in the ESMs (Lian et al., 2018). In different LSMs, the process representing the partitioning of the total terrestrial evapotranspiration into different components differs from each other. For example, GLDAS_CLSM shows the higher fraction of soil evaporation, while GLDAS_NOAH shows the higher fraction of transpiration over the Huai River Basin (Feng et al., 2023). In general, soil evaporation is mainly controlled by surface soil moisture, while the transpiration is controlled by the available water in the root zone. Therefore, the soil

evaporation fraction is inversely proportional to leaf area index, while the transpiration fraction is proportional to leaf area index. The difference in the fractions of evapotranspiration components between GLDAS_CLSM and GLDAS_NOAH could be related to the model parameterization associated with sol evaporation and transpiration. Furthermore, the transpiration of crops is highly dependent on the growing season, which might be not well represented in the LSMs.

**5.4 What are the difference between the three CLSM-based RZSM products?**

Regarding the *in situ* validation in Sect. 4.1, the superior skill metrics of GLDAS_CLSM among the three CLSM-based RZSM products (GLDAS_CLSM, SMAP L4 and MERRA-2), can be attributed to its more accurate representation of precipitation. While GRACE TWS observations have been assimilated into GLDAS_CLSM, previous studies have indicated that the assimilation of GRACE TWS has no or negligible effect on RZSM. This
could be attributed to the faster response of soil moisture to atmospheric forcing than groundwater (Zaitchik et al., 2008; Houborg et al., 2012; Girotto et al., 2016), the short *in situ* data record or insufficient spatial sampling (Li et al., 2012). Tian et al. (2017) and Tangdamrongsub et al. (2020) jointly assimilated terrestrial water storage (GRACE TWS) and SSM products. The soil moisture-only assimilation improved the performance of soil moisture estimates relative to *in situ* measurements but degraded the performance of groundwater estimates. The GRACE-
only assimilation only enhanced the skill metrics of groundwater estimates.

Regarding the intercomparison in Sect. 4.2, the very good correlation and low ubRMSE between MERRA-2 and SMAP L4 shown in Fig. 5 can be partly attributed to the fact that both products are based on the CLSM and both use atmospheric forcing data generated from GEOS-5. However, it should be noted that SMAP L4 uses a more recent version of CLSM with a different representation of soil hydraulic and thermal properties. In addition,
MERRA-2 and SMAP L4 use different model background precipitation (i.e. GEOS-5 FP system for SMAP L4 and GEOS-5 FP-IT system for MERRA-2) (Reichle et al., 2017d). In MERRA-2, the CPCU precipitation is used in its native climatology to correct the GEOS FP-IT model background precipitation, while in SMAP L4 the CPCU precipitation is rescaled to the climatology of the GPCPv2.2 pentad precipitation product climatology before being corrected by the GEOS-5 FP system.

**5.5 Why does SMOS L4 underestimate RZSM?**

The SMOS L4 RZSM is derived from the SMOS L3 3-day SSM by applying a modified exponential filter (Pablos et al., 2018). Figure 12 shows the comparison of the SMOS L3 SSM and L4 RZSM with the *in situ* soil moisture observations. It is evident that both SMOS L3 SSM and L4 RZSM underestimate the *in situ* observations with average bias values of -0.069 and -0.047 $m^3\,m^{-3}$, respectively. By partitioning the total error composed of the
exponential filter model and the inherent SMOS *in situ* differences, Ford et al. (2014) have shown that the mismatch between *in situ* observations and the estimates is much larger than the error caused by the exponential filter method. The underestimation of *in situ* observations by SMOS L3 SSM has been reported in previous studies (Djamai et al., 2015; Cui et al., 2017; Pablos et al., 2018; Ma et al., 2019; Wang et al., 2021b). Therefore, it can be inferred that the underestimation of *in situ* observations by the SMOS L3 SSM propagates to the SMOS L4
RZSM. The L-band microwave signal is sensitive to soil moisture, soil temperature and vegetation optical depth (VOD) (Kerr et al., 2012). Using the L-band Microwave Emission of the Biosphere (L-MEB) model (Wigneron et al., 2021), SMOS L3 soil moisture and Vegetation Optical Depth (VOD) can be retrieved simultaneously from

multiple orbits using multi-angular (~0-60°) and dual-polarisation TB measurements (Al Bitar et al., 2017; Li et al., 2021). Numerous studies have shown that the SMOS L3 physical surface temperature used in the forward radiative transfer model was underestimated (Cui et al., 2017; Ma et al., 2019; Wang et al., 2021b; Zheng et al., 2022). In the SMOS L3 retrieval algorithm, underestimation of soil temperature leads to overestimation of soil emissivity, which ultimately results in the underestimation of soil moisture retrieval. In general, the SMOS L3 VOD retrievals are relatively noisy, which may be related to retrieval instabilities and Radio Frequency Interference (RFI) effects (Cui et al., 2017; Wang et al., 2021b; Wigneron et al., 2021; Zheng et al., 2022). Therefore, it is difficult to quantify its relationship with soil moisture. In addition, the ECMWF ERA-Interim soil moisture is also used in the operational SMOS L3 SSM retrieval algorithm. For a given pixel, the total TB is simulated as the sum of several fractional contributions ($F_{NO}$: nominal (bare soil, low vegetation), $F_{FO}$: forest, and others as urban, water, etc.), i.e. $TB_{total} = TB_{FNO} + TB_{FFO} + TB_{others}$ (Fernandez-Moran et al., 2017). SMOS L3 retrievals are computed only over a fraction of the pixel (the "dominant" fraction where SM retrieval is meaningful over certain surface types) (Fernandez-Moran et al., 2017; Wigneron et al., 2021). For the remaining fraction of pixels, only their contributions to the total signal need to be estimated using the ECMWF ERA-Interim SM (0-7 cm) as an auxiliary input, but no SM retrievals are performed. Previous studies have shown that the ERA-Interim soil moisture over China is overestimated (Yang et al., 2020; Ling et al., 2021). Therefore, the overestimated ECMWF ERA-Interim SM (0-7 cm) leads to an underestimation of the forest $TB_{FFO}$ contribution, which in turn leads to an overestimation of $TB_{FNO}$ and to a dry bias in the retrieved SMOS L3 SM (as there is a negative correlation between brightness temperature and soil moisture (Rao et al., 2007)).

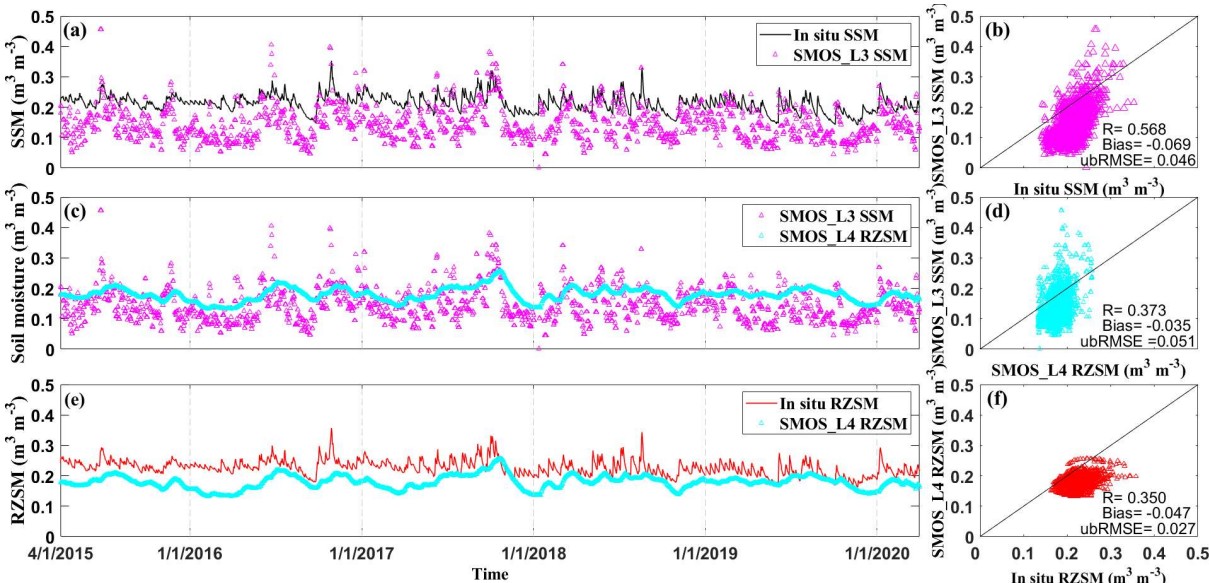

**Fig. 12 Comparison of time series (left panel) and scatterplots (right panel) of SMOS L3 SSM vs. *in situ* SSM (Fig. 12a and b), SMOS L3 SSM vs. SMOS L4 RZSM (Fig. 12c and d) and SMOS L4 RZSM vs. *in situ* RZSM (Fig. 12e and f).**

## 6 Conclusion

In this study, eight RZSM products were quantitatively evaluated against observations from 58 *in situ* soil moisture stations over the HRB in China. The impact of several potential confounding factors on the uncertainty of RZSM products was investigated, including meteorological forcing variables, soil properties, soil stratification, vegetation

parameterization and spatial scale mismatch. Nevertheless, there are still some shortcomings to be overcome in
this study. The land cover type affects the dynamics of soil moisture, future study should focus on the effect of
different land cover types on soil moisture simulation. The main conclusions of this study are as follows:

(1) GLDAS_CLSM outperformed the other RZSM products over the HRB, followed by MERRA-2, CLDAS,
SMAP, ERA5, NCEP CFSv2, and GLDAS_NOAH. The SMOS L4 product presented the worst performance due
to the fact that SMOS L4 does not contain precipitation information and has a weaker response to precipitation.
Seven RZSM products based on land surface models overestimated the *in situ* observations with median bias
values ranging from 0.033 $m^3 m^{-3}$ (SMAP L4) to 0.116 $m^3 m^{-3}$ (CLDAS). While SMOS L4 underestimated the
RZSM with a median bias value of -0.050 $m^3 m^{-3}$.

(2) The intercomparison of RZSM products shows that the correlation coefficient *R* between any two of the
seven model-based RZSM products varied from 0.68 (ERA5 vs. CLDAS) to 0.95 (SMAP L4 vs. MERRA-2). In
contrast, SMOS L4 presented a lower correlation with the other seven RZSM products with *R* ranging from 0.30
(MERRA-2) to 0.41 (GLDAS_NOAH). The comparison of the frequency distribution between eight RZSM
products and *in situ* observations indicates that all RZSM products overestimate the frequency of wet soil moisture
and underestimate the frequency of dry soil moisture. Besides, the frequency peaks of eight RZSM products are
underestimated and show an obvious offset towards wet soil moisture compared to the *in situ* observations.
Therefore, the Richard's equation in LSMs should focus on producing less wet soil moisture and more dry soil
moisture.

(3) Except for CLDAS, the overestimated RZSM products based on land surface models could be associated
with the overestimated precipitation amounts and frequency, underestimated air temperature and ratio of
transpiration to the total terrestrial evapotranspiration existing in most earth system models, which consumes less
water in the root zone for transpiration. The underestimation of the SMOS L4 RZSM is related to the
underestimation of the SMOS L3 SSM.

(4) The model-based RZSM products generally perform better in the wet season than in the dry season due
to the enhanced ability to capture of the temporal dynamics of *in situ* observations in the wet season and the inertia
of remaining high soil moisture values even in the dry season. While SMOS L4 performs better in the dry season
than in the wet season, as the ground microwave radiation signal is more attenuated in the wet season due to a
substantial increase in water vapor absorption and scattering, which is propagated to SMOS L4 RZSM.

(5) Spatial-average validation could reduce the spatial noise of *in situ* soil moisture measured at different
locations and improve the representativeness of soil moisture observations to model-based grid values.

(6) The study could provide some insights into how to improve the ability of land surface models to perform
the land surface analysis by addressing the above issues. Furthermore, these results can be extended to other
regions to improve the numerical simulation capability of land surface models at global scale.


*Data availability.* The soil moisture observations in Huai River Basin is not publicly available but could be requested from the Huaihe River Commission of the Ministry of Water Resources, P. R. C. (https://hrc.gov.cn).
We provide a sample data set of these measurements for a subset of 10 stations
(https://doi.org/10.6084/m9.figshare.23497502).


*Author contributions.* EL, YHZ, JCC and HSL conceptualized the project. EL led the investigation, determined the methodology and wrote the original draft of the paper. All the co-authors contributed to the review and editing of the paper.


*Competing interests.* The authors declare that they have no conflict of interest.


*Acknowledgement.* We acknowledge the European Centre for Medium-Range Weather Forecasts (ECMWF), Goddard Earth Sciences Data and Information Services Center (GES DISC), National Center for Atmospheric Research (NCAR), China Meteorological Administration (CMA), National Snow & Ice Data Center (NSIDC) and Centre Aval de Traitement des Données (CATDS) for providing data free of charge.


*Financial support.* This research was funded by National Key Research and Development Program (grant nos. 2019YFC1510504); National Natural Science Foundation of China (grant nos. 41830752, 42071033 and 41961134003).

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
