# Peer review of "Evaluation of root-zone soil moisture products over the Huai river basin"

_EGUsphere, 2023_

## Referee Comment (RC2)

**Review EGUSPHERE-2023-1597**

**Evaluation of root-zone soil moisture products over the Huai river basin**

This paper presents a study on the evaluation of different global root zone soil moisture products and local observations for the Huai River Basin in China. The authors present detailed information on the local conditions, and the different gridded products used. They comprehensively compare the different products with each other and with the observations. Also, the authors provide a discussion on the potential reasons for the differences found. In general, it is an interesting study with a lot of analyses and clear visualization of the results. Nevertheless, I have a few comments that must be addressed before the manuscript can be published.

**General comments:**

The influence of land cover, vegetation and root representation:

The authors clearly discuss potential reasons for the mismatch between in situ RZSM observations and the global products, such as forcing data and soil texture maps. Also, shortly 'different model structures and parameterizations' (L89) are mentioned as potential cause for differences. I do think there is one more very important aspect that is missed here: the role of land cover and vegetation, vegetation roots, and soil evaporation and transpiration model representation. Vegetation is usually represented by land cover maps (that are usually prescribed similar to soil maps), which can be very different for the different models. Other relevant vegetation model properties could be Leaf Area Index (see for example Nogueira et al., 2020) or the root parameterization (e.g. Stevens et al., 2020 and Van Oorschot et al., 2021). Furthermore, transpiration of crops is very dependent on the growing season, which might be not represented by the global products. I think these issues should be specifically addressed in the introduction and discussion of the results.

Introduction

L49-67: I think this paragraph is intended to describe the state-of-the-art of global surface soil moisture, and root zone soil moisture products. The authors mention many long names of different products, which shows the detailed literature review done for this study. However, for the reader it would be more clear if the paragraphs gives a more general overview, rather than all the specific products, by answering questions such as: Why do we only have SSM direct retrievals, and not RZSM? What is available for global RZSM? How are the RZSM products generated in general?

L76-94: I think this should go before L68-75.

Scale issue:

The authors mention that the scale mismatch is a relevant aspect for the differences between RZSM observed in this study (L75 and Sect. 5.4). However, it is not clear from the methodology how the gridded products are aggregated to match the in situ measurements, and how the scale of the different products compare to the size of the HRB selected area. Moreover, how heterogeneous is the selected area in terms of precipitation, temperature, vegetation, and soil moisture observations? Since the results are mostly based on averages of the area, does the heterogeneity play a role?

Discussion:

The authors explain potential causes for the mismatches between the satellite products and the observations by using specific analyses of the precipitation, temperature and soil type. Many performance metrics have been used throughout the analyses, but I think the use of these different

metrics could be exploited more in the discussion. Different metrics represent different aspects of the timeseries, which could explain different processes. The authors could relate the causes in section 5.1 and 5.2 more specifically to the different metrics used. Here, also the vegetation/land cover aspect should be included as mentioned before. Lastly, how easily can we extrapolate these results to other regions?

Irrigation

The role of irrigation in this study is confusing, due to the following statements:

- L119: '76% is irrigated'
- L135: 'Stations are located in areas without irrigation'
- L562: 'heavily irrigated HRB in China'
- L452: 'a signature of irrigation'
- L573: 'indirectly account for irrigation'

I understand that the entire HRB is heavily irrigated, but in this study we only look at the Huaibei Plain which has only rainfed crops as indicated in Fig. 1. It remains unclear to me which area is used for the gridded products, the entire HRB or only the Huaibai Plain? This is not entire clear from the methods. If it is the Huaibai plain (which would make more sense), then irrigation is not an issue in this paper, and should not be emphasized.

**Specific comments:**

- L20,21: What are L4 and L3 here?
- L59: I think ERA5, MERRA2 and NCEP CFSv2 are not the only existing global products, it might be good to emphasize this with for example 'amongst others'.
- L75: etc. is not very scientific
- L99: it is maybe not 'difficult', but 'out of scope'
- L113: Fig. 1 instead of Figure 1
- Figure 1: There are two red lines in the figure, so the legend is not entirely clear
- L144-145: performance metrics of P and T with respect to ground observations? This is not clear.
- L200: 'Saha and coauthors, 2011' is not a valid reference
- Chapter 2 Datasets: The authors use many different units for scale, for example 0.5°x0.5° (L137); 1:5 million (L150); 30x30 arcseconds (L163). It would be helpful for the reader to include for each scale metric a rough comparison to for instance kilometre to easily compare the resolution of the different products.
- Section 2.1 'The HRB study area': for the reader the full name of HRB would be more clear
- Section 2.3 and 2.4: the authors describe a lot of different products, but it is not directly clear from these sections what is actually used for this study. Both sections could be much more concise when only referring to the relevant information for this study. For example, it is not directly relevant that ERA5 'covers the period from January 1940 to present … ocean waves' (L170). Table 1 gives a very concise overview, and could be valued more and referred to more often in the text.
- Table 1: a reference would be more informative than the 'data access' column
- Section 4.1: the first paragraph is not easily readable for the author due to all the numbers. All the numbers are also presented in Table 3, so it suffices to only mention the highly relevant numbers in the text here. Table 3 could also be combined with Fig. 2, same for Fig. 8 and Table 4.

- Figure 2 and 8: to improve visualization I would recommend to include density of points with colours (for example https://stackoverflow.com/questions/20105364/how-can-i-make-a-scatter-plot-colored-by-density )
- Figure 4:
    o The labels are not readable because of the small font
    o What are the lines? A fit through the data points? Why do the authors use a different lay-out for a scatterplot than in Fig. 2 and 8?
- Figure 8 and Table 4: I think it is more convenient to use degree Celsius than Kelvin for temperatures.
- L381: 'good agreement' is a subjective statement, I think it is questionable if a $R>0.4$ is 'good'.
- Figure 9: This figure implies that also the models differentiate soil moisture for layer 0-30cm and 30-100cm, while for most models this is not the case?
- Abstract: the statements about the explanations of the differences are quite strong, because we know there is many other factors that play a role.

---

## Referee Comment (RC3)

Review of the manuscript egusphere-2023-1597 Evaluation of root-zone soil moisture products over the Huai river basin by Liu et al.

**SUMMARY**

The manuscript presents an intercomparison between eight root zone soil moisture (RZSM) products and in situ measurements. The differences are discussed in the light of the uncertainty in precipitation, air temperature and soil properties.

Overall, I think the study could be a valuable scientific contribution. However, at the current status, I think the manuscript should be improved in several parts before reaching good quality for a possible publication. Specifically, I think the Authors should put major effort into improving the descriptions of the products, the methods should be extended, the discussion should be integrated accordingly. Below I provide additional details about my major concerns followed by specific comments.

**GENERAL COMMENTS**

**Irrigation**

A major confusion in my opinion in the intercomparison is the role of irrigation. At L119 and L124 it stated that the study is conducted over a highly irrigated area. At L135 it is stated that soil moisture sensors are in areas without irrigation. It is not clear by reading if the RZSM products account for irrigation or not and if this can be a concern. Only later in the discussion (L452), it is stated that the overestimation of RZSM by ERA5 (Fig. 3) could be a signature of irrigation because the in situ RZSM observations do not capture irrigation. Does this mean that some RZSM products are based on model that take into account irrigation and others not? On the one hand, this information should better explained and discussed. On the other hand, I wonder what is the scientific meaning of comparing soil moisture in rainfed area to model that are accounting for irrigation.

**Soil map and soil parameters (section 4.3.2 and section 5.2).**

The assessment of the soil properties is valuable. It should be noted, however, that high discrepancies come with the use of different pedotransfer functions (PTF) to derive soil hydraulic parameters (retention curve and hydraulic conductivity). I guess these parameters are used in each RZSM product but estimated with different PTFs. This could also explain part of the uncertainty. This information is missing in the manuscript but should be integrated.

**Spatial aggregation and comparison at each site should be clarified and improved**

As far as I have understood (L267), the comparison is conducted between the spatial average of the 58 in situ observations. It is not well reported how much is the spatial extent of the 58 stations but looking at figure 1 I guess the station covers an area of around 300 x 200 km2. I then deduce that this spatial average is compared to the spatial average of the gridded products (i.e., each product has different resolutions, but more than one cell of the gridded products covers the area of the 58 stations). So first of all it should be better explain how many cells have also been aggregated for each product. Only later in the results section (L317) I discovered that a comparison has also been performed without aggregating spatially. So, first of all, this information should be provided also before in the methods. Moreover, I would also considering moving some plots that are now in supplement to the main manuscript to strengthen the

analysis. Anyway, I'm confused by the fact that the comparison is performed at each station. Does this mean that you have always one station against one cell of the gridded products? Please clarify.

**Section 2.4: description of the eight RZSM products**

The description of the eight products should be improved in several parts. The information provided for each product is not always consistent. Some products are better described and with more details than others. E.g., for MERRA-2, the description focuses on the precipitation. Instead, ERA5 does not have any information about. NCEP CFSv2 description is very short. Who provided that? What are the main properties? Some characteristics provided should also be put more in relation to the focus of the paper. E.g., for ERA5, the data assimilation system is described in detail. Is that relevant for the purpose of the paper. If yes, it should be clarified. Overall, the main differences between the products relevant for the present study (e.g., soil map, precipitation, land use, irrigation etc.) should be better highlighted. Table 1 should be extended accordingly. Please note that some relevant information are discussed only later but in my opinion they should be moved to the method section. This would help understanding and strengthening the discussion of the results. E.g., L410 The soil properties data used in the eight RZSM products were all derived from the FAO/UNESCO soil map of World except for CLDAS, which used the soil data developed by Shangguan et al. (2013), and SMAP L4, which used the HWSD soil properties over China. L426. Global precipitation and air temperature forcing data are used in the production of all RZSM products except for SMOS L4. L452. The overestimation of RZSM by ERA5 (Fig. 3) could be a signature of irrigation because the in situ RZSM observations do not capture irrigation.

**Figure quality**

Figures are not always readable and meaningful. I suggest putting some more effort into evaluating how to present the results. E.g.,

Fig 1. could also shows the pixel size of the products. This would help understanding spatial extend and intercomparisons.

Fig. 4 is not readable at all. Plots and texts are too small.

Fig. 5 can be improved by having only 8 histograms of the RZSM products and overlapping each histogram with the observation's histogram. This could help to visualize the differences.

Fig.6. The plots are hardly comparable. It could be evaluated to present one plot with all the cumulative precipitation, or histogram of the precipitation etc.

Fig.7. It is not clear to me what is actually presented. This is in line to the general comment above about aggregation. Are you comparing spatial averaged precipitation? What does standard deviation refer to? If you compare each rain gauge to pixel wise, how have you aggregated?

**Take-home-message**

I'm expecting to read the overall take-home-message. After performing this intercomparison, what can you conclude? Could we trust this products? Where and which conditions? How would you suggest further improving, studies etc.? This is missing throughout the manuscript but should be understandable from the abstract and more extended at the conclusions.

**SPECIFIC COMMENTS IN ORDER OF APPEARANCE (L = LINE NUMBER)**

L13. I would not use the term "direct validation" but "assessment".

L29. Th abstract focused on describing the actual results. This is fine but I would also expect at the end to read the take-home-message. E.g., what do we learn by this study? Can we trust, use, apply RZSM products? Where? In which conditions? What in our view and based on this study would further suggest to improve the performances?

L108-109. Are these two lines really needed here? I would integrate this information later when you speak about comparison. E.g., L133 for the definition of RZSM.

L128. Table S1 shows the results of the assessment, and it does not provide additional information about the in situ stations. I would remove this cross reference here and rather add Figure 1 where the locations of the in situ stations are shown.

L263. I would extend a bit on the meaning and interpretation for PD, FAR and CSI.

L280. I was expecting to read more about the description of the equation after L280. Any text missing?

L293. I think here is a good place to cite table 3 as well.

L295. It is stated that SMOS-L4 underestimates and the other overestimated the observations. By looking at figure 2, I see the opposite. Please double check what you are plotting.

L306. Figure 3 shows spatial average of in situ soil moisture and its spatial variability. As far as I have understood (see general comment above on the spatial aggregation), the spatial average of the RZSM products is shown but, if possible, it could be interesting to show here also the spatial variability of the RZSM products.

L317. Description of the comparison at each site should be reported as well in the method (section 3.1) See also general comment above on the spatial aggregation.

L319. The method to calculate the anomalies is reported only in the supplement. I think should be moved to the method (section 3.1)

L585. After summarizing the mani conclusions, I suggest summarizing the outlook of the study. See also general comments above.

Supplement. I think the supplement should have a title with the name of authors as well.

---

## Author Comment (AC2)

**Response to Reviewer #1's comments on the manuscript egusphere-2023-1597**

**RC1: 'Comment on egusphere-2023-1597', Anonymous Referee #1, 07 Nov 2023**

The authors attempted to evaluate the predictive performance of eight satellite data-derived root-zone soil moisture data. The authors provided a quantitative statistical analysis of each RZSM product based on average values of in-situ, and remote sensing estimates for the Huai River Basin. The authors concluded that the GLDAS CLSM RZMS products outperform other RZSM products. While the information contained in the manuscript is quantitative, I ended up questioning the potential contribution of this paper to the readers. I have tried to address why I think

The authors thank the Reviewer #1 for her/his constructive and insightful comments that help us improve the quality of the manuscript. The original comments from Reviewer #1 are in black font, and our responses are in blue font.

 - Major comments:

1. Although the study provides a significant amount of comparative analysis between remote sensing-derived RZSM products as well as against observational data, the mechanistic understanding and explanation of each result are widely missing across the manuscript. Therefore, the author's rationale about the causes of differences was often too obvious or uncertain.

Response: Thank you for your valuable comment. In this study, we presented an intercomparison between eight root zone soil moisture (RZSM) products and in situ observations. Although the different RZSM products are evaluated quantitatively, it should be noted that the focus of this study is to investigate the sources of error of the RZSM products. The SMOS Level 4 (L4) RZSM is produced by combining a modified exponential filter and SMOS Level 3 surface soil moisture (SSM). The other seven RZSM products (except SMOS L4) are produced by land surface models driven by surface atmospheric forcing data from atmospheric general circulation models; the ability of the land surface model to simulate states (e.g., RZSM) and fluxes is limited by uncertainties in meteorological forcing and parameters, as well as inadequate model physics. It is difficult to quantify the model physics because different land surface models are used to produce different RZSM products. Therefore, we analyzed the atmospheric forcing data (especially for precipitation, which dominates the terrestrial water cycle), which is considered as the most important factor in determining the accuracy of the modeled RZSM, and the model static parameters (soil properties), which strongly influence the movement of water in the vadose zone, and local underlying surface conditions. The objective of this study is to provide some insights on how to improve the ability of land surface models to simulate land surface states and fluxes by finding the common characteristics of bias existing in different datasets used in land surface models.

2. Also, the authors argued that GLDAS CLSM-derived RZSM outperforms other RZSM estimates. However, rather than trying to explain the different predictive performances of each RZSM product (against in-situ observations) in relation to soil properties, land cover/use, and vegetation in the study catchment, the authors just used the average of 58 in-situ data as well as satellite products, resulting in 'all-lumped' single time series for each dataset. Thus, it is not convincible to say a certain remote sensing-derived estimates outperform others since the performance differences can be revealed differently depending on soil properties, vegetation, land cover/use, etc.

Response: Most of the in situ stations are located in the Huaibei Plain, which is a major grain production area. According to the land cover map of the Huai River basin (Figure 1c), most of them (56 of 58) is located in the cropland regions, we will add it in the supplement (see Table S1 below). In terms of soil properties, the lime concretion black soil is the main soil type in the Huaibei plain, which is shown in line 482. Therefore, we believe that soil properties, vegetation, and land cover/use are homogeneous across different *in situ* stations. In addition, two validation strategies were used in the study. The first is to compare the mean RZSM averaged over all in situ stations with the mean RZSM averaged over all grids. The second one is the point-grid validation, the station measurements are compared directly with the grid value where the station is located, if more than one station, the measurements of these stations are averaged. The point-grid validation has been provided in the supplement and draws the same conclusion as the station-averaged validation that GLDAS_CLSM outperforms other RZSM products. The section "3.3 Validation strategies" will be added in chapter 3 Methods.

Table S1 Overview of in situ stations in Huai River Basin

| Station Name | Longitude (E) | Latitude (N) | Elevation (m) | Land cover |
|---|---|---|---|---|
| Taolaoba | 117.16 | 32.18 | 48 | Irrigated Crop |
| Chahua | 116.02 | 33.03 | 39 | Rainfed Crop |
| Hanting | 116.32 | 33.02 | 28 | Rainfed Crop |
| Songji | 115.27 | 32.82 | 39 | Rainfed Crop |
| Funan | 115.57 | 32.64 | 33 | Rainfed Crop |
| Santa | 115.70 | 32.81 | 33 | Rainfed Crop |
| Yaoli | 116.17 | 31.82 | 58 | Irrigated Crop |
| Guanting | 116.85 | 31.80 | 51 | Irrigated Crop |
| Zhuangmu | 117.11 | 32.36 | 27 | Irrigated Crop |
| Guiji | 116.62 | 32.78 | 23 | Irrigated Crop |
| Xiaji | 116.54 | 32.65 | 25 | Rainfed Crop |
| Shuangfu | 115.57 | 33.34 | 37 | Rainfed Crop |
| Fentai | 115.73 | 33.45 | 35 | Rainfed Crop |
| Santang | 115.83 | 33.31 | 32 | Rainfed Crop |
| Lixin | 116.21 | 33.14 | 28 | Rainfed Crop |

| | | | | |
|---|---|---|---|---|
| Jieshou | 115.36 | 33.27 | 42 | Rainfed Crop |
| Yangqiao | 115.39 | 33.02 | 28 | Rainfed Crop |
| Guangwu | 115.33 | 33.37 | 42 | Rainfed Crop |
| Huangling | 115.13 | 33.04 | 37 | Rainfed Crop |
| Quanyang | 115.44 | 33.11 | 35 | Rainfed Crop |
| Kanheliu | 115.85 | 33.10 | 33 | Rainfed Crop |
| Kouziji | 116.09 | 32.84 | 26 | Rainfed Crop |
| Sanshilipu | 116.11 | 32.70 | 27 | Rainfed Crop |
| Xiaqiao | 116.38 | 32.64 | 26 | Rainfed Crop |
| Hengpaitou | 116.36 | 31.59 | 72 | Woodland |
| Xianghongdianxia | 116.18 | 31.58 | 116 | Woodland |
| Wangchenggang | 116.53 | 31.74 | 76 | Irrigated Crop |
| Lumiao | 115.80 | 34.00 | 39 | Rainfed Crop |
| Dasi | 115.87 | 33.80 | 42 | Rainfed Crop |
| Youhe | 115.79 | 33.63 | 38 | Rainfed Crop |
| Huagou | 116.06 | 33.51 | 33 | Rainfed Crop |
| Dahu | 116.35 | 33.52 | 31 | Rainfed Crop |
| Chenqiao | 116.56 | 33.09 | 25 | Rainfed Crop |
| Heliu | 116.97 | 33.03 | 25 | Rainfed Crop |
| Linhuanzha | 116.57 | 33.67 | 29 | Rainfed Crop |
| Guzhenzha | 117.33 | 33.30 | 18 | Rainfed Crop |
| Wudaogou | 117.34 | 33.16 | 21 | Rainfed Crop |
| Hexiangzha | 117.18 | 33.00 | 18 | Rainfed Crop |
| Tancheng | 116.56 | 33.44 | 29 | Rainfed Crop |
| Xibakou | 117.87 | 33.15 | 11 | Rainfed Crop |
| Xulouzha | 116.75 | 33.92 | 30 | Rainfed Crop |
| Suxianzha | 117.08 | 33.67 | 28 | Rainfed Crop |
| Gukouzha | 116.45 | 34.27 | 39 | Rainfed Crop |
| Kuaitanggou | 117.55 | 33.75 | 20 | Rainfed Crop |
| Yanglou | 116.78 | 34.32 | 39 | Rainfed Crop |
| Langanji | 117.23 | 33.93 | 25 | Rainfed Crop |
| Dulou | 116.85 | 34.20 | 37 | Rainfed Crop |
| Xiangyang | 117.58 | 33.47 | 24 | Rainfed Crop |
| Shuangdui | 116.90 | 33.42 | 25 | Rainfed Crop |
| Shuoli | 116.90 | 34.03 | 32 | Rainfed Crop |
| Huangmiao | 117.65 | 33.08 | 19 | Rainfed Crop |
| Baoji | 117.11 | 33.16 | 22 | Rainfed Crop |
| Dinghouying | 117.34 | 33.46 | 24 | Rainfed Crop |
| Xuanmiao | 116.27 | 34.52 | 54 | Rainfed Crop |
| Longhai | 116.35 | 34.40 | 45 | Rainfed Crop |
| Zhangzhuangzhai | 116.60 | 34.12 | 37 | Rainfed Crop |
| Sixian | 117.92 | 33.43 | 16 | Rainfed Crop |
| Dazhuang | 117.87 | 33.67 | 20 | Rainfed Crop |

3. It is also not indicated how each satellite-based soil moisture (at multiple depths) and RZSM 'with different spatiotemporal resolutions were aggregated (again, spatially and temporally) to come up with the sets of time series that require consistent temporal scales between them. The method used for spatial aggregation of the gridded-RZSM also needs to be manifested (i.e., methods).

Response: The following text (section 3.3 Validation strategies) will be added in chapter 3 Methods.

"In terms of the temporal resolution, except for the RZSM products (e.g., GLDAS_CLSM, SMOS L4) provided on daily time steps, the other sub-daily RZSM datasets (hourly/3-hourly/6-hourly time steps, shown in Table 1) are aggregated to daily average values. Therefore, the aggregated RZSM products could match the observations at daily time intervals. In terms of spatial resolution, we didn't change the spatial resolution of any RZSM products and used the original grid resolution. Two validation strategies were used in the study. The first is to compare the RZSM time series averaged over all in situ stations with the RZSM time series averaged over all model grids where the stations are located (Fig.2 and 3 shown in this study). The second one is the single point-grid validation, the measurements at each station are compared directly with the grid values where the station is located. If there is more than one station within a grid, the measurements of each station that located in the grid are compared to the grid values separately. The point-grid validation has been provided in the supplement (Fig. S2 and S3)."

4. As this is site-specific, it sounds even less convincing that CLSM-derived soil moisture products outperform, and thus it gets more confusing what the authors want to argue from the RZSM products comparison.

Response: On the one hand, we want to evaluate the performance of eight RZSM products in the agricultural crop area, which could provide a more accurate RZSM dataset for agricultural drought monitoring. The results show that GLDAS_CLSM outperforms the other RZSM products. However, it doesn't mean CLSM-derived soil moisture outperforms, because SMAP L4 and MERRA-2 also use CLSM. More importantly, the focus of this study is to investigate the sources of error of the different RZSM products, which could provide some insights about how to improve the ability of land surface models to simulate the land surface states and fluxes.

5. There is significant inconsistency (due to the randomness in estimating RZSM from the remote sensing data) between RZSM estimation methods. For example, the authors tried to estimate RZSM using a depth-weighted method, but equation 1 used for in-situ RZSM is different from equation 2, which was used for RZSM estimation from satellite-derived modeled soil moisture.

Response: The in situ soil moisture measurements are available at four depths (10, 20, 40 and 100 cm). However, in addition to the GLDAS_CLSM, MERRA-2, SMAP L4 and SMOS L4, which directly provide the 0-100 cm RZSM, the other model-based

soil moisture datasets are provided in different soil layers, i.e., NCEP CFSv2, CLDAS and GLDAS_NOAH ($\theta_{0-10\ cm}$, $\theta_{10-40\ cm}$, $\theta_{40-100\ cm}$), ERA5 ($\theta_{0-7\ cm}$, $\theta_{7-28\ cm}$, $\theta_{28-1\quad cm}$). The in situ measurements are for each soil depth, but the model-based RZSM products are for each soil layer. They are not consistent. Therefore, the study uses two different equations to calculate the RZSM. The in situ RZSM is calculated using a depth-weighted mean of the measurements at four soil depths (10, 20, 40 and 100 cm). This method (equation 1) has been used in the study by Gao et al., (2017) and Xing et al., (2021). The model-based RZSM is calculated with a weighted average of the 0-100 cm RZSM. This method (equation 2) has been used in the study by González-Zamora et al., (2016) and Xing et al., (2021) and calculation of SMOS L4 RZSM (Al bitar et al., 2021).

-Specific comments:

line 39-40: is this sentence needed?

Response: We will delete this sentence in the revised manuscript.

line 42: duplicate definition of RZSM?

Response: We will delete this sentence in the revised manuscript.

line 99-100: by this sentence, do you intend not to include any process-based explanation for the soil moisture products? What about attempting to explain the performance differences found among the RZSM products (as this is essentially modeled data) in relation to model structure? Why does CLSM outperform other land models in terms of RZSM products?

Response: This study attempts to investigate the error sources of RZSM products without considering the model structure. While only evaluating the atmospheric forcing, soil texture, and local conditions, we analyze the effects of these error sources on RZSM estimation from the perspective of physical processes. For example, overestimated precipitation tends to lead to overestimated water-related states (soil moisture) or fluxes (runoff). The clay exhibits stronger water retention capacity compared to sand at the same matric potential, and high soil organic carbon leads to high soil porosity. Therefore, the overestimated clay fraction and soil organic carbon lead to higher water stored in the soil. In addition, different land surface models are used to produce different RZSM products. For example, ERA5 (HTESSEL), MERRA-2 (CLSM), NCEP CFSv2 (Noah), GLDAS_NOAH (Noah), GLDAS_CLSM (CLSM), CLDAS (CLM, CoLM, Noah-MP), SMAP L4 (CLSM), SMOS L4 (exponential filter, not land surface model). Even the same CLSM land surface model is used for both MERRA-2 and SMAP L4, but the model version is a bit different. Therefore, it is difficult to directly quantify the effect of model structure on RZSM. In this study, the GLDAS_CLSM RZSM product outperforms other model-based RZSM products, but this doesn't mean that CLSM outperforms other land surface models. The accuracy of RZSM depends on the meteorological forcing, the structure of the land surface model, and the parameterization scheme.

Chapter 2.4: the information on the spatial and temporal resolution of each data needs to be revisited and clearly indicated.

Response: The information on the spatial and temporal resolution of eight RZSM products is shown in Table 1.

Chapter 3.2: why did you estimate satellite-derived RZSM different from in-situ RZSM? Why equation 1 and 2 are different? How convincing are the RZSM comparisons based on equation 1 and 2?

Response: The in situ soil moisture measurements are available at four depths (10, 20, 40 and 100 cm). However, in addition to the GLDAS_CLSM, MERRA-2, SMAP L4 and SMOS L4, which directly provide the 0-100 cm RZSM, the other model-based soil moisture datasets are provided in different soil layers, i.e., NCEP CFSv2, CLDAS and GLDAS_NOAH ($\theta_{0-10\,cm}$, $\theta_{10-40\,cm}$, $\theta_{40-10\,\,cm}$), ERA5 ($\theta_{0-7\,cm}$, $\theta_{7-28\,cm}$, $\theta_{28-100\,cm}$). The in situ measurements are for each soil depth, but the model-based RZSM products are for each soil layer. They are not consistent. Therefore, the study uses two different equations to calculate the RZSM. The in situ RZSM is calculated using a depth-weighted mean of the measurements at four soil depths (10, 20, 40 and 100 cm). This method (equation 1) has been used in the study by Gao et al., (2017) and Xing et al., (2021). The model-based RZSM is calculated with a weighted average of the 0-100 cm RZSM. This method (equation 2) has been used in the study by González-Zamora et al., (2016) and Xing et al., (2021) and calculation of SMOS L4 RZSM (Al bitar et al., 2021).

line 293: instead of averaging all in-situ stations, can you think of disaggregating the study basin (and stations) using any available information such as surface soil properties, orography (e.g., slope, and elevation), land cover, and/or vegetation? That will help the readers get more generalizable information and references.

Response: It is a very good and useful suggestion. However, this study pays more attention to soil moisture measured at 58 stations rather than a specific station. The underlying surface conditions (e.g. surface soil properties, orography, land cover and vegetation) is considered as homogeneous. 56 of 58 in situ stations are located in crop lands of the Huaibei Plain, and the elevation is quite similar (Table S1). The lime concretion black soil is the main soil types in the Huaibei plain. In future study, we will attempt to investigate the effect of different underlying surface conditions (vegetation types, etc.) on soil moisture estimations for specific station.

line 306-309: This needs to be rephrased. It is hard to understand what is meant.

Response: The text (line 306-309) will be rephrased from "Figure 3 shows time series of in situ RZSM observations averaged over all in situ stations with its spatial variability, and of 3 RZSM products, ERA5, SMOS L4, and GLDAS_CLSM, presenting a marked overestimation, a marked underestimation, and the best overall agreement with in situ observations, respectively. Other products can be seen in Fig. S1."

To "Figure 3 shows the time series of observed and model-based RZSM averaged over all in situ stations and the grids where the *in situ* stations are located. ERA5, SMOS L4, and GLDAS_CLSM show overestimation, underestimation, and the best overall agreement with in situ observations, respectively. Other products are shown in Fig. S1".

line 311: can you explain why SMOS L4 showed less rapid changes and smoother trends?

Response: It is well known that the SSM shows a faster response to atmospheric variations than RZSM, especially for precipitation. Therefore, RZSM shows less rapid changes and smoother trends than SSM, which shows a strong variability. On the one hand, SMOS L4 RZSM is estimated from SMOS L3 SSM together with a modified exponential filter with different parameter T (characteristic time length) proposed by Wagner et al., (1999). The exponential filter can smooth the trend of SSM, the higher the T value, the smoother the RZSM trend. On the other hand, precipitation with high spatial and temporal variability is the main forcing input of other model-based RZSM, which show a strong response to precipitation. Precipitation is not used in producing SMOS L4 RZSM. Therefore, SMOS L4 shows less rapid changes and smoother trends.

line 321: can you explain why they did a better job in the wet season compared to the dry season?

Response: In the Huai river basin, more than 60 % of the annual precipitation falls between June and September (wet season), which significantly increases the RZSM. According to Figure 1 and S1, it is obvious that RZSM shows a strong response to precipitation events. In general, the model-based RZSM datasets increase with the increasing in situ observations after a precipitation event. The model-based RZSM datasets show strong variability and a good agreement with observations in wet season. However, the in situ RZSM shows stronger variability than the model-based RZSM datasets in dry season, the model-based RZSM datasets show little variability and don't capture the temporal trend of in situ observations. It indicates that the land surface models are sensitive to precipitation events than no precipitation events and show better skill in simulating RZSM when there is a precipitation event. This could explain the better performance of model-based RZSM datasets in the wet season than that in the dry season.

line 360: can you explain why individual satellite-based RZSM products showed different probabilistic distributions? Some are log-normal and the others are normal. Can you add more explanation on this matter?

Response: The peak of the relative frequency for model-based RZSM products ranges from 0.3 to 0.6. RZSM products with log-normal distribution show that low values dominate the RZSM time series, which could be caused by low precipitation. The precipitation field derived from the atmospheric general circulation model (AGCM) generally has too many drizzle events ($<1$ mm day$^{-1}$). The modeled RZSM is affected by meteorological forcing, model structure and parameterization, etc. The RZSM estimates are subject to random error and systematic bias, and it is difficult to directly quantify

which factor affects the probability distributions of different RZSM products. In addition, the probability distribution of RZSM may depend on the research periods, the probability distribution of RZSM in wet season may be different from that in dry season.

line 375: how does this ground-based observation of precipitation (840 mm/year) represent the average precipitation of the basin area? You also compared gridded-precipitation with this in-situ precipitation observation (line 430). Can you clarify how solid the comparison of this in-situ precipitation with gridded precipitation is?

Response: In this study, the ground-based precipitation observation doesn't represent the average precipitation of the watershed area. We only compare the ground-based precipitation observation with the modeled precipitation of the grid where the station is located from the grid perspective.

line 432: do you think MERRA-2 and GLDAS-CLSM would outperform other satellite-derived RZSM in other basins (or area) as well? What if you perform a continental-scale study, will you still think there will be a certain winner? If not, how can you limit the scale of this sort of comparison study to be meaningful and convincing?

Response: In this study, MERRA-2 and GLDAS-CLSM outperform other model-based RZSM products in the Huaibei Plain, where cropland dominates. It is uncertain whether MERRA-2 and GLDAS-CLSM would still outperform other products if this study were conducted in other basins (areas) or on a continental scale. Because the precipitation data derived from the atmospheric general circulation model perform differently in different regions. For example, these large-scale atmospheric processes over the extra-tropics are better resolved in the AGCM than convective processes over the tropics. It is a study for specific underlying surface conditions (agricultural crop region). On the one hand, the evaluation of eight RZSM products in the agricultural crop region could provide a more accurate RZSM dataset for agricultural drought monitoring. More importantly, the focus of this study is to investigate the sources of error of the different RZSM products, which could provide some insights to improve the ability of land surface models to simulate the land surface states and fluxes.

line 436-438: the sentences need to be re-structured to clarify the argument.

Response: The text (line 436-438) will be rephrased from "The MERRA-2 model background precipitation corrected with NOAA CPCU gauge-based precipitation observations was implemented in the coupled land-atmosphere reanalysis system, which may also contribute to the high consistency with the ground-based precipitation"

To "Before driving the land surface water budget, the MERRA-2 model background precipitation was corrected with NOAA CPCU gauge-based precipitation in the coupled land-atmosphere reanalysis system, resulting in more accurate precipitation fields for MERRA-2".

line 453: in-situ RZSM observation does not capture irrigation effect? Can you explain how the irrigation water supply does not impact the soil moisture content?

Response: We didn't express it clearly. The original meaning of this sentence is that the in situ station does not capture the irrigation signal. Because the in situ stations are usually installed away from the cropland to avoid the effect of anthropogenic irrigation on the original soil water content supplied by precipitation. In addition, reviewer2 and reviewer3 also raise question about question. The overall comments from three reviewers indicate that the irrigation is not an issue in this paper, and should not be emphasized. We will delete relevant statements about irrigation.

line 485-489: can you add more information on how the soil properties could end up in certain ranges of soil moisture values?

Response: We have illustrated the effect of soil properties on ranges of soil moisture values (line 457-462). "In general, soil texture is closely related to the ability of the soil to retain water, as water molecules adhere more tightly to fine-textured clay particles than coarse-textured sand particles. Consequently, clay exhibits stronger water retention capacity and higher water content stored in the soil compared to sand at the same matric potential. In addition, the overestimated FAO/UNESCO soil organic carbon content (Fig. 9) leads to higher soil porosity and lower bulk density. As a result, water can infiltrate more quickly and more water can flow through the soil and can be retained in the soil".

Reference:

Gao, X., Zhao, X., Brocca, L., Huo, G., Lv, T., Wu, P., 2017. Depth scaling of soil moisture content from surface to profile: multistation testing of observation operators. Hydrol. Earth Syst. Sci. 1–25, https://doi.org/10.5194/hess-2017-292.

Xing, Z., Fan, L., Zhao, L., et al. A first assessment of satellite and reanalysis estimates of surface and root-zone soil moisture over the permafrost region of Qinghai-Tibet Plateau, Remote Sens. Environ., 265, 112666, https://doi.org/10.1016/j.rse.2021.112666, 2021.

González-Zamora, Á., Sánchez, N., Martínez-Fernández, J., Wagner, W., 2016. Root-zone plant available water estimation using the SMOS-derived soil water index. Adv. Water Resour. 96, 339–353. https://doi.org/10.1016/j.advwatres.2016.08.001.

Al Bitar, A., Mahmoodi, A., Kerr, Y., Rodriguez-Fernandez, N., Parrens, M. and Tarot, S.: Global Assessment of Droughts in the Last Decade from SMOS Root Zone Soil Moisture, 2021 IEEE International Geoscience and Remote Sensing Symposium (IGARSS), 8628-8631, https://doi.org/10.1109/igarss47720.2021.9554773, 2021.

Wagner, W.; Lemoine, G.; Rott, H. A Method for Estimating Soil Moisture from ERS Scatterometer and Soil Data. Remote Sens. Environ. 1999, 70, 191–207, https://doi.org/10.1016/s0034-4257(99)00036-x.

---

## Author Comment (AC3)

**Response to Reviewer #2's comments on the manuscript egusphere-2023-1597**

**RC2: 'Comment on egusphere-2023-1597', Anonymous Referee #2, 10 Nov 2023**

This paper presents a study on the evaluation of different global root zone soil moisture products and local observations for the Huai River Basin in China. The authors present detailed information on the local conditions, and the different gridded products used. They comprehensively compare the different products with each other and with the observations. Also, the authors provide a discussion on the potential reasons for the differences found. In general, it is an interesting study with a lot of analyses and clear visualization of the results. Nevertheless, I have a few comments that must be addressed before the manuscript can be published.

The authors thank the Reviewer #2 for her/his constructive and insightful comments that help us improve the quality of the manuscript. The original comments from Reviewer #2 are in black font, and our responses are in blue font.

**General comments:**
The influence of land cover, vegetation and root representation:
The authors clearly discuss potential reasons for the mismatch between in situ RZSM observations and the global products, such as forcing data and soil texture maps. Also, shortly 'different model structures and parameterizations' (L89) are mentioned as potential cause for differences. I do think there is one more very important aspect that is missed here: the role of land cover and vegetation, vegetation roots, and soil evaporation and transpiration model representation. Vegetation is usually represented by land cover maps (that are usually prescribed similar to soil maps), which can be very different for the different models. Other relevant vegetation model properties could be Leaf Area Index (see for example Nogueira et al., 2020) or the root parameterization (e.g. Stevens et al., 2020 and Van Oorschot et al., 2021). Furthermore, transpiration of crops is very dependent on the growing season, which might be not represented by the global products. I think these issues should be specifically addressed in the introduction and discussion of the results.

Response: Thank you for rigorous consideration. We do agree with your idea. We will mention these issues in the introduction and discussion. Because the inconsistent vegetation type and parameterization schemes used in LSMs, it is difficult to compare them quantitatively.

The following text will be added in the introduction.

The text (Line 88-89) will be rephrased from "Finally, the accuracy of soil moisture simulations is also affected by different model structures and parameterisations"

To "Finally, the accuracy of soil moisture simulations is also affected by inadequate model structures and inaccurate parameterization schemes. Especially for vegetation parameterizations (e.g., canopy and root tissues), which show large uncertainties in simulating the water and heat fluxes in different LSMs (Nogueira et al., 2020; Stevens et al., 2020; van Oorschot et al., 2021). For example, van Oorschot et al. (2021) proposed a climate-controlled root zone storage capacity by calculating a time-varying total soil depth based on a moisture depth model instead of using a constant of 2.84 m in the original HTESSEL land model and

improving water flux simulations."

The following text will be added in the discussion.

"Vegetation also plays a crucial role in the water and carbon exchange between atmosphere and land surface through transpiration and photosynthesis, which has significant effect on the simulation of soil moisture by LSMs, especially for RZSM. On the one hand, different land cover maps are employed in LSMs to participate in the terrestrial water and carbon cycles. For example, GLDAS_NOAH uses modified IGBP MODIS (Moderate Resolution Imaging Spectroradiometer) 20-category vegetation classification, and GLDAS_CLSM uses the University of Maryland (UMD) land cover classification based on AVHRR (Advanced Very High Resolution Radiometer) land cover map (Rui et al., 2021), MERRA-2 uses the global land cover characteristics database, version 2.0 (Reichle et al., 2017). On the other hand, the parameterization schemes for vegetation canopy (e.g., Leaf Area Index high-and low-vegetation fraction, type and density, Nogueira et al. (2020)) and root tissues (root distribution, rooting depth, root density and root zone water storage, Stevens et al. (2020) and van Oorschot et al. (2021)) vary considerably across different LSMs. Therefore, it is difficult to depict consistently and accurately the dynamic evolution of vegetation for different LSMs. Furthermore, transpiration of crops is very dependent on the growing season, which might be not well represented in the LSMs."

Introduction

L49-67: I think this paragraph is intended to describe the state-of-the-art of global surface soil moisture, and root zone soil moisture products. The authors mention many long names of different products, which shows the detailed literature review done for this study. However, for the reader it would be more clear if the paragraphs gives a more general overview, rather than all the specific products, by answering questions such as: Why do we only have SSM direct retrievals, and not RZSM? What is available for global RZSM? How are the RZSM products generated in general?

Response: The text L49-67 will be rephrased from "Recent satellite soil moisture…exponential filter model (Albergel et al., 2008; Al Bitar and Mahmoodi, 2020)."

To "Recently, microwave-based satellite missions provide global surface soil moisture (SSM) retrievals with approximately 3-day temporal resolution, for example, SMAP and SMOS SSM is retrieved from the brightness temperature of the passive microwave radiometer. ASCAT SSM is retrieved from the backscatter coefficient of the active microwave scatterometer. However, the soil moisture is limited to the top few centimeters (0-5 cm for L-band) due to the limitations of microwave penetration depth (Kerr et al., 2001; Reichle et al., 2017b). Therefore, various approaches have been developed to estimate the RZSM and are roughly divided into three categories (Liu et al., 2023). Including, (1) statistics-based methods, such as linear regression (Zhang et al., 2017) and cumulative distribution function (Gao et al., 2019), (2) data-driven machine learning methods, such as rand forest (Carranza et al., 2021) and artificial neural network (Kornelsen et al., 2014), (3) physically based methods, such as data assimilation of satellite-derived observations into LSMs (Albergel et al., 2017; Bonan et al., 2020). Among them, the assimilation of satellite-derived observations into LSMs is considered as the most accurate method to estimate RZM due to the explicit physical mechanism, while requiring large

amounts of input data (air temperature, surface pressure, wind speed, solid and liquid precipitation, incoming shortwave and longwave radiation). To date, several RZSM products have been developed for broader global applications, such as the Global Land Data Assimilation System (GLDAS_NOAH and GLDAS_CLSM) (Rodell et al., 2004), the China Land Data Assimilation System (CLDAS) (Shi et al., 2014) and the Soil Moisture Active Passive (SMAP) Level 4 (L4) (Reichle et al., 2012; Reichle et al., 2017a), the European Centre for Medium-Range Weather Forecasts (ECMWF) fifth generation reanalysis (ERA5) (Hersbach et al., 2020), the Modern-Era Retrospective Analysis for Research and Applications version 2 (MERRA-2) (Gelaro et al., 2017), and the National Centers for Environmental Prediction Climate Forecast System version 2 (NCEP CFSv2) (Saha et al., 2014). These RZSM products are generated by combining LSMs driven by meteorological forcing fields from atmospheric general circulation model (AGCM) and satellite-derived data using different data assimilation techniques (Calvet and Noilhan, 2000; Rodell et al., 2004) and provide optimal land surface states and fluxes. In addition, the Soil Moisture and Ocean Salinity (SMOS) Centre Aval de Traitement des Données (CATDS) provides SMOS L4 RZSM products, which are derived from SMOS Level 3 (L3) 3-day SSM retrievals using a statistical exponential filter model (Albergel et al., 2008; Al Bitar and Mahmoodi, 2020).

L76-94: I think this should go before L68-75.

Response: We will move Line 76-94 before Line 68-95 in the revised manuscript.

Scale issue:
The authors mention that the scale mismatch is a relevant aspect for the differences between RZSM observed in this study (L75 and Sect. 5.4). However, it is not clear from the methodology how the gridded products are aggregated to match the in situ measurements, and how the scale of the different products compare to the size of the HRB selected area. Moreover, how heterogeneous is the selected area in terms of precipitation, temperature, vegetation, and soil moisture observations? Since the results are mostly based on averages of the area, does the heterogeneity play a role?

Response: The following text (section 3.3 Validation strategies) will be added in chapter 3 Methods.

"In terms of the temporal resolution, except for the RZSM products (e.g., GLDAS_CLSM, SMOS L4) provided on daily time steps, the other sub-daily RZSM datasets (hourly/3-hourly/6-hourly time steps, shown in Table 1) are aggregated to daily average values. Therefore, the aggregated RZSM products could match the observations at daily time intervals. In terms of spatial resolution, we didn't change the spatial resolution of any RZSM products and used the original grid resolution. Two validation strategies were used in the study. The first is to compare the RZSM time series averaged over all in situ stations with the RZSM time series averaged over all model grids where the stations are located. The second one is the single point-grid validation, the measurements at each station are compared directly with the grid values where the station is located, if there is more than one station, the measurements of these stations are averaged. The point-grid validation has been provided in the supplement (Fig. S2 and S3)."

Most of the in situ stations are located in the Huaibei Plain, which is a major grain production area. According to the land cover map of the Huai River basin (Figure 1c), most of them (56 of 58) is located in the cropland regions, we will add it in the supplement (see Table S1 below). In terms of soil properties, the lime concretion black soil is the main soil type in the Huaibei plain, which is shown in line 482. Since most of the stations in the Huaibei Plain are located in a semi-humid region, which shares the similar meteorological conditions and topography. Therefore, we believe that soil properties, vegetation, precipitation and temperature are homogeneous among different in situ stations. Moreover, the point-grid validation draws the same conclusion as the station-averaged validation. Therefore, the heterogeneity may have little effect on moisture.

Table S1 Overview of in situ stations in Huai River Basin

| Station Name | Longitude (E) | Latitude (N) | Elevation (m) | Land cover |
|---|---|---|---|---|
| Taolaoba | 117.16 | 32.18 | 48 | Irrigated Crop |
| Chahua | 116.02 | 33.03 | 39 | Rainfed Crop |
| Hanting | 116.32 | 33.02 | 28 | Rainfed Crop |
| Songji | 115.27 | 32.82 | 39 | Rainfed Crop |
| Funan | 115.57 | 32.64 | 33 | Rainfed Crop |
| Santa | 115.70 | 32.81 | 33 | Rainfed Crop |
| Yaoli | 116.17 | 31.82 | 58 | Irrigated Crop |
| Guanting | 116.85 | 31.80 | 51 | Irrigated Crop |
| Zhuangmu | 117.11 | 32.36 | 27 | Irrigated Crop |
| Guiji | 116.62 | 32.78 | 23 | Irrigated Crop |
| Xiaji | 116.54 | 32.65 | 25 | Rainfed Crop |
| Shuangfu | 115.57 | 33.34 | 37 | Rainfed Crop |
| Fentai | 115.73 | 33.45 | 35 | Rainfed Crop |
| Santang | 115.83 | 33.31 | 32 | Rainfed Crop |
| Lixin | 116.21 | 33.14 | 28 | Rainfed Crop |
| Jieshou | 115.36 | 33.27 | 42 | Rainfed Crop |
| Yangqiao | 115.39 | 33.02 | 28 | Rainfed Crop |
| Guangwu | 115.33 | 33.37 | 42 | Rainfed Crop |
| Huangling | 115.13 | 33.04 | 37 | Rainfed Crop |
| Quanyang | 115.44 | 33.11 | 35 | Rainfed Crop |
| Kanheliu | 115.85 | 33.10 | 33 | Rainfed Crop |
| Kouziji | 116.09 | 32.84 | 26 | Rainfed Crop |
| Sanshilipu | 116.11 | 32.70 | 27 | Rainfed Crop |
| Xiaqiao | 116.38 | 32.64 | 26 | Rainfed Crop |
| Hengpaitou | 116.36 | 31.59 | 72 | Woodland |
| Xianghongdianxia | 116.18 | 31.58 | 116 | Woodland |
| Wangchenggang | 116.53 | 31.74 | 76 | Irrigated Crop |
| Lumiao | 115.80 | 34.00 | 39 | Rainfed Crop |
| Dasi | 115.87 | 33.80 | 42 | Rainfed Crop |
| Youhe | 115.79 | 33.63 | 38 | Rainfed Crop |

| | | | | |
|---|---|---|---|---|
| Huagou | 116.06 | 33.51 | 33 | Rainfed Crop |
| Dahu | 116.35 | 33.52 | 31 | Rainfed Crop |
| Chenqiao | 116.56 | 33.09 | 25 | Rainfed Crop |
| Heliu | 116.97 | 33.03 | 25 | Rainfed Crop |
| Linhuanzha | 116.57 | 33.67 | 29 | Rainfed Crop |
| Guzhenzha | 117.33 | 33.30 | 18 | Rainfed Crop |
| Wudaogou | 117.34 | 33.16 | 21 | Rainfed Crop |
| Hexiangzha | 117.18 | 33.00 | 18 | Rainfed Crop |
| Tancheng | 116.56 | 33.44 | 29 | Rainfed Crop |
| Xibakou | 117.87 | 33.15 | 11 | Rainfed Crop |
| Xulouzha | 116.75 | 33.92 | 30 | Rainfed Crop |
| Suxianzha | 117.08 | 33.67 | 28 | Rainfed Crop |
| Gukouzha | 116.45 | 34.27 | 39 | Rainfed Crop |
| Kuaitanggou | 117.55 | 33.75 | 20 | Rainfed Crop |
| Yanglou | 116.78 | 34.32 | 39 | Rainfed Crop |
| Langanji | 117.23 | 33.93 | 25 | Rainfed Crop |
| Dulou | 116.85 | 34.20 | 37 | Rainfed Crop |
| Xiangyang | 117.58 | 33.47 | 24 | Rainfed Crop |
| Shuangdui | 116.90 | 33.42 | 25 | Rainfed Crop |
| Shuoli | 116.90 | 34.03 | 32 | Rainfed Crop |
| Huangmiao | 117.65 | 33.08 | 19 | Rainfed Crop |
| Baoji | 117.11 | 33.16 | 22 | Rainfed Crop |
| Dinghouying | 117.34 | 33.46 | 24 | Rainfed Crop |
| Xuanmiao | 116.27 | 34.52 | 54 | Rainfed Crop |
| Longhai | 116.35 | 34.40 | 45 | Rainfed Crop |
| Zhangzhuangzhai | 116.60 | 34.12 | 37 | Rainfed Crop |
| Sixian | 117.92 | 33.43 | 16 | Rainfed Crop |
| Dazhuang | 117.87 | 33.67 | 20 | Rainfed Crop |

Discussion:

The authors explain potential causes for the mismatches between the satellite products and the observations by using specific analyses of the precipitation, temperature and soil type. Many performance metrics have been used throughout the analyses, but I think the use of these different metrics could be exploited more in the discussion. Different metrics represent different aspects of the timeseries, which could explain different processes. The authors could relate the causes in section 5.1 and 5.2 more specifically to the different metrics used. Here, also the vegetation/land cover aspect should be included as mentioned before. Lastly, how easily can we extrapolate these results to other regions?

Response: The section 5.1 and 5.2 will be rephrased as suggested. We will use more metrics and link them to different processes.

Actually, caution is required when extrapolating these results to other regions. Firstly, the uncertainty of precipitation derived from the AGCM varies considerably across different

regions. For example, the large-scale stratiform precipitation is better resolved than the small-scale convective precipitation processes in the AGCM. Therefore, the precipitation simulation in the extratropical zone generally performs better than in the tropical zone, and performs better in winter than in summer (Beck et al., 2019; Lavers et al., 2022). In addition, the underlying surface conditions (e.g., land cover, soil properties) are strongly dependent on the local climate conditions. Regarding the soil properties, there are no observed soil profiles incorporated into the global soil datasets (e.g., HWSD) for some regions.

Irrigation
The role of irrigation in this study is confusing, due to the following statements:
- L119: '76% is irrigated'
- L135: 'Stations are located in areas without irrigation'
- L562: 'heavily irrigated HRB in China'
- L452: 'a signature of irrigation'
- L573: 'indirectly account for irrigation'
I understand that the entire HRB is heavily irrigated, but in this study we only look at the Huaibei Plain which has only rainfed crops as indicated in Fig. 1. It remains unclear to me which area is used for the gridded products, the entire HRB or only the Huaibai Plain? This is not entire clear from the methods. If it is the Huaibai plain (which would make more sense), then irrigation is not an issue in this paper, and should not be emphasized.

Response: The Huaibei Plain is used for the gridded products in this study. According to the land cover map (Figure 1c), the Huaibei Plain has only the rainfed crops (e.g. winter wheat, corn, Soybean, sorghum, sesame, etc.). The irrigated crops in this study mainly refer to rice fields. However, it should be noted that the Huaibei Plain still requires large amounts of irrigation, because the mean annual precipitation is less than mean annual evaporation demand. Therefore, the Huaibei Plain is prone to agricultural drought (Gou et al., 2022). Natural precipitation is the main source of water for the rainfed crops in the Huaibei Plain, supplemented by irrigation when there is no precipitation for a long time.

We completely agree with the comment that the irrigation factor is irrelevant and should not be emphasized. We will delete related statements about irrigation in the revised manuscript. L135 "Stations are located in areas without irrigation" and L452-453 "The overestimation of RZSM by ERA5 (Fig. 3) could be a signature of irrigation because the in situ RZSM observations do not capture irrigation" will be deleted in the revised manuscript.

Specific comments:
- L20,21: What are L4 and L3 here?

Response: L4 and L3 refer to Level 4 and Level 3, respectively. We will replace them with Level 4 (L4) and Level 3 (L3) in the revised manuscript.

- L59: I think ERA5, MERRA2 and NCEP CFSv2 are not the only existing global products, it might be good to emphasize this with for example 'amongst others'.

Response: Agree. L49-67 has been reworded.

"To date, several RZSM products have been developed for broader global applications, such as

the Global Land Data Assimilation System (GLDAS_NOAH and GLDAS_CLSM) (Rodell et al., 2004), the China Land Data Assimilation System (CLDAS) (Shi et al., 2014) and the Soil Moisture Active Passive (SMAP) Level 4 (L4) (Reichle et al., 2012; Reichle et al., 2017a), the European Centre for Medium-Range Weather Forecasts (ECMWF) fifth generation reanalysis (ERA5) (Hersbach et al., 2020), the Modern-Era Retrospective Analysis for Research and Applications version 2 (MERRA-2) (Gelaro et al., 2017), and the National Centers for Environmental Prediction Climate Forecast System version 2 (NCEP CFSv2) (Saha et al., 2014)."

- L75: etc. is not very scientific

Response: We will delete "etc." in the revised manuscript.

- L99: it is maybe not 'difficult', but 'out of scope'

Response: L99-100 "As it is difficult…in this study" will be removed in the revised manuscript.

- L113: Fig. 1 instead of Figure 1

Response: The Figure 1 will be replaced by Fig. 1.

- Figure 1: There are two red lines in the figure, so the legend is not entirely clear

Response: We will revise the Figure 1.

- L144-145: performance metrics of P and T with respect to ground observations? This is not clear.

Response: The text (L142-143) will be rephrased from "The dataset has been extensively validated and is of high quality."

To "The dataset has been extensively validated against ground observations and is of high quality."

- L200: 'Saha and coauthors, 2011' is not a valid reference

Response: We will revise this reference, the "Saha and coauthors, 2011" was replaced by "Saha et al. 2011" in the revised manuscript.

- Chapter 2 Datasets: The authors use many different units for scale, for example 0.5°x0.5° (L137); 1:5 million (L150); 30x30 arcseconds (L163). It would be helpful for the reader to include for each scale metric a rough comparison to for instance kilometre to easily compare the resolution of the different products.

Response: L137 will be replaced by "with a spatial resolution of 0.5° (approximately 55.6 km)".

L150 will be replaced by "Soil databases used in many global LSMs have traditionally relied on the FAO/UNESCO 1:5 million scale World Soil Map with a spatial resolution of 5 arc minutes (approximately 10 km)".

L156 will be replaced by "with a resolution of 30 arcseconds (approximately 1 km)".

L163 will be replaced by "The dataset provides information on soil properties for eight layers

(0-2.3 m) at a spatial resolution of 30×30 arcseconds (approximately 1 km)".

- Section 2.1 'The HRB study area': for the reader the full name of HRB would be more clear

Response: We will use "The Huai River Basin study area" instead of "The HRB study area".

- Section 2.3 and 2.4: the authors describe a lot of different products, but it is not directly clear from these sections what is actually used for this study. Both sections could be much more concise when only referring to the relevant information for this study. For example, it is not directly relevant that ERA5 'covers the period from January 1940 to present … ocean waves' (L170). Table 1 gives a very concise overview, and could be valued more and referred to more often in the text.

Response: The section 2.3 and 2.4 will be rephrased for a more concise description in the revised manuscript.

At the end of section 2.3, the following text will be added in the revised manuscript

"The FAO/UNESCO and HWSD V1.2 soil datasets are employed in different LSMs, respectively. A China soil dataset developed by Shangguan et al., (2013) is used as a reference to evaluate the soil properties of FAO/UNESCO and HWSD V1.2 datasets.

- Table 1: a reference would be more informative than the 'data access' column

Response: The data access will be replaced by the following reference.

GLDAS_NOAH:

Beaudoing, H. and M. Rodell, NASA/GSFC/HSL (2020), GLDAS Noah Land Surface Model L4 3 hourly 0.25 x 0.25 degree V2.1, Greenbelt, Maryland, USA, Goddard Earth Sciences Data and Information Services Center (GES DISC), Accessed: [21 September 2021], 10.5067/E7TYRXPJKWOQ.

GLDAS_CLSM:

Li, B., H. Beaudoing, and M. Rodell, NASA/GSFC/HSL (2020), GLDAS Catchment Land Surface Model L4 daily 0.25 x 0.25 degree GRACE-DA1 V2.2, Greenbelt, Maryland, USA, Goddard Earth Sciences Data and Information Services Center (GES DISC), Accessed: [22 September 2021], 10.5067/TXBMLX370XX8.

ERA5:

Hersbach, H. and Coauthors: ERA5 hourly data on single levels from 1979 to present, Copernicus Climate Change Service (C3S) Climate Data Store (CDS), Accessed: 22 September 2021, https://doi.org/10.24381/cds.adbb2d47, 2018.

MERRA-2:

Global Modeling and Assimilation Office (GMAO) (2015), MERRA-2 tavg1_2d_lnd_Nx: 2d,1-Hourly, Time-Averaged, Single-Level, Assimilation, Land Surface Diagnostics V5.12.4, Greenbelt, MD, USA, Goddard Earth Sciences Data and Information Services Center (GES DISC), Accessed: 26 September 2021, 10.5067/RKPHT8KC1Y1T.

NCEP CFSv2:

Saha, S., et al. 2011, updated monthly. NCEP Climate Forecast System Version 2 (CFSv2) Selected Hourly Time-Series Products. Research Data Archive at the National Center for Atmospheric Research, Computational and Information Systems Laboratory. https://doi.org/10.5065/D6N877VB. Accessed 28 October 2021.

SMAP L4:

Reichle, R., G. De Lannoy, R. D. Koster, W. T. Crow, J. S. Kimball, and Q. Liu. (2020). SMAP L4 Global 3-hourly 9 km EASE-Grid Surface and Root Zone Soil Moisture Geophysical Data, Version 5 [Data Set]. Boulder, Colorado USA. NASA National Snow and Ice Data Center Distributed Active Archive Center. https://doi.org/10.5067/9LNYIYOBNBR5. Date Accessed 06-04-2021.

SMOS

CATDS (2021), CATDS-PDC L4SM RZSM – 1 day global map of root zone soil moisture values from SMOS satellite. CATDS (CNES, IFREMER, CESBIO). http://dx.doi.org/10.12770/316e77af-cb72-4312-96a3-3011cc5068d4. Date Accessed 17-09-2021.

CLDAS:

CMA (2020), The near-real-time product dataset of the China Meteorological Administration Land Data Assimilation System (CLDAS-V2.0). Available at http://data.cma.cn/en/?r=search/uSearch&keywords=cldas. Date Accessed 16-11-2021.

- Section 4.1: the first paragraph is not easily readable for the author due to all the numbers. All the numbers are also presented in Table 3, so it suffices to only mention the highly relevant numbers in the text here. Table 3 could also be combined with Fig. 2, same for Fig. 8 and Table 4.

Response: The first paragraph will be replaced by the following text:

"Figure 2 shows scatterplots of RZSM products against the in situ measurements averaged across all in situ stations over the HRB, from 1 April 2015 to 31 March 2020. Regarding the bias, except for the underestimation by SMOS L4 (-0.047 $m^3$ $m^{-3}$), all the other products overestimate the RZSM observations by 0.030 $m^3$ $m^{-3}$ to 0.117 $m^3$ $m^{-3}$ (SMAP L4 and ERA5, respectively). ERA5 and CLDAS have the largest RMSE values among all the RZMS products due to the relatively large bias. Regarding correlation and ubRMSE, GLDAS_CLSM (R = 0.69, ubRMSE = 0.018 $m^3$ $m^{-3}$) outperforms the other RZSM products, followed by MERRA-2, ERA5, CLDAS, SMAP L4, GLDAS_NOAH, NCEP CFSv2 and SMOS L4. Overall, GLDAS_CLSM performs best among the eight RZSM products in terms of R, ubRMSE and bias values, while SMAP L4 presents the lowest RMSE and the lowest bias. SMOS L4 presents the worst performance with the lowest R value. The detailed statistics are shown in Table 3.

The description about Fig. 8 and Table 4 (L394-398) will be replaced by "

The daily air temperature data derived from ERA5, MERRA-2, NCEP CFSv2, GLDAS_CLSM,

CLDAS, GLDAS_NOAH and SMAP L4 are validated against in situ observations of daily air temperature after aggregating all sub-daily products to daily time steps. Figures 8 and S4 shows that the modelled air temperature captures the observed temporal variation well, with R values above 0.96. However, all of them show slight underestimation, indicated by negative bias values ranging from -4.0 to -5.2 K. In terms of the comprehensive scores of the four statistical metrics, GLDAS_NOAH air temperature outperforms the other datasets and SMAP L4 shows the worst performance. Detailed statistics are shown in Table 4.

- Figure 2 and 8: to improve visualization I would recommend to include density of points with colours (for example https://stackoverflow.com/questions/20105364/how-can-i-make-a-scatter-plot-colored-by-density)

Response: Figures 2 and 8 will be revised to use the same layout as Figure 4.

- Figure 4:
    o The labels are not readable because of the small font

    Response: We will revise Fig. 4.

    o What are the lines? A fit through the data points? Why do the authors use a different layout for a scatterplot than in Fig. 2 and 8?

    Response: The line in each subplot is a fit through the data points. Figures 2 and 8 will be revised to use the same layout as Figure 4. And we will add the explanation in the legend of Fig. 2, 4 and 8.

- Figure 8 and Table 4: I think it is more convenient to use degree Celsius than Kelvin for temperatures.

Response: The Kelvin is the international system (SI) of unit for thermodynamic temperature. The SI units should be used according to the requirement of HESS submission.

- L381: 'good agreement' is a subjective statement, I think it is questionable if a R>0.4 is 'good'.

Response: This L381 will be replaced by "Overall, the R values between precipitation products and the observed precipitation is higher than 0.4 (left panel of Fig. 7)"

- Figure 9: This figure implies that also the models differentiate soil moisture for layer 0-30cm and 30-100cm, while for most models this is not the case?

Response: This figure doesn't mean that the models differentiate soil moisture for layer 0-30cm and 30-100cm. In most LSMs, the soil layers for soil moisture and temperature simulations are generally not for layer 0-30 cm and 30-100 cm, e.g., 0-10 cm, 10-40 cm, 40-100cm and 100-200cm. When the soil properties (top layer: 0-30 cm and subsurface layer 30-100 cm) provided by FAO/UNESCO and HWSD are used in different LSMs, they are processed differently in the LSMs. For example, GLDAS uses the top layer soil parameter data for all layers (see GLDAS Soil Land Surface | LDAS (nasa.gov)). Such as the top layer soil properties are used to represent four soil layers (0-10 cm, 10-40 cm, 40-100 cm and 100-200 cm) for GLDAS_NOAH and two soil layers (surface:0-2 cm, root zone: 0-100 cm) for GLDAS_CLSM in terms of soil moisture.

- Abstract: the statements about the explanations of the differences are quite strong, because

we know there is many other factors that play a role.

Response: We will weaken the statements about the explanations of the differences. The abstract will be replaced by the following text in the revised manuscript.

"Root zone soil moisture (RZSM) is critical for water resource management, drought monitoring and sub-seasonal flood climate prediction. While RZSM is not directly observable from space, several RZSM products are available and widely used at global and continental scales. This study conducts a comprehensive quantitative evaluation of eight RZSM products over the Huai River Basin (HRB) in China. The assessment is performed using observations from 58 in situ soil moisture stations from 1 April 2015 to 31 March 2020. Attention is drawn to the potential factors that contribute to the uncertainties of model-based RZSM, including errors in atmospheric forcing (precipitation, air temperature), vegetation parameterizations, soil properties, and spatial scale mismatch, etc. The results show that the Global Land Data Assimilation System Catchment Land Surface Model (GLDAS_CLSM) outperforms other RZSM products with the highest correlation coefficient (R=0.69) and the lowest unbiased root mean square error (ubRMSE=0.018 $m^3$ $m^{-3}$), respectively. All RZSM products tend to overestimate in situ soil moisture values, except for the Soil Moisture and Ocean Salinity Level 4 (SMOS L4) product, which underestimates RZSM. The underestimation of Surface Soil Moisture (SSM) in SMOS Level 3 (L3), caused by underestimated physical surface temperature and overestimated ERA interim soil moisture, may contribute to the underestimation of RZSM in SMOS L4. The other model-based RZSM products show an overestimation of in situ observations, which could be associated with the overestimation of the precipitation amounts and precipitation events (drizzle effects) and the underestimation of air temperature. In addition, the biased soil texture (organic carbon, clay and sand fractions) and flawed vegetation parameterizations (e.g., canopy and root tissues) affect the hydrothermal transport processes represented in different LSMs, leading to inaccurate soil moisture. The intercomparison of the eight RZSM products shows that MERRA-2 and SMAP L4 RZSM have the highest correlation, which could be attributed to the fact that both products use the catchment land surface model and the atmospheric forcing provided by the Goddard Earth Observing System Model, version 5 (GEOS-5), although the versions differ slightly. This in situ validation shows that GLDAS_CLSM could be used for drought monitoring and flood forecast in Huaibei Plain. Moreover, the RSZM intercomparison indicates that the model should focus on increasing the frequency of dry soil moisture, decreasing the frequency of wet soil moisture and the ability to capture the frequency peak of soil moisture. The uncertainty analysis implies that the model-based RZSM can be improved by correcting precipitation, using more accurate soil properties and more prefect vegetation parameterization schemes, etc."

Reference

Beck, H. E., Pan, M., Roy, T., Weedon, G. P., Pappenberger, F., van Dijk, A. I. J. M., Huffman, G. J., Adler, R. F. and Wood, E. F.: Daily evaluation of 26 precipitation datasets using Stage-IV gauge-radar data for the CONUS, Hydrology and Earth System Sciences, 23, 207-224, 10.5194/hess-23-207-2019, 2019.

Gou, Q., Zhu, Y., Lü, H., Horton, R., Yu, X., Zhang, H., Wang, X., Su, J., Liu, E., Ding, Z., Wang, Z. and Yuan, F.: Application of an improved spatio-temporal identification

method of flash droughts, J. Hydro., 604, 127224, https://doi.org/10.1016/j.jhydrol.2021.127224, 2022.

Lavers, D. A., Simmons, A., Vamborg, F. and Rodwell, M. J.: An evaluation of ERA5 precipitation for climate monitoring, Quarterly Journal of the Royal Meteorological Society, 148, 3152-3165, https://doi.org/10.1002/qj.4351, 2022.

Nogueira, M., Albergel, C., Boussetta, S., Johannsen, F., Trigo, I. F., Ermida, S. L., Martins, J. P. A. and Dutra, E.: Role of vegetation in representing land surface temperature in the CHTESSEL (CY45R1) and SURFEX-ISBA (v8.1) land surface models: a case study over Iberia, Geoscientific Model Development, 13, 3975-3993, https://doi.org/10.5194/gmd-13-3975-2020, 2020.

Reichle, R. H., Draper, C. S., Liu, Q., Girotto, M., Mahanama, S. P. P., Koster, R. D. and De Lannoy, G. J. M.: Assessment of MERRA-2 Land Surface Hydrology Estimates, Journal of Climate, 30, 2937-2960, https://doi.org/10.1175/jcli-d-16-0720.1, 2017.

Rui, H., Beaudoing, H. and Loeser, C.: README Document for NASA GLDAS Version 2 Data Products, Available at https://hydro1.gesdisc.eosdis.nasa.gov/data/GLDAS/GLDAS_NOAH025_3H.2.1/doc/README_GLDAS2.pdf,2021.

Stevens, D., Miranda, P. M. A., Orth, R., Boussetta, S., Balsamo, G. and Dutra, E.: Sensitivity of Surface Fluxes in the ECMWF Land Surface Model to the Remotely Sensed Leaf Area Index and Root Distribution: Evaluation with Tower Flux Data, Atmosphere, 11, http://doi.org/10.3390/atmos11121362, 2020.

van Oorschot, F., van der Ent, R. J., Hrachowitz, M. and Alessandri, A.: Climate-controlled root zone parameters show potential to improve water flux simulations by land surface models, Earth System Dynamics, 12, 725-743, https://doi.org/10.5194/esd-12-725-2021, 2021.

---

## Author Comment (AC4)

**Response to Reviewer #3's comments on the manuscript egusphere-2023-1597**

**RC3: 'Comment on egusphere-2023-1597', Anonymous Referee #3, 23 Nov 2023**

The manuscript presents an intercomparison between eight root zone soil moisture (RZSM) products and in situ measurements. The differences are discussed in the light of the uncertainty in precipitation, air temperature and soil properties.

Overall, I think the study could be a valuable scientific contribution. However, at the current status, I think the manuscript should be improved in several parts before reaching good quality for a possible publication. Specifically, I think the Authors should put major effort into improving the descriptions of the products, the methods should be extended, the discussion should be integrated accordingly. Below I provide additional details about my major concerns followed by specific comments.

The authors thank the Reviewer #3 for her/his constructive and insightful comments that help us improve the quality of the manuscript. The original comments from Reviewer #3 are in black font, and our responses are in blue font.

GENERAL COMMENTS

**Irrigation**

A major confusion in my opinion in the intercomparison is the role of irrigation. At L119 and L124 it stated that the study is conducted over a highly irrigated area. At L135 it is stated that soil moisture sensors are in areas without irrigation. It is not clear by reading if the RZSM products account for irrigation or not and if this can be a concern. Only later in the discussion (L452), it is stated that the overestimation of RZSM by ERA5 (Fig. 3) could be a signature of irrigation because the in situ RZSM observations do not capture irrigation. Does this mean that some RZSM products are based on model that take into account irrigation and others not? On the one hand, this information should better explained and discussed. On the other hand, I wonder what is the scientific meaning of comparing soil moisture in rainfed area to model that are accounting for irrigation.

Response: We completely agree with this comment from reviewer 3, which is also proposed by reviewer 2. For the sake of clarification, we will not emphasize the role of irrigation any more. L135 "Stations are located in areas without irrigation" and L452-453 "The overestimation of RZSM by ERA5 (Fig. 3) could be a signature of irrigation because the in situ RZSM observations do not capture irrigation" will be deleted in the revised manuscript.

And we try to address the confusion mentioned above by reviewer 3.

First, the Huai River Basin is highly irrigated.

Second, L135 "Stations are located in areas without irrigation", it means that the soil moisture probes are installed away from crops, which intends to avoid capturing the irrigation signal and obtain the natural soil moisture states.

Finally, the RZSM products don't take irrigation into account. And L452-453 are incorrect statements and will be removed. For example, ERA5 didn't model irrigation (Lavers et al., 2022).

**Soil map and soil parameters (section 4.3.2 and section 5.2).**

The assessment of the soil properties is valuable. It should be noted, however, that high discrepancies come with the use of different pedotransfer functions (PTF) to derive soil hydraulic parameters (retention curve and hydraulic conductivity). I guess these parameters are used in each RZSM product but estimated with different PTFs. This could also explain part of the uncertainty. This information is missing in the manuscript but should be integrated.

Response: Thank you for the valuable suggestion, we will add the statements about PTFs in section 5.2.

"The soil hydraulic parameters (SHPs), such as the hydraulic conductivity and matric potential, are crucial parameters for the transport of soil moisture between soil layers through the Richards' equation within the LSMs. Generally speaking, the SHPs are derived from a combination of soil properties (clay, sand, silt fractions and organic content, etc.) with pedotransfer functions (PTFs). However, different input variables and functional forms of the continuous PTFs are used in the LSMs. For example, The PTFs could be constructed by multivariate regression models, nonlinear regression models or artificial neural networks (Harrison et al., 2012). The SHPs can also be derived from a combination of soil properties with PTFs or remotely sensed soil moisture retrievals (Santanello et al., 2007). Therefore, soil moisture show great uncertainty and varies considerably across different LSMs. SMAP L4 soil moisture product adopts PTFs provided by Wösten et al. (2001) which takes the organic carbon affecting soil hydraulic and thermal parameters into consider. MERRA-2 adopts PTFs adapted from Cosby et al. (1984) without consideration of organic carbon (De Lannoy et al., 2014). Therefore, the PTFs could also explain part of the uncertainty.

**Spatial aggregation and comparison at each site should be clarified and improved**

As far as I have understood (L267), the comparison is conducted between the spatial average of the 58 in situ observations. It is not well reported how much is the spatial extent of the 58 stations but looking at figure 1 I guess the station covers an area of around 300 x 200 km2. I then deduce that this spatial average is compared to the spatial average of the gridded products (i.e., each product has different resolutions, but more than one cell of the gridded products covers the area of the 58 stations). So first of all it should be better explain how many cells have also been aggregated for each product. Only later in the results section (L317) I discovered that a comparison has also been performed without aggregating spatially. So, first of all, this information should be provided also before in the methods. Moreover, I would also considering moving some plots that are now in supplement to the main manuscript to strengthen the analysis. Anyway, I'm confused by the fact that the comparison is performed at each station. Does this mean that you have always one station against one cell of the gridded products? Please clarify.

Response: We feel sorry to make you confused. And we will try to resolve your confusion.

1. The spatial extent of the 58 stations is approximately $310 \times 330$ km$^2$.

2. For different RZSM products, different numbers of grids are aggregated. For example, CLDAS: 58, GLDAS_CLSM: 50, GLDAS_NOAH: 50, ERA5: 48, MERRA-2: 50, NCEP CFSv2: 55, SMAP L4: 58, SMOS L4: 51.

3. In this manuscript, Fig.2 and 3 represents that the comparison of the RZSM time series

averaged over all in situ stations with the RZSM time series averaged over all model grids where the stations are located. L317 represents the single point-grid validation, the measurements at each station are compared directly with the grid values where the station is located.

The following text (section 3.3 Validation strategies) will be added in chapter 3 Methods.

"In terms of the temporal resolution, except for the RZSM products (e.g., GLDAS_CLSM, SMOS L4) provided on daily time steps, the other sub-daily RZSM datasets (hourly/3-hourly/6-hourly time steps, shown in Table 1) are aggregated to daily average values. Therefore, the aggregated RZSM products could match the observations at daily time intervals. In terms of spatial resolution, we didn't change the spatial resolution of any RZSM products and used the original grid resolution. Two validation strategies were used in the study. The first is to compare the RZSM time series averaged over all in situ stations with the RZSM time series averaged over all model grids where the stations are located (Fig.2 and 3 shown in this study). The second one is the single point-grid validation, the measurements at each station are compared directly with the grid values where the station is located. If there is more than one station within a grid, the measurements of each station that located in the grid are compared to the grid values separately. The point-grid validation has been provided in the supplement (Fig. S2 and S3)."

**Section 2.4: description of the eight RZSM products**
The description of the eight products should be improved in several parts. The information provided for each product is not always consistent. Some products are better described and with more details than others. E.g., for MERRA-2, the description focuses on the precipitation. Instead, ERA5 does not have any information about. NCEP CFSv2 description is very short. Who provided that? What are the main properties? Some characteristics provided should also be put more in relation to the focus of the paper. E.g., for ERA5, the data assimilation system is described in detail. Is that relevant for the purpose of the paper. If yes, it should be clarified. Overall, the main differences between the products relevant for the present study (e.g., soil map, precipitation, land use, irrigation etc.) should be better highlighted. Table 1 should be extended accordingly.

Response: We will revise the whole section 2.4 thoroughly and Table 1, and provide more focused and consistent description of the eight RZSM products in the revised manuscript.

Please note that some relevant information are discussed only later but in my opinion they should be moved to the method section. This would help understanding and strengthening the discussion of the results. E.g., L410 The soil properties data used in the eight RZSM products were all derived from the FAO/UNESCO soil map of World except for CLDAS, which used the soil data developed by Shangguan et al. (2013), and SMAP L4, which used the HWSD soil properties over China. L426. Global precipitation and air temperature forcing data are used in the production of all RZSM products except for SMOS L4. L452. The overestimation of RZSM by ERA5 (Fig. 3) could be a signature of irrigation because the in situ RZSM observations do not capture irrigation.

Response: We will add the following contents to section "3.3 Validation strategies" to make

it easier for the readers to follow and understand. L452 will be removed.

"The Global precipitation and air temperature forcing data are used in the production of all RZSM products except for SMOS L4, which are validated against the China Daily Gridded Ground Precipitation and Air Temperature dataset V2.0 described in section 2.2. The soil properties data used in the eight RZSM products were all derived from the FAO/UNESCO soil map of World except for CLDAS, which used the soil data developed by Shangguan et al. (2013), and SMAP L4, which used the HWSD V1.2 soil properties over China. The China soil dataset developed by Shangguan et al. (2013) is used as a reference to evaluate the accuracy of FAO/UNESCO and HWSD V1.2 soil properties (clay and sand content, organic carbon content and bulk density)."

Figure quality
Figures are not always readable and meaningful. I suggest putting some more effort into evaluating how to present the results. E.g.,
Fig 1. could also shows the pixel size of the products. This would help understanding spatial extend and intercomparisons.

Response: Fig. 1 has been revised and shows the grid size of 0.25 degree covering the in situ stations.

Fig. 4 is not readable at all. Plots and texts are too small.

Response: Fig. 4 has been revised with improved plots and texts.

Fig. 5 can be improved by having only 8 histograms of the RZSM products and overlapping each histogram with the observation's histogram. This could help to visualize the differences.

Response: Fig. 5 has been revised.

Fig.6. The plots are hardly comparable. It could be evaluated to present one plot with all the cumulative precipitation, or histogram of the precipitation etc.

Response: We have presented one new plot with all the cumulative precipitation. Since most of the daily precipitation ranges from 1 to 10 mm day$^{-1}$, the histogram or probability density function of different precipitation datasets can't be well distinguished from each other. Therefore, it is not included in the plot.

Fig.7. It is not clear to me what is actually presented. This is in line to the general comment above about aggregation. Are you comparing spatial averaged precipitation? What does standard deviation refer to? If you compare each rain gauge to pixel wise, how have you aggregated?

Response: For Fig. 7, we compare each rain gauge to pixel wise where the rain gauge is located. The 6 statistical metrics in Fig.7 are calculated at each station, so there are 58 data points for each statistical metric. The histogram represents the median of 58 data points for each statistical metrics between modelled precipitation and observations. The stand deviation represents the variability in the statistical metrics. We will add more detailed descriptions to the legend of Fig. 7.

**Take-home-message**

I'm expecting to read the overall take-home-message. After performing this intercomparison, what can you conclude? Could we trust this products? Where and which conditions? How would you suggest further improving, studies etc.? This is missing throughout the manuscript but should be understandable from the abstract and more extended at the conclusions.

Response: We will provide concise and focused statement about the intercomparison in the abstract, and the main limitations and inspirations of the intercomparison will be illustrated in the conclusions. Overall, the GLDAS_CLSM outperforms the other RZSM products and could be used for drought monitoring and flood forecast in the Huaibei Plain. The intercomparison (revised Fig. 4) shows that SMOS L4 shows the very low correlation (around 0.4) and high dry bias with any one of the other RZSM products. However, except for SMOS L4, the model-based RZSM products show high consistency (R above 0.7) with each other. This phenomenon could be caused by the fact that the precipitation is not used in the production of SMOS L4. So, the SMOS L4 RZSM has a weaker response to precipitation than other RZSM products. Moreover, SMOS L4 RZSM is consistently lower than any one of the other RZSM products. The correlation R between SMAP L4 RZSM and MERRA-2 RZSM is the highest among any two of the seven LSM-based RZSM products, which could be related to both RZSM products use the same CLSM and meteorological forcing derived from GEOS-5 model system. The revised Fig. 5 shows the frequency distribution of normalized soil moisture for eight RZSM products and in situ observations. It is clear that the histograms MERRA-2, GLDAS_CLSM and SMAP L4 shows the better consistence with observations, although they also slightly overestimate the frequency of wet soil moisture. However, all of them didn't capture the peak and underestimate the frequency of normalized soil moisture ranging from 0.2 to 0.4. The other RZSM products show an obvious offset towards wet soil moisture. The data provider should produce less wet soil moisture and more dry soil moisture.

[Figure]

Fig. 5 The histograms of normalised RZSM products (dashed and red lines) and in situ observations (black and solid lines).

**SPECIFIC COMMENTS IN ORDER OF APPEARANCE (L = LINE NUMBER)**

L13. I would not use the term "direct validation" but "assessment".

Response: "A direct validation" will be replaced by "The assessment".

L29. Th abstract focused on describing the actual results. This is fine but I would also expect at the end to read the take-home-message. E.g., what do we learn by this study? Can we trust, use, apply RZSM products? Where? In which conditions? What in our view and based on this study would further suggest to improve the performances?

Response: The abstract will be replaced by the following text in the revised manuscript.

"Root zone soil moisture (RZSM) is critical for water resource management, drought monitoring and sub-seasonal flood climate prediction. While RZSM is not directly observable from space, several RZSM products are available and widely used at global and continental scales. This study conducts a comprehensive quantitative evaluation of eight RZSM products over the Huai River Basin (HRB) in China. The assessment is performed using observations from 58 in situ soil moisture stations from 1 April 2015 to 31 March 2020. Attention is drawn to the potential factors that contribute to the uncertainties of model-based RZSM, including errors in atmospheric forcing (precipitation, air temperature), vegetation parameterizations, soil properties, and spatial scale mismatch, etc. The results show that the Global Land Data Assimilation System Catchment Land Surface Model (GLDAS_CLSM) outperforms other RZSM products with the highest correlation coefficient (R=0.69) and the lowest unbiased root mean square error (ubRMSE=0.018 m$^3$ m$^{-3}$), respectively. All RZSM products tend to overestimate in situ soil moisture values, except for the Soil Moisture and Ocean Salinity Level 4 (SMOS L4) product, which underestimates RZSM. The underestimation of Surface Soil Moisture (SSM) in SMOS Level 3 (L3), caused by underestimated physical surface temperature and overestimated ERA interim soil moisture, may contribute to the underestimation of RZSM in SMOS L4. The other model-based RZSM products show an overestimation of in situ observations, which could be associated with the overestimation of the precipitation amounts and precipitation events (drizzle effects) and the underestimation of air temperature. In addition, the biased soil texture (organic carbon, clay and sand fractions) and flawed vegetation parameterizations (e.g., canopy and root tissues) affect the hydrothermal transport processes represented in different LSMs, leading to inaccurate soil moisture. The intercomparison of the eight RZSM products shows that MERRA-2 and SMAP L4 RZSM have the highest correlation, which could be attributed to the fact that both products use the catchment land surface model and the atmospheric forcing provided by the Goddard Earth Observing System Model, version 5 (GEOS-5), although the versions differ slightly. This in situ validation shows that GLDAS_CLSM could be used for drought monitoring and flood forecast in Huaibei Plain. Moreover, the RSZM intercomparison indicates that the model should focus on increasing the frequency of dry soil moisture, decreasing the frequency of wet soil moisture and the ability to capture the frequency peak of soil moisture. The uncertainty analysis implies that the model-based RZSM can be improved by correcting precipitation, using more accurate soil properties and more prefect vegetation parameterization schemes, etc."

L108-109. Are these two lines really needed here? I would integrate this information later when you speak about comparison. E.g., L133 for the definition of RZSM.

Response: L108-109 will be deleted. L133 will be replaced by "Since the study aims to evaluate

the accuracy of eight RZSM products (0-100 cm) which are summarised in Table 1, the in situ soil moisture measurements at the four depths are depth-weighted averaged to obtain the 0-100 cm soil moisture data."

L128. Table S1 shows the results of the assessment, and it does not provide additional information about the in situ stations. I would remove this cross reference here and rather add Figure 1 where the locations of the in situ stations are shown.

Response: "Table S1" will be replaced by "Fig. 1".

L263. I would extend a bit on the meaning and interpretation for PD, FAR and CSI.

Response: The following text will be added in section "3.1 Statistical metrics".

"POD is the proportion of real precipitation events simulated by AGCM relative to the actual precipitation events, reflecting the ability of AGCM to detect precipitation. FAR is the fraction of unreal precipitation events out of the total precipitation events simulated by AGCM. CSI is a more balanced score that combines the characteristics of false alarms and missed events, representing the probability of successful simulation of AGCM precipitation.

L280. I was expecting to read more about the description of the equation after L280. Any text missing?

Response: We will add the following description after L280.

"where $\theta_{RZSM}$ denotes 0-100 cm RZSM (m$^3$ m$^{-3}$), $\theta_{0-10cm}$, $\theta_{10-40cm}$ and $\theta_{40-100cm}$ denote the 0-10 cm, 10-40 cm and 40-100 cm soil moisture, respectively."

L293. I think here is a good place to cite table 3 as well.

Response: L293-294 will be replaced by "Figure 2 shows scatterplots of RZSM products against the in situ measurements averaged across all in situ stations over the HRB, from 1 April 2015 to 31 March 2020 (see Table 3)."

L295. It is stated that SMOS-L4 underestimates and the other overestimated the observations. By looking at figure 2, I see the opposite. Please double check what you are plotting.

Response: Thank you for bringing up the mistake. We reversed the labels for the x and y axes and we have revised the Figure 2.

L306. Figure 3 shows spatial average of in situ soil moisture and its spatial variability. As far as I have understood (see general comment above on the spatial aggregation), the spatial average of the RZSM products is shown but, if possible, it could be interesting to show here also the spatial variability of the RZSM products.

Response: L306 will be replaced by "Figure 3 shows time series of in situ RZSM observations averaged over all in situ stations". We will add one figure in the supplement regarding the spatial distribution of eight RZSM products averaged from 1 April 2015 to 31 March 2020.

L317. Description of the comparison at each site should be reported as well in the method (section 3.1) See also general comment above on the spatial aggregation.
Response: See the response in **Spatial aggregation and comparison at each site should be**

**clarified and improved (Page 2).**

L319. The method to calculate the anomalies is reported only in the supplement. I think should be moved to the method (section 3.1)

Response: The anomalies metrics are not used in the manuscript.

L585. After summarizing the mani conclusions, I suggest summarizing the outlook of the study. See also general comments above.

Response: The following text will be added in L585.

"(5) Eight RZSM products are evaluated over the Huaibei Plain under the homogeneous underlying surface conditions. However, different vegetation cover has a large impact on the soil moisture simulations. The role of vegetation cover types in the uncertainty of model-based soil moisture can be further explored, and more detailed vegetation parameterizations (canopy and root tissue) can be discussed at the point scale.

(6) Based on the main conclusions of the study, many factors contribute to the uncertainty of model-based RZSM simulations. Precipitation plays a crucial role. The AGCM-derived precipitation can be corrected in different ways to drive LSMs and hydrological models, and to analyze how the simulated land surface states and fluxes to respond the corrected precipitation."

Supplement. I think the supplement should have a title with the name of authors as well.

Response: "Supplement of

Evaluation of root-zone soil moisture products over the Huai river basin"

Liu En et al.

Correspondence to: Yonghua Zhu (zhuyonghua@hhu.edu.cn)" will be added in the supplement.

Reference

Cosby, B., G. Hornberger, R. Clapp and Ginn, T.: A statisticsl exploration of the relationships of soil moisture characteristics to the physical propertie of soil, Water Resour Research, 20, 682-690,1984.

De Lannoy, G. J. M., Koster, R. D., Reichle, R. H., Mahanama, S. P. P. and Liu, Q.: An updated treatment of soil texture and associated hydraulic properties in a global land modeling system, J. Adv. Model Earth Syst., 6, 957-979, https://doi.org/10.1002/2014ms000330, 2014.

Harrison, K. W., Kumar, S. V., Peters-Lidard, C. D. and Santanello, J. A.: Quantifying the change in soil moisture modeling uncertainty from remote sensing observations using Bayesian inference techniques, Water Resources Research, 48, 10.1029/2012wr012337, 2012.

Lavers, D. A., Simmons, A., Vamborg, F. and Rodwell, M. J.: An evaluation of ERA5 precipitation for climate monitoring, Quarterly Journal of the Royal Meteorological Society, 148, 3152-3165, https://doi.org/10.1002/qj.4351, 2022.

Santanello, J. A., Peters-Lidard, C. D., Garcia, M. E., Mocko, D. M., Tischler, M. A., Moran,

M. S. and Thoma, D. P.: Using remotely-sensed estimates of soil moisture to infer soil texture and hydraulic properties across a semi-arid watershed, Remote Sens. Environ., 110, 79–97, 10.1016/j.rse.2007.02.007, 2007.

Wösten, J. H. M., Pachepsky, Y. A. and Rawls, W. J.: Pedotransfer functions: Bridging the gap between available basic soil data and missing soil hydraulic characteristics, J. Hydrol., 251, 123-150, https://doi.org/10.1016/S0022-1694(01)00464-4, 2001.

---

## Author Response (AR1)

**Dear Editor and Reviewers,**

We would like to thank the Editor and three Reviewers for their efforts in handling the manuscript and for their valuable comments to improve the manuscript. We have revised our manuscript thoroughly according to the comments and provide a point-by-point response to the comments from three Reviewers below. The original comments from three Reviewers are in black font, and our responses are in blue font.

On behalf of all co-authors,
En Liu

**Response to Reviewer #1's comments on the manuscript egusphere-2023-1597**

**RC1: 'Comment on egusphere-2023-1597', Anonymous Referee #1, 07 Nov 2023**

The authors attempted to evaluate the predictive performance of eight satellite data-derived root-zone soil moisture data. The authors provided a quantitative statistical analysis of each RZSM product based on average values of in-situ, and remote sensing estimates for the Huai River Basin. The authors concluded that the GLDAS CLSM RZMS products outperform other RZSM products. While the information contained in the manuscript is quantitative, I ended up questioning the potential contribution of this paper to the readers. I have tried to address why I think

The authors thank the Reviewer #1 for her/his constructive and insightful comments that help us improve the quality of the manuscript. The original comments from Reviewer #1 are in black font, and our responses are in blue font.

- Major comments:

1. Although the study provides a significant amount of comparative analysis between remote sensing-derived RZSM products as well as against observational data, the mechanistic understanding and explanation of each result are widely missing across the manuscript. Therefore, the author's rationale about the causes of differences was often too obvious or uncertain.

   Response: Thank you very much for your valuable comment. In this study, we presented an intercomparison between eight root zone soil moisture (RZSM) products and in situ observations. Although the different RZSM products are evaluated quantitatively, it should be noted that the focus of this study is to investigate the uncertainty in RZSM products caused by potential factors. The SMOS Level 4 (L4) RZSM is produced by combining a modified exponential filter and SMOS Level 3 surface soil moisture (SSM). The other seven RZSM products (except SMOS L4) are produced by land surface models driven by surface atmospheric forcing data from atmospheric general circulation models; the ability of the land surface model to simulate states (e.g., RZSM) and fluxes is limited by uncertainties in the meteorological forcing and parameterization schemes, as well as

inadequate model physics. It is difficult to quantify the model physics because different land surface models are used to produce different RZSM products. Therefore, we analyzed the atmospheric forcing data (especially for precipitation, which dominates the terrestrial water cycle), which is considered as the most important factor in determining the accuracy of the modeled RZSM, and the model static parameters (soil properties), which strongly influence the movement of water in the vadose zone, and local underlying surface conditions. The objective of this study is to provide some insights on how to improve the ability of land surface models to simulate land surface states and fluxes by finding the common characteristics of bias existing in different datasets used in land surface models.

2.  Also, the authors argued that GLDAS CLSM-derived RZSM outperforms other RZSM estimates. However, rather than trying to explain the different predictive performances of each RZSM product (against in-situ observations) in relation to soil properties, land cover/use, and vegetation in the study catchment, the authors just used the average of 58 in-situ data as well as satellite products, resulting in 'all-lumped' single time series for each dataset. Thus, it is not convincible to say a certain remote sensing-derived estimates outperform others since the performance differences can be revealed differently depending on soil properties, vegetation, land cover/use, etc.

Response: According to the elevation and land cover map of the Huai River basin (Figure 1b-c) and the overview of in situ stations (see Table S1 below, we will add it in the supplement), most of the in situ stations are located in the Huaibei Plain, which is a major grain production area and has a relatively flat topography. Most of the in situ stations (56 of 58) are located in the cropland regions (see Table S1 below) and have quite similar topography (elevation and slope, etc.). In terms of soil properties, the lime concretion black soil is the dominant soil type in the Huaibei plain, which is shown in line 482. Since most of the stations in the Huaibei Plain are located in a semi-humid region, which shares the similar meteorological conditions and topography. Therefore, we believe that soil properties, vegetation, precipitation and temperature are homogeneous among different in situ stations. In addition, two validation strategies were used in the study. The first is to compare the mean RZSM averaged over all in situ stations with the mean RZSM averaged over all grids. The second one is the point-grid validation, the station measurements are compared directly with the grid value where the station is located, if more than one station, the measurements of these stations are averaged. The point-grid validation has been provided in the supplement and draws the same conclusion as the station-averaged validation that GLDAS_CLSM outperforms other RZSM products. The section "3.3 RZSM products aggregation and validation strategies" will be added in chapter 3 Methods.

Table S1 Overview of in situ stations in the Huai River Basin

| Station Name | Longitude (E) | Latitude (N) | Elevation (m) | Land cover |
|---|---|---|---|---|

| | | | | |
|---|---|---|---|---|
| Taolaoba | 117.16 | 32.18 | 48 | Irrigated Crop |
| Chahua | 116.02 | 33.03 | 39 | Rainfed Crop |
| Hanting | 116.32 | 33.02 | 28 | Rainfed Crop |
| Songji | 115.27 | 32.82 | 39 | Rainfed Crop |
| Funan | 115.57 | 32.64 | 33 | Rainfed Crop |
| Santa | 115.70 | 32.81 | 33 | Rainfed Crop |
| Yaoli | 116.17 | 31.82 | 58 | Irrigated Crop |
| Guanting | 116.85 | 31.80 | 51 | Irrigated Crop |
| Zhuangmu | 117.11 | 32.36 | 27 | Irrigated Crop |
| Guiji | 116.62 | 32.78 | 23 | Irrigated Crop |
| Xiaji | 116.54 | 32.65 | 25 | Rainfed Crop |
| Shuangfu | 115.57 | 33.34 | 37 | Rainfed Crop |
| Fentai | 115.73 | 33.45 | 35 | Rainfed Crop |
| Santang | 115.83 | 33.31 | 32 | Rainfed Crop |
| Lixin | 116.21 | 33.14 | 28 | Rainfed Crop |
| Jieshou | 115.36 | 33.27 | 42 | Rainfed Crop |
| Yangqiao | 115.39 | 33.02 | 28 | Rainfed Crop |
| Guangwu | 115.33 | 33.37 | 42 | Rainfed Crop |
| Huangling | 115.13 | 33.04 | 37 | Rainfed Crop |
| Quanyang | 115.44 | 33.11 | 35 | Rainfed Crop |
| Kanheliu | 115.85 | 33.10 | 33 | Rainfed Crop |
| Kouziji | 116.09 | 32.84 | 26 | Rainfed Crop |
| Sanshilipu | 116.11 | 32.70 | 27 | Rainfed Crop |
| Xiaqiao | 116.38 | 32.64 | 26 | Rainfed Crop |
| Hengpaitou | 116.36 | 31.59 | 72 | Woodland |
| Xianghongdianxia | 116.18 | 31.58 | 116 | Woodland |
| Wangchenggang | 116.53 | 31.74 | 76 | Irrigated Crop |
| Lumiao | 115.80 | 34.00 | 39 | Rainfed Crop |
| Dasi | 115.87 | 33.80 | 42 | Rainfed Crop |
| Youhe | 115.79 | 33.63 | 38 | Rainfed Crop |
| Huagou | 116.06 | 33.51 | 33 | Rainfed Crop |
| Dahu | 116.35 | 33.52 | 31 | Rainfed Crop |
| Chenqiao | 116.56 | 33.09 | 25 | Rainfed Crop |
| Heliu | 116.97 | 33.03 | 25 | Rainfed Crop |
| Linhuanzha | 116.57 | 33.67 | 29 | Rainfed Crop |
| Guzhenzha | 117.33 | 33.30 | 18 | Rainfed Crop |
| Wudaogou | 117.34 | 33.16 | 21 | Rainfed Crop |
| Hexiangzha | 117.18 | 33.00 | 18 | Rainfed Crop |
| Tancheng | 116.56 | 33.44 | 29 | Rainfed Crop |
| Xibakou | 117.87 | 33.15 | 11 | Rainfed Crop |
| Xulouzha | 116.75 | 33.92 | 30 | Rainfed Crop |
| Suxianzha | 117.08 | 33.67 | 28 | Rainfed Crop |
| Gukouzha | 116.45 | 34.27 | 39 | Rainfed Crop |

| Kuaitanggou | 117.55 | 33.75 | 20 | Rainfed Crop |
|---|---|---|---|---|
| Yanglou | 116.78 | 34.32 | 39 | Rainfed Crop |
| Langanji | 117.23 | 33.93 | 25 | Rainfed Crop |
| Dulou | 116.85 | 34.20 | 37 | Rainfed Crop |
| Xiangyang | 117.58 | 33.47 | 24 | Rainfed Crop |
| Shuangdui | 116.90 | 33.42 | 25 | Rainfed Crop |
| Shuoli | 116.90 | 34.03 | 32 | Rainfed Crop |
| Huangmiao | 117.65 | 33.08 | 19 | Rainfed Crop |
| Baoji | 117.11 | 33.16 | 22 | Rainfed Crop |
| Dinghouying | 117.34 | 33.46 | 24 | Rainfed Crop |
| Xuanmiao | 116.27 | 34.52 | 54 | Rainfed Crop |
| Longhai | 116.35 | 34.40 | 45 | Rainfed Crop |
| Zhangzhuangzhai | 116.60 | 34.12 | 37 | Rainfed Crop |
| Sixian | 117.92 | 33.43 | 16 | Rainfed Crop |
| Dazhuang | 117.87 | 33.67 | 20 | Rainfed Crop |

3. It is also not indicated how each satellite-based soil moisture (at multiple depths) and RZSM 'with different spatiotemporal resolutions were aggregated (again, spatially and temporally) to come up with the sets of time series that require consistent temporal scales between them. The method used for spatial aggregation of the gridded-RZSM also needs to be manifested (i.e., methods).

Response: The following text (section 3.3 RZSM products aggregation and validation strategies) will be added in chapter 3 Methods.

"In terms of the temporal resolution, except for the RZSM products (e.g., GLDAS_CLSM, SMOS L4) provided at daily time intervals, the other sub-daily RZSM datasets (hourly/3-hourly/6-hourly time steps, shown in Table 1) are aggregated to daily average values to match the daily sampling frequency of the in situ observations. In terms of spatial resolution, we didn't change the spatial resolution of any RZSM products and used the RZSM time series for each grid where the in situ stations are located. Two validation strategies were used in the study. The first is to compare the RZSM time series averaged over all in situ stations with the RZSM time series averaged over all model grids where the in situ stations are located (Fig.2 and 3 shown in this study). The second one is the point-grid validation, the RZSM measurements at each in situ station are compared directly with the RZSM values for the grid where the in situ station is located, if there is more than one in situ station within a grid, the RZSM measurements at each station are compared to the grid values separately. The point-grid validation is provided in the supplement (Fig. S1 and S2)."

As this is site-specific, it sounds even less convincing that CLSM-derived soil moisture products outperform, and thus it gets more confusing what the authors want to argue from the RZSM products comparison.

Response: On the one hand, we want to evaluate the performance of eight RZSM products in the agricultural crop area, which could provide a more accurate RZSM dataset

for agricultural drought monitoring. The results show that GLDAS_CLSM outperforms the other RZSM products. However, it doesn't mean CLSM-derived soil moisture outperforms, because SMAP L4 and MERRA-2 also use CLSM. More importantly, the focus of this study is to investigate the uncertainty in different RZSM products caused by potential factors, which could provide some insights into how to improve the ability of land surface models to simulate the land surface states and fluxes.

4. There is significant inconsistency (due to the randomness in estimating RZSM from the remote sensing data) between RZSM estimation methods. For example, the authors tried to estimate RZSM using a depth-weighted method, but equation 1 used for in-situ RZSM is different from equation 2, which was used for RZSM estimation from satellite-derived modeled soil moisture.

Response: The in situ soil moisture measurements are available at four depths (10, 20, 40 and 100 cm). However, in addition to the GLDAS_CLSM, MERRA-2, SMAP L4 and SMOS L4, which directly provide the 0-100 cm RZSM, the other model-based soil moisture datasets are provided in different soil layers, i.e., NCEP CFSv2, CLDAS and GLDAS_NOAH ($\theta_{0-10\,cm}$, $\theta_{10-40\,cm}$, $\theta_{40-100\,cm}$), ERA5 ($\theta_{0-7\,cm}$, $\theta_{7-28\,cm}$, $\theta_{28-100\,cm}$). The in situ measurements are for each soil depth, but the model-based RZSM products are for each soil layer. They are not consistent. Therefore, the study uses two different equations to calculate the 0-100 cm RZSM. The in situ RZSM is calculated using a depth-weighted mean of the measurements at four soil depths (10, 20, 40 and 100 cm). The equation 1 was used in the studies by Gao et al. (2017) and Xing et al. (2021). The model-based RZSM is calculated with a weighted average soil moisture at different layers is calculated based on equation 2, which was used in the study by González-Zamora et al. (2016) and Xing et al. (2021) and calculation of SMOS L4 RZSM (Al Bitar et al., 2021).

-Specific comments:

line 39-40: is this sentence needed?

Response: We have deleted this sentence in the revised manuscript.

line 42: duplicate definition of RZSM?

Response: We have deleted this sentence in the revised manuscript.

line 99-100: by this sentence, do you intend not to include any process-based explanation for the soil moisture products? What about attempting to explain the performance differences found among the RZSM products (as this is essentially modeled data) in relation to model structure? Why does CLSM outperform other land models in terms of RZSM products?

Response: This study attempts to investigate the error sources of RZSM products without considering the model structure. While only evaluating the atmospheric forcing, soil texture, and local conditions, we analyze the effects of these error sources on RZSM estimation from

the perspective of physical processes. For example, overestimated precipitation tends to lead to overestimated water-related states (soil moisture) or fluxes (runoff). The clay exhibits stronger water retention capacity compared to sand at the same matric potential, and high soil organic carbon leads to high soil porosity. Therefore, the overestimated clay fraction and soil organic carbon lead to higher water stored in the soil. In addition, different land surface models are used to produce different RZSM products. For example, ERA5 (HTESSEL), MERRA-2 (CLSM), NCEP CFSv2 (Noah), GLDAS_NOAH (Noah), GLDAS_CLSM (CLSM), CLDAS (CLM, CoLM, Noah-MP), SMAP L4 (CLSM), SMOS L4 (exponential filter, not land surface model). Even the same CLSM land surface model is used for both MERRA-2 and SMAP L4, but the model version is a bit different. Therefore, it is difficult to directly quantify the effect of model structure on RZSM. For example, regarding the water and energy balance represented in different LSMs, the partitioning of net radiative energy into latent heat flux, sensible heat flux and ground heat fluxes, and the partitioning the precipitation into interception, evaporation, runoff and infiltration in the land surface as well as the transfer and exchange of water, heat in the vadose zone vary considerably (Chen et al., 2013; Xia et al., 2014; Reichle et al., 2017a; Zheng et al., 2022). For instance, different LSMs simulate different vertical levels for soil moisture and temperature. NOAH LSM, HTESSEL and CLM have 4-, 4- and 10-layers vertical levels for soil moisture and temperature, respectively (Oleson et al., 2004; Rui et al., 2021a). CLSM doesn't have explicit vertical levels for soil moisture which is represented in surface layer (0-2 cm) and root zone layer (0-100 cm) but has six layers for soil temperature (Rui et al., 2021a). In addition, the fundamental land surface element in CLSM is hydrological catchment, the adjacent catchments are deemed as independent of each other and have no fluxes exchange (Koster et al., 2000; Reichle and Koster, 2003). However, the computational unit of CLM is grid cell, where three nested grid levels are included for representing the spatial heterogeneity of land surface (Oleson et al., 2004). NOAH LSM describes the incomplete hydrological cycle process at the grid scale, and it neglects the heterogeneity of soil in a single grid cell, which has great effect on infiltration, then affect the generation and convergence of runoff (Wang and Chen, 2013). HETSSEL also calculates the water and energy balance at the grid cell and does not account for lateral exchange of soil water between adjacent grid cell. Regarding the surface runoff parameterization scheme, the CLM adopts a conceptual form of original TOPMODEL to configure the runoff parameters. On the condition that the top soil layer is impermeable, the runoff is calculated through saturated and unsaturated fractions combined with the sum of the melt water from snowpack and liquid precipitation falling to the land surface (Oleson et al., 2004). A Simple Water Balance (SWB) model is used to parameterize surface runoff in the NOAH LSM, where the surface runoff is obtained from precipitation minus the maximum infiltration. Meanwhile, the process of runoff generation is considered only in the vertical direction in the SWB model. In fact, the range of runoff generation area is variational in the horizontal direction during the precipitation occurs (Wang et al., 2016). Therefore, the inaccurate infiltration and runoff generation scheme lead to the uncertainty of model-generated RZSM. HTESSEL also adopts water balance equation to calculate surface runoff by precipitation plus snowmelt and minus maximum infiltration. Compared with the SWB used in NOAH LSM, an additional snowmelt item was considered in that of HTESSEL. In addition, different maximum infiltration schemes were adopted in HTESSEL and NOAH

LSM, respectively. Unlike the traditional, layer-based models, the catchment-based LSM takes definitely the control of topography on the spatial variability of soil water and its effect on evaporation and runoff into account. In each catchment, CLSM incorporates different parameterization schemes describing the energy budget processes in specific hydrological regimes into each hydrological catchment model depicting the redistribution of water based on topography, which results in reliable estimates of evaporation and runoff (Ducharne et al., 2000; Koster et al., 2000). In this study, the GLDAS_CLSM RZSM product outperforms other model-based RZSM products, but this doesn't mean that CLSM outperforms other land surface models. The accuracy of RZSM also depends on the meteorological forcing.

Chapter 2.4: the information on the spatial and temporal resolution of each data needs to be revisited and clearly indicated.

Response: To provide a concise and clear description of datasets, the information about the spatial and temporal resolution of eight RZSM products is shown in Table 1.

Chapter 3.2: why did you estimate satellite-derived RZSM different from in-situ RZSM? Why equation 1 and 2 are different? How convincing are the RZSM comparisons based on equation 1 and 2?

Response: The in situ soil moisture measurements are available at four depths (10, 20, 40 and 100 cm). However, in addition to the GLDAS_CLSM, MERRA-2, SMAP L4 and SMOS L4, which directly provide the 0-100 cm RZSM, the other model-based soil moisture datasets are provided in different soil layers, i.e., NCEP CFSv2, CLDAS and GLDAS_NOAH ($\theta_{0-10\,cm}$, $\theta_{10-4\,\,cm}$, $\theta_{40-1\,\,\,cm}$), ERA5 ($\theta_{0-7\,cm}$, $\theta_{7-28\,cm}$, $\theta_{28-10\,\,cm}$). The in situ measurements are for each soil depth, but the model-based RZSM products are for each soil layer. They are not consistent. Therefore, the study uses two different equations to calculate the 0-100 cm RZSM. The in situ RZSM is calculated using a depth-weighted mean of the measurements at four soil depths (10, 20, 40 and 100 cm). The equation 1 was used in the studies by Gao et al. (2017) and Xing et al. (2021). The model-based RZSM is calculated with a weighted average soil moisture at different layers is calculated based on equation 2, which was used in the study by González-Zamora et al. (2016) and Xing et al. (2021) and calculation of SMOS L4 RZSM (Al Bitar et al., 2021).

line 293: instead of averaging all in-situ stations, can you think of disaggregating the study basin (and stations) using any available information such as surface soil properties, orography (e.g., slope, and elevation), land cover, and/or vegetation? That will help the readers get more generalizable information and references.

Response: It is a very good and useful suggestion. However, this study pays more attention to the overall soil moisture dynamic measured at 58 stations in croplands rather than a specific station. The underlying surface conditions (e.g. surface soil properties, orography, land cover and vegetation) is considered as homogeneous. 56 of 58 in situ stations are located in crop lands of the Huaibei Plain, and the elevation is quite similar (Table S1). The lime concretion black soil is the dominant soil type in the Huaibei plain. In future study, we will attempt to

investigate the effect of different underlying surface conditions (vegetation types, etc.) on soil moisture estimations for specific station.

line 306-309: This needs to be rephrased. It is hard to understand what is meant.

Response: The text (line 306-309) will be rephrased from "Figure 3 shows time series of in situ RZSM observations averaged over all in situ stations with its spatial variability, and of 3 RZSM products, ERA5, SMOS L4, and GLDAS_CLSM, presenting a marked overestimation, a marked underestimation, and the best overall agreement with in situ observations, respectively. Other products can be seen in Fig. S1."

To "Figure 3 shows the time series of observation- and model-based RZSM averaged over all in situ stations and the grids where the *in situ* stations are located. ERA5, SMOS L4, and GLDAS_CLSM show the highest overestimation, the lowest underestimation, and the best overall agreement with in situ observations, respectively".

The revised Figure 3 is shown below.

[Figure]

Fig. 3 Time series of RZSM (0-100 cm) products and in situ soil moisture observations averaged across all in situ stations from 1 April 2015 to 31 March 31 2020. The dark line and the gray-shaded areas represent the mean and standard deviation of in situ stations observations. Colored lines represent different RZSM products. Daily precipitation is represented by the orange vertical bars.

line 311: can you explain why SMOS L4 showed less rapid changes and smoother trends?

Response: It is well known that the SSM shows a faster response to atmospheric variations than RZSM, especially for precipitation. Therefore, RZSM shows less rapid changes and smoother trends than SSM, which shows a strong variability. On the one hand, SMOS L4 RZSM is estimated from SMOS L3 SSM together with a modified exponential filter with

different parameter T (characteristic time length) proposed by Wagner et al. (1999). The exponential filter can smooth the trend of SSM, the higher the T value, the smoother the trend of RZSM. Most importantly, precipitation with high spatial and temporal variability is the main forcing input of other model-based RZSM products, which show rapid changes and strong response to precipitation. However, precipitation is not used in producing SMOS L4 RZSM. Therefore, SMOS L4 RZSM doesn't respond well to precipitation, and shows less rapid changes and smoother trends.

line 321: can you explain why they did a better job in the wet season compared to the dry season?

Response: In the Huai river basin, more than 60 % of the annual precipitation falls between June and September (wet season), which significantly affects the temporal dynamics of RZSM. The model-based RZSM products generally perform better in the wet season than in the dry season due to the enhanced ability to capture of the temporal dynamics of in situ observations in the wet season and the inertia of remaining high soil moisture values even in the dry season. Generally speaking, the model-based RZSM products show a strong response to precipitation events during the wet season. The temporal dynamics of the model-based RZSM products are in good agreement with RZSM observations. However, the RZSM observations show stronger variability in the dry season than the model-based RZSM datasets, which remain almost unchanged and don't reproduce the temporal dynamics of in situ observations well. This indicates that the land surface models are more sensitive to precipitation events to no precipitation events and show better skill in simulating RZSM when precipitation events occur. For SMOS L4 RZSM, which performs better in the dry season than in the wet season. This may be due to the fact that SMOS L4 RZSM doesn't contain precipitation information, but integrates the ground microwave radiation information captured by SMOS passive radiometers, which is propagated to SMOS L4 RZSM products through SMOS L3 SSM. During the wet season, the signal intensity of the ground microwave radiation obtained by SMOS sensor is attenuated by a substantial increase in water vapor absorption and scattering, leading to the distortion of SMOS L3 SSM retrieval. During the dry season, the physical shape of soil is less disturbed due to less precipitation, and the ground microwave radiation signal is less disturbed by a small amount of water vapor. SMOS sensor can receive the microwave radiation signal that is closer to the actual radiation strength, and retrieve the more accurate soil moisture state. Therefore, SMOS L4 RZSM performs better in the dry season than in the wet season.

line 360: can you explain why individual satellite-based RZSM products showed different probabilistic distributions? Some are log-normal and the others are normal. Can you add more explanation on this matter?

Response: The peak of the relative frequency for model-based RZSM products ranges from 0.3 to 0.6. RZSM products with log-normal distribution show that low values dominate the RZSM time series, which could be caused by low precipitation. The precipitation field derived from the atmospheric general circulation model (AGCM) generally has too many drizzle events (<1 mm day$^{-1}$). The modeled RZSM is affected by meteorological forcing,

model structure and parameterization, etc. The RZSM estimates are subject to random error and systematic bias, and it is difficult to directly quantify which factor affects the probability distributions of different RZSM products. And it is beyond the scope of this study.

line 375: how does this ground-based observation of precipitation (840 mm/year) represent the average precipitation of the basin area? You also compared gridded-precipitation with this in-situ precipitation observation (line 430). Can you clarify how solid the comparison of this in-situ precipitation with gridded precipitation is?

Response: In this study, the ground-based precipitation observation doesn't represent the average precipitation of the watershed area. We only compare the ground-based precipitation observations at each in situ station with the modeled precipitation values of the grid where the in situ station is located.

line 432: do you think MERRA-2 and GLDAS-CLSM would outperform other satellite-derived RZSM in other basins (or area) as well? What if you perform a continental-scale study, will you still think there will be a certain winner? If not, how can you limit the scale of this sort of comparison study to be meaningful and convincing?

Response: In this study, MERRA-2 and GLDAS-CLSM outperform other model-based RZSM products in the Huaibei Plain, where cropland dominates. It is uncertain whether MERRA-2 and GLDAS-CLSM would still outperform other products if this study were conducted in other basins (areas) or on a continental scale. Because the accuracy of RZSM simulated by land surface model dependent on many factors, such as the local climate conditions and underlying surface conditions. The accuracy of precipitation derived from the atmospheric general circulation model vary considerably across different regions. For example, these large-scale atmospheric processes over the extra-tropics are better resolved in the AGCM than convective processes over the tropics. It is a study for specific underlying surface conditions (agricultural crop region), the evaluation of eight RZSM products in the agricultural crop region could provide a more accurate RZSM dataset for agricultural drought monitoring. More importantly, the focus of this study is to investigate the sources of error of the different RZSM products, which could provide some insights to improve the ability of land surface models to simulate the land surface states and fluxes.

line 436-438: the sentences need to be re-structured to clarify the argument.

Response: The text (line 436-438) will be rephrased from "The MERRA-2 model background precipitation corrected with NOAA CPCU gauge-based precipitation observations was implemented in the coupled land-atmosphere reanalysis system, which may also contribute to the high consistency with the ground-based precipitation"

To "Before driving the land surface water budget, the MERRA-2 model background precipitation was corrected with NOAA CPCU gauge-based precipitation in the coupled land-atmosphere reanalysis system. This correction leads to more accurate precipitation fields for MERRA-2".

line 453: in-situ RZSM observation does not capture irrigation effect? Can you explain how the irrigation water supply does not impact the soil moisture content?

Response: We didn't express it clearly. The original meaning of this sentence is that the in situ station does not capture the irrigation signal. Because the in situ stations are usually installed away from the cropland to avoid the effect of anthropogenic irrigation on the original soil water content supplied by precipitation. In addition, reviewer2 and reviewer3 also raise question about question. The overall comments from three reviewers indicate that the irrigation is not an issue in this paper, and should not be emphasized. We will delete relevant statements about irrigation.

line 485-489: can you add more information on how the soil properties could end up in certain ranges of soil moisture values?

Response: The text (line 485-489) will be replaced by

"For a given soil profile, porosity decreases with depth and clay content increases with depth, resulting in a decrease in hydraulic conductivity. Expansive montmorillonite clay minerals are the main constituents of the lower black soil layer, giving the soil strong expansion and contraction and a high dry bulk density. During dry periods, cracks in the soil column widen and deepen, resulting in capillary breakage. This makes it difficult for groundwater and RZSM to recharge crops, even though the groundwater is shallow. In addition, the increased cracks in the soil column exacerbates the evaporation of soil moisture in the root zone, ultimately leading to frequent droughts. During wet periods, when precipitation or irrigation occurs, soil particles absorb water and swell, closing the cracks and preventing water infiltration. Water is then lost mainly in the form of surface runoff. The crops are prone to waterlogging disasters. This could explain the lower RZSM values ranging from 0.2 to 0.3 $m^3$ $m^{-3}$ observed in the Huaibei plain and the higher RZSM values ranging from 0.3 to 0.4 $m^3$ $m^{-3}$ observed in Jiangsu. The larger amount of precipitation in Jiangsu could be another possible reason."


This paper presents a study on the evaluation of different global root zone soil moisture products and local observations for the Huai River Basin in China. The authors present detailed information on the local conditions, and the different gridded products used. They comprehensively compare the different products with each other and with the observations. Also, the authors provide a discussion on the potential reasons for the differences found. In general, it is an interesting study with a lot of analyses and clear visualization of the results. Nevertheless, I have a few comments that must be addressed before the manuscript can be published.

The authors thank the Reviewer #2 for her/his constructive and insightful comments that help us improve the quality of the manuscript. The original comments from Reviewer #2 are in black font, and our responses are in blue font.

**General comments:**

The influence of land cover, vegetation and root representation:

The authors clearly discuss potential reasons for the mismatch between in situ RZSM observations and the global products, such as forcing data and soil texture maps. Also, shortly 'different model structures and parameterizations' (L89) are mentioned as potential cause for differences. I do think there is one more very important aspect that is missed here: the role of land cover and vegetation, vegetation roots, and soil evaporation and transpiration model representation. Vegetation is usually represented by land cover maps (that are usually prescribed similar to soil maps), which can be very different for the different models. Other relevant vegetation model properties could be Leaf Area Index (see for example Nogueira et al., 2020) or the root parameterization (e.g. Stevens et al., 2020 and Van Oorschot et al., 2021). Furthermore, transpiration of crops is very dependent on the growing season, which might be not represented by the global products. I think these issues should be specifically addressed in the introduction and discussion of the results.

Response: Thank you for rigorous consideration. We do agree with your idea. We will mention these issues in the introduction and discussion.

The following text will be added in the introduction.

The text (Line 88-89) will be replaced by

[revised manuscript text omitted]

Introduction

L49-67: I think this paragraph is intended to describe the state-of-the-art of global surface soil moisture, and root zone soil moisture products. The authors mention many long names of different products, which shows the detailed literature review done for this study. However, for the reader it would be more clear if the paragraphs gives a more general overview, rather than all the specific products, by answering questions such as: Why do we only have SSM direct retrievals, and not RZSM? What is available for global RZSM? How are the RZSM products generated in general?

Response: The text L49-67 will be rephrased from "Recent satellite soil moisture…exponential filter model (Albergel et al., 2008; Al Bitar and Mahmoodi, 2020)."

To "Recently, microwave-based satellite missions provide global soil moisture retrievals with approximately 3-day temporal resolution, but are limited to the top few centimeters (0-5 cm for L-band) due to the limitations of microwave penetration depth (Kerr et al., 2001; Reichle et al., 2017b). Therefore, various approaches have been developed to estimate the RZSM and are roughly divided into three categories (Liu et al., 2023), including, (1) statistics-based methods, such as linear regression (Zhang et al., 2017) and cumulative distribution function (Gao et al., 2019), (2) data-driven machine learning methods, such as rand forest (Carranza et al., 2021) and artificial neural network (Kornelsen et al., 2014), (3) physically based methods, such as data assimilation of satellite-derived observations into LSMs (Albergel et al., 2017; Bonan et al., 2020). Among them, the assimilation of satellite-derived observations into LSMs is considered as the most accurate method to estimate RZM due to the explicit physical mechanism, while requiring large amounts of input data (precipitation, air temperature, radiation, etc.). To date, several RZSM products have been developed for broader global-scale applications, such as the Global Land Data Assimilation System (GLDAS_NOAH and GLDAS_CLSM) (Rodell et al., 2004), the China Land Data Assimilation System (CLDAS) (Shi et al., 2014) and the Soil Moisture Active Passive (SMAP) Level 4 (L4) (Reichle et al., 2012; Reichle et al., 2017a), the European Centre for Medium-Range Weather Forecasts (ECMWF) fifth generation reanalysis (ERA5) (Hersbach et al., 2020), the Modern-Era

Retrospective Analysis for Research and Applications version 2 (MERRA-2) (Gelaro et al., 2017), and the National Centers for Environmental Prediction Climate Forecast System version 2 (NCEP CFSv2) (Saha et al., 2014). These RZSM products are generated by combining LSMs driven by meteorological forcing fields from atmospheric general circulation model (AGCM) and satellite-derived data using different data assimilation techniques (Calvet and Noilhan, 2000; Rodell et al., 2004). In addition, the Soil Moisture and Ocean Salinity (SMOS) Centre Aval de Traitement des Données (CATDS) provides SMOS L4 RZSM products, which are derived from SMOS Level 3 (L3) 3-day SSM retrievals using a statistical exponential filter model (Albergel et al., 2008; Al Bitar and Mahmoodi, 2020)."

L76-94: I think this should go before L68-75.

Response: Line 76-94 has been moved before Line 68-95 in the revised manuscript.

Scale issue:
The authors mention that the scale mismatch is a relevant aspect for the differences between RZSM observed in this study (L75 and Sect. 5.4). However, it is not clear from the methodology how the gridded products are aggregated to match the in situ measurements, and how the scale of the different products compare to the size of the HRB selected area. Moreover, how heterogeneous is the selected area in terms of precipitation, temperature, vegetation, and soil moisture observations? Since the results are mostly based on averages of the area, does the heterogeneity play a role?

Response: The following text (section 3.3 RZSM products aggregation and validation strategies) will be added in chapter 3 Methods.

"In terms of the temporal resolution, except for the RZSM products (e.g., GLDAS_CLSM, SMOS L4) provided at daily time intervals, the other sub-daily RZSM datasets (hourly/3-hourly/6-hourly time steps, shown in Table 1) are aggregated to daily average values to match the daily sampling frequency of the in situ observations. In terms of spatial resolution, we didn't change the spatial resolution of any RZSM products and used the RZSM time series for each grid where the in situ stations are located. Two validation strategies were used in the study. The first is to compare the RZSM time series averaged over all in situ stations with the RZSM time series averaged over all model grids where the in situ stations are located (Fig.2 and 3 shown in this study). The second one is the point-grid validation, the RZSM measurements at each in situ station are compared directly with the RZSM values for the grid where the in situ station is located, if there is more than one in situ station within a grid, the RZSM measurements at each station are compared to the grid values separately. The point-grid validation is provided in the supplement (Fig. S1 and S2)."

According to the elevation and land cover map of the Huai River basin (Figure 1b-c) and the overview of in situ stations (see Table S1 below, we will add it in the supplement), most of the in situ stations are located in the Huaibei Plain, which is a major grain production area and has a relatively flat topography. Most of the in situ stations (56 of 58) are located in the cropland regions (see Table S1 below) and have quite similar topography (elevation and slope, etc.). In terms of soil properties, the lime concretion black soil is the dominant soil type in the Huaibei plain, which is shown in line 482. Since most of the stations in the Huaibei Plain are located in

a semi-humid region, which shares the similar meteorological conditions and topography. Therefore, we believe that soil properties, vegetation, precipitation and temperature are homogeneous among different in situ stations.   Moreover, the point-grid validation draws the same conclusion as the station-averaged validation. Therefore, the heterogeneity may have little effect on moisture.

Table S1 Overview of in situ stations in the Huai River Basin

| Station Name | Longitude (E) | Latitude (N) | Elevation (m) | Land cover |
|---|---|---|---|---|
| Taolaoba | 117.16 | 32.18 | 48 | Irrigated Crop |
| Chahua | 116.02 | 33.03 | 39 | Rainfed Crop |
| Hanting | 116.32 | 33.02 | 28 | Rainfed Crop |
| Songji | 115.27 | 32.82 | 39 | Rainfed Crop |
| Funan | 115.57 | 32.64 | 33 | Rainfed Crop |
| Santa | 115.70 | 32.81 | 33 | Rainfed Crop |
| Yaoli | 116.17 | 31.82 | 58 | Irrigated Crop |
| Guanting | 116.85 | 31.80 | 51 | Irrigated Crop |
| Zhuangmu | 117.11 | 32.36 | 27 | Irrigated Crop |
| Guiji | 116.62 | 32.78 | 23 | Irrigated Crop |
| Xiaji | 116.54 | 32.65 | 25 | Rainfed Crop |
| Shuangfu | 115.57 | 33.34 | 37 | Rainfed Crop |
| Fentai | 115.73 | 33.45 | 35 | Rainfed Crop |
| Santang | 115.83 | 33.31 | 32 | Rainfed Crop |
| Lixin | 116.21 | 33.14 | 28 | Rainfed Crop |
| Jieshou | 115.36 | 33.27 | 42 | Rainfed Crop |
| Yangqiao | 115.39 | 33.02 | 28 | Rainfed Crop |
| Guangwu | 115.33 | 33.37 | 42 | Rainfed Crop |
| Huangling | 115.13 | 33.04 | 37 | Rainfed Crop |
| Quanyang | 115.44 | 33.11 | 35 | Rainfed Crop |
| Kanheliu | 115.85 | 33.10 | 33 | Rainfed Crop |
| Kouziji | 116.09 | 32.84 | 26 | Rainfed Crop |
| Sanshilipu | 116.11 | 32.70 | 27 | Rainfed Crop |
| Xiaqiao | 116.38 | 32.64 | 26 | Rainfed Crop |
| Hengpaitou | 116.36 | 31.59 | 72 | Woodland |
| Xianghongdianxia | 116.18 | 31.58 | 116 | Woodland |
| Wangchenggang | 116.53 | 31.74 | 76 | Irrigated Crop |
| Lumiao | 115.80 | 34.00 | 39 | Rainfed Crop |
| Dasi | 115.87 | 33.80 | 42 | Rainfed Crop |
| Youhe | 115.79 | 33.63 | 38 | Rainfed Crop |
| Huagou | 116.06 | 33.51 | 33 | Rainfed Crop |
| Dahu | 116.35 | 33.52 | 31 | Rainfed Crop |
| Chenqiao | 116.56 | 33.09 | 25 | Rainfed Crop |
| Heliu | 116.97 | 33.03 | 25 | Rainfed Crop |

| Linhuanzha | 116.57 | 33.67 | 29 | Rainfed Crop |
|---|---|---|---|---|
| Guzhenzha | 117.33 | 33.30 | 18 | Rainfed Crop |
| Wudaogou | 117.34 | 33.16 | 21 | Rainfed Crop |
| Hexiangzha | 117.18 | 33.00 | 18 | Rainfed Crop |
| Tancheng | 116.56 | 33.44 | 29 | Rainfed Crop |
| Xibakou | 117.87 | 33.15 | 11 | Rainfed Crop |
| Xulouzha | 116.75 | 33.92 | 30 | Rainfed Crop |
| Suxianzha | 117.08 | 33.67 | 28 | Rainfed Crop |
| Gukouzha | 116.45 | 34.27 | 39 | Rainfed Crop |
| Kuaitanggou | 117.55 | 33.75 | 20 | Rainfed Crop |
| Yanglou | 116.78 | 34.32 | 39 | Rainfed Crop |
| Langanji | 117.23 | 33.93 | 25 | Rainfed Crop |
| Dulou | 116.85 | 34.20 | 37 | Rainfed Crop |
| Xiangyang | 117.58 | 33.47 | 24 | Rainfed Crop |
| Shuangdui | 116.90 | 33.42 | 25 | Rainfed Crop |
| Shuoli | 116.90 | 34.03 | 32 | Rainfed Crop |
| Huangmiao | 117.65 | 33.08 | 19 | Rainfed Crop |
| Baoji | 117.11 | 33.16 | 22 | Rainfed Crop |
| Dinghouying | 117.34 | 33.46 | 24 | Rainfed Crop |
| Xuanmiao | 116.27 | 34.52 | 54 | Rainfed Crop |
| Longhai | 116.35 | 34.40 | 45 | Rainfed Crop |
| Zhangzhuangzhai | 116.60 | 34.12 | 37 | Rainfed Crop |
| Sixian | 117.92 | 33.43 | 16 | Rainfed Crop |
| Dazhuang | 117.87 | 33.67 | 20 | Rainfed Crop |

Discussion:

The authors explain potential causes for the mismatches between the satellite products and the observations by using specific analyses of the precipitation, temperature and soil type. Many performance metrics have been used throughout the analyses, but I think the use of these different metrics could be exploited more in the discussion. Different metrics represent different aspects of the timeseries, which could explain different processes. The authors could relate the causes in section 5.1 and 5.2 more specifically to the different metrics used. Here, also the vegetation/land cover aspect should be included as mentioned before. Lastly, how easily can we extrapolate these results to other regions?

Response: The section 5.1 and 5.2 have been rephrased as suggested. We use the metrics more to explain different processes.

Actually, caution is required when extrapolating these results to other regions. Firstly, the uncertainty of precipitation derived from the AGCM varies considerably across different regions. For example, the large-scale stratiform precipitation is better resolved than the small-scale convective precipitation processes in the AGCM. Therefore, the precipitation simulation in the extratropical zone generally performs better than in the tropical zone, and performs better in winter than in summer (Beck et al., 2019; Lavers et al., 2022). In addition, the underlying

surface conditions (e.g., land cover, soil properties) are strongly dependent on the local climate conditions. Especially the high-vegetation regions, the parameterization for vegetation canopy and root is different with that for low vegetation, which affect the partitioning of rainfall into runoff and infiltration, and the local soil evaporation and transpiration. These differences further affect the near surface air temperature via land-atmosphere coupling, and finally affect the land surface states. Regarding the soil properties, there are no observed soil profiles incorporated into the global soil datasets (e.g., HWSD) for some regions, which will lead to the flawed soil hydraulic parameters and the inaccurate description of the vertical movement of water in the soil column.

Irrigation

The role of irrigation in this study is confusing, due to the following statements:
- L119: '76% is irrigated'
- L135: 'Stations are located in areas without irrigation'
- L562: 'heavily irrigated HRB in China'
- L452: 'a signature of irrigation'
- L573: 'indirectly account for irrigation'

I understand that the entire HRB is heavily irrigated, but in this study we only look at the Huaibei Plain which has only rainfed crops as indicated in Fig. 1. It remains unclear to me which area is used for the gridded products, the entire HRB or only the Huaibai Plain? This is not entire clear from the methods. If it is the Huaibai plain (which would make more sense), then irrigation is not an issue in this paper, and should not be emphasized.

Response: The Huaibei Plain is used for the gridded products in this study. According to the land cover map (Figure 1c), the Huaibei Plain has only the rainfed crops (e.g. winter wheat, corn, soybean, sorghum, sesame, etc.). The irrigated crops in this study mainly refer to rice fields. However, it should be noted that the Huaibei Plain still requires large amounts of irrigation, because the mean annual precipitation of 888 mm is less than mean annual evaporation demand (900-1500 mm). Therefore, the Huaibei Plain is prone to agricultural drought (Gou et al., 2022). Natural precipitation is the main source of water for the rainfed crops in the Huaibei Plain, supplemented by irrigation when there is no precipitation for a long time.

We completely agree with the comment that the irrigation factor is irrelevant and should not be emphasized. We have deleted related statements about irrigation in the revised manuscript. L135, L452-453 and L573-574 are deleted.

Specific comments:
- L20,21: What are L4 and L3 here?

Response: L4 and L3 refer to Level 4 and Level 3, respectively. We have replaced them with Level 4 (L4) and Level 3 (L3) in the revised manuscript.

- L59: I think ERA5, MERRA2 and NCEP CFSv2 are not the only existing global products, it might be good to emphasize this with for example 'amongst others'.

Response: Agree. L49-67 has been reworded, refer to the response to Introduction.

- L75: etc. is not very scientific

Response: "etc." has been deleted.

- L99: it is maybe not 'difficult', but 'out of scope'

Response: L99-100 has been removed.

- L113: Fig. 1 instead of Figure 1

Response: Correction done.

- Figure 1: There are two red lines in the figure, so the legend is not entirely clear

Response: Figure 1 has been revised.

[Figure]

Fig. 1 Overview of the study area and distribution and land cover of in situ soil moisture stations (green pentagon). The squares in Fig.1b and c represent 0.25° grid.

- L144-145: performance metrics of P and T with respect to ground observations? This is not clear.

Response: The text (L142-143) has been rephrased from "The dataset has been extensively validated and is of high quality."

To "The dataset has been extensively validated against ground observations and is of high quality."

- L200: 'Saha and coauthors, 2011' is not a valid reference

Response: We have revised this reference, the "Saha and coauthors, 2011" was replaced by

"Saha et al. 2011".

- Chapter 2 Datasets: The authors use many different units for scale, for example 0.5°x0.5° (L137); 1:5 million (L150); 30x30 arcseconds (L163). It would be helpful for the reader to include for each scale metric a rough comparison to for instance kilometre to easily compare the resolution of the different products.

Response: L137 has been replaced by "with a spatial resolution of 0.5° (approximately 55.6 km)".

L150 has been replaced by "Soil databases used in many global LSMs have traditionally relied on the FAO/UNESCO 1:5 million scale World Soil Map with a spatial resolution of 5 arc minutes (approximately 10 km)".

L156 will be replaced by "with a resolution of 30 arcseconds (approximately 1 km)".

L163 will be replaced by "The dataset provides information on soil properties for eight layers (0-2.3 m) at a spatial resolution of 30×30 arcseconds (approximately 1 km)".

- Section 2.1 'The HRB study area': for the reader the full name of HRB would be more clear

Response: "The HRB study area" has been replaced by "The Huai River Basin study area".

- Section 2.3 and 2.4: the authors describe a lot of different products, but it is not directly clear from these sections what is actually used for this study. Both sections could be much more concise when only referring to the relevant information for this study. For example, it is not directly relevant that ERA5 'covers the period from January 1940 to present … ocean waves' (L170). Table 1 gives a very concise overview, and could be valued more and referred to more often in the text.

Response: The section 2.3 and 2.4 has been rephrased for a more concise and focused description in the revised manuscript.

The following text has been added in "section 3.3 RZSM products aggregation and validation strategies".

[revised manuscript text omitted]

o What are the lines? A fit through the data points? Why do the authors use a different layout for a scatterplot than in Fig. 2 and 8?

Response: The line in each subplot is a fit through the data points. Figures 2 and 8 have been revised to use the same layout as Figure 4. And we will add the explanation in the legend of Fig. 2, 4 and 8.

- Figure 8 and Table 4: I think it is more convenient to use degree Celsius than Kelvin for temperatures.

Response: The Kelvin is the international system (SI) of unit for thermodynamic temperature. The SI units should be used according to the requirement of HESS submission.

- L381: 'good agreement' is a subjective statement, I think it is questionable if a R>0.4 is 'good'.

Response: This L381 will be replaced by "Overall, the R values between precipitation products and the observed precipitation are higher than 0.4 (left panel of Fig. 7)"

- Figure 9: This figure implies that also the models differentiate soil moisture for layer 0-30cm and 30-100cm, while for most models this is not the case?

Response: This figure doesn't mean that the models differentiate soil moisture for layer 0-30 cm and 30-100 cm. In most LSMs, the soil layers for soil moisture and temperature simulations are generally not for layer 0-30 cm and 30-100 cm, such as the Noah LSM simulates four soi layers (0-10 cm, 10-40 cm, 40-100 cm and 100-200 cm). When the soil properties (top layer: 0-30 cm and subsurface layer 30-100 cm) provided by FAO/UNESCO and HWSD are used in

different LSMs, they are processed differently in the LSMs. For example, GLDAS uses the top layer soil parameter data for all layers (see GLDAS Soil Land Surface | LDAS (nasa.gov)), i.e. the soil properties for the top layer are used to represent four soil layers (0-10 cm, 10-40 cm, 40-100 cm and 100-200 cm) for GLDAS_NOAH and two soil layers (surface:0-2 cm, root zone: 0-100 cm) for GLDAS_CLSM in terms of soil moisture.

- Abstract: the statements about the explanations of the differences are quite strong, because we know there is many other factors that play a role.

Response: We will weaken the statements about the explanations of the differences. The abstract has been replaced by the following text in the revised manuscript.

"Root zone soil moisture (RZSM) is critical for water resource management, drought monitoring and sub-seasonal flood climate prediction. While RZSM is not directly observable from space, several RZSM products are available and widely used at global and continental scales. This study conducts a comprehensive and quantitative evaluation of eight RZSM products using observations from 58 *in situ* soil moisture stations over the Huai River Basin (HRB) in China. Attention is drawn to the potential factors that contribute to the uncertainties of model-based RZSM, including the errors in atmospheric forcing, vegetation parameterizations, soil properties, and spatial scale mismatch. The results show that the Global Land Data Assimilation System Catchment Land Surface Model (GLDAS_CLSM) outperforms the other RZSM products with the highest correlation coefficient ($R = 0.69$) and the lowest unbiased root mean square error (ubRMSE = 0.018 $m^3 m^{-3}$), and shows the potential for drought monitoring and flood forecast in Huaibei Plain. While SMOS Level 4 (L4) RZSM shows a much lower correlation with in situ observations than model-based RZSM products forced by precipitation, this could be due to the fact that SMOS L4 does not contain precipitation information and has a weaker response to precipitation. The model-based RZSM products generally perform better in the wet season than in the dry season due to the enhanced ability to capture of the temporal dynamics of in situ observations in the wet season and the inertia of remaining high soil moisture values even in the dry season. While SMOS L4 performs better in the dry season than in the wet season, because the ground microwave radiation signal is more attenuated in the wet season due to a substantial increase in water vapor absorption and scattering than in the dry season, which is used to retrieve SMOS Level 3 (L3) SSM and is propagated to SMOS L4 RZSM. The underestimation of Surface Soil Moisture (SSM) in SMOS Level 3 (L3), caused by underestimated physical surface temperature and overestimated ERA interim soil moisture, may trigger the underestimation of RZSM in SMOS L4. The seven model-based RZSM products show an overestimation of in situ observations, which could be associated with the overestimation of precipitation amounts, the frequency of precipitation events (drizzle effects) and the underestimation of air temperature and the underestimated ratio of transpiration to the total terrestrial evapotranspiration. In addition, the biased soil properties (organic carbon, clay and sand fractions) and flawed vegetation parameterizations (e.g., canopy, root tissue and soil evaporation and transpiration model representation) affect the hydrothermal transport processes represented in different LSMs and lead to inaccurate soil moisture simulation. The scale mismatch between point and footprint also introduces representative errors. The comparison of frequency of normalized soil moisture between RZSM products and in situ observations indicates that the LSMs should focus on reducing the frequency of wet soil

moisture, increasing the frequency of dry soil moisture and the ability to capture the frequency peak of soil moisture."


The manuscript presents an intercomparison between eight root zone soil moisture (RZSM) products and in situ measurements. The differences are discussed in the light of the uncertainty in precipitation, air temperature and soil properties.
Overall, I think the study could be a valuable scientific contribution. However, at the current status, I think the manuscript should be improved in several parts before reaching good quality for a possible publication. Specifically, I think the Authors should put major effort into improving the descriptions of the products, the methods should be extended, the discussion should be integrated accordingly. Below I provide additional details about my major concerns followed by specific comments.

The authors thank the Reviewer #3 for her/his constructive and insightful comments that help us improve the quality of the manuscript. The original comments from Reviewer #3 are in black font, and our responses are in blue font.

GENERAL COMMENTS

**Irrigation**

A major confusion in my opinion in the intercomparison is the role of irrigation. At L119 and L124 it stated that the study is conducted over a highly irrigated area. At L135 it is stated that soil moisture sensors are in areas without irrigation. It is not clear by reading if the RZSM products account for irrigation or not and if this can be a concern. Only later in the discussion (L452), it is stated that the overestimation of RZSM by ERA5 (Fig. 3) could be a signature of irrigation because the in situ RZSM observations do not capture irrigation. Does this mean that some RZSM products are based on model that take into account irrigation and others not? On the one hand, this information should better explained and discussed. On the other hand, I wonder what is the scientific meaning of comparing soil moisture in rainfed area to model that are accounting for irrigation.

Response: We completely agree with this comment from reviewer 3, which is also proposed by reviewer 2. For the sake of clarification, we will not emphasize the role of irrigation any more. The irrigation-related statements (L135 "Stations are located in areas without irrigation" and L452-453 "The overestimation of RZSM by ERA5 (Fig. 3) could be a signature of irrigation because the in situ RZSM observations do not capture irrigation") will be deleted in the revised manuscript.

And we try to address the confusion mentioned above by reviewer 3.

At L119 and L124 it stated that the study is conducted over a highly irrigated area.

The Huai River Basin is highly irrigated because the mean annual precipitation of 888 mm is less than mean annual evaporation demand (900-1500 mm).

At L135 it is stated that soil moisture sensors are in areas without irrigation.

L135 "Stations are located in areas without irrigation", it means that the soil moisture probes are installed away from crops, which intends to avoid capturing the irrigation signal and obtain

the natural soil moisture states.

Does this mean that some RZSM products are based on model that take into account irrigation and others not?

Traditionally, irrigation is not modelled in the global LSMs (Zohaib and Choi, 2020). Therefore, the model-based RZSM products don't take into account irrigation, and L452-453 is incorrect statement and has been removed. For example, ERA5 didn't model irrigation (Lavers et al., 2022). Therefore, the irrigation process that is not considered in the LSMs is not related to the overestimation of the model-based RZSM products. The role of irrigation will not be mentioned in the revised manuscript.

**Soil map and soil parameters (section 4.3.2 and section 5.2).**
The assessment of the soil properties is valuable. It should be noted, however, that high discrepancies come with the use of different pedotransfer functions (PTF) to derive soil hydraulic parameters (retention curve and hydraulic conductivity). I guess these parameters are used in each RZSM product but estimated with different PTFs. This could also explain part of the uncertainty. This information is missing in the manuscript but should be integrated.

Response: Thank you for the valuable suggestion, we have added the following statements about PTFs in section 5.2.

"The soil hydraulic parameters (SHPs), such as the hydraulic conductivity and matric potential, are crucial parameters to describe the vertical transport of water in the soil column through the Richard's equation employed in the LSMs. Generally speaking, the SHPs are derived from a combination of soil properties (clay, sand, silt fractions and organic content, etc.) with pedotransfer functions (PTFs), which can be constructed by multivariate regression models, nonlinear regression models or artificial neural networks (Harrison et al., 2012). Therefore, different input variables and functional forms of the continuous PTFs are used to derive SHPs in the LSMs. The Richard's equation relying on the SHPs shows great uncertainty in the simulated soil moisture. For example, the HWSD soil properties used in SMAP L4 are more consistent with the reference dataset than FAO soil properties used in MERRA-2 by revising the underestimated sand content and the overestimated clay content in FAO. In addition, SMAP L4 adopts PTFs from Wösten et al. (2001) which take into account the organic carbon affecting soil hydraulic and thermal properties. MERRA-2 adopts PTFs adapted from Cosby et al. (1984) without considering organic carbon (De Lannoy et al., 2014). The revised soil parameters and new PTFs employed in SMAP L4 yield smaller shape parameter of water retention curve and result in less water retention than in MERRA-2, and increase the hydraulic conductivity. Thus, SMAP L4 has the smaller soil moisture estimates and less RZSM bias against in situ measurements than MERRA-2, which is consistent with the result of this study. Therefore, the soil properties and PTFs could also explain part of the uncertainty.

**Spatial aggregation and comparison at each site should be clarified and improved**
As far as I have understood (L267), the comparison is conducted between the spatial average of the 58 in situ observations. It is not well reported how much is the spatial extent of the 58 stations but looking at figure 1 I guess the station covers an area of around 300 x 200 km2. I then deduce that this spatial average is compared to the spatial average of the gridded products

(i.e., each product has different resolutions, but more than one cell of the gridded products covers the area of the 58 stations). So first of all it should be better explain how many cells have also been aggregated for each product. Only later in the results section (L317) I discovered that a comparison has also been performed without aggregating spatially. So, first of all, this information should be provided also before in the methods. Moreover, I would also considering moving some plots that are now in supplement to the main manuscript to strengthen the analysis. Anyway, I'm confused by the fact that the comparison is performed at each station. Does this mean that you have always one station against one cell of the gridded products? Please clarify.

Response: We feel sorry to make you confused. And we will try to resolve your confusion.

1. The spatial extent of the 58 stations is approximately $310 \times 330$ km$^2$.

2. For different RZSM products, different numbers of grids are aggregated. For example, CLDAS: 58, GLDAS_CLSM: 50, GLDAS_NOAH: 50, ERA5: 48, MERRA-2: 50, NCEP CFSv2: 55, SMAP L4: 58, SMOS L4: 51.

3. In this manuscript, Fig.2 and 3 represents that the comparison of the RZSM time series averaged over all in situ stations with the RZSM time series averaged over all model grids where the stations are located. L317 represents the single point-grid validation, the measurements at each station are compared directly with the grid values where the station is located.

The following text (section 3.3 RZSM products aggregation and validation strategies) will be added in chapter 3 Methods.

"In terms of the temporal resolution, except for the RZSM products (e.g., GLDAS_CLSM, SMOS L4) provided at daily time intervals, the other sub-daily RZSM datasets (hourly/3-hourly/6-hourly time steps, shown in Table 1) are aggregated to daily average values to match the daily sampling frequency of the in situ observations. In terms of spatial resolution, we didn't change the spatial resolution of any RZSM products and used the RZSM time series for each grid where the in situ stations are located. Two validation strategies were used in the study. The first is to compare the RZSM time series averaged over all in situ stations with the RZSM time series averaged over all model grids where the in situ stations are located (Fig.2 and 3 shown in this study). The second one is the point-grid validation, the RZSM measurements at each in situ station are compared directly with the RZSM values for the grid where the in situ station is located, if there is more than one in situ station within a grid, the RZSM measurements at each station are compared to the grid values separately. The point-grid validation is provided in the supplement (Fig. S1 and S2)."

The RZSM validation uses two methods, (1) the spatial average validation is shown in this study (the figures are presented as time series, such as the Fig. 3 and Fig. 10), (2) the point-grid validation at each station is provided in the supplement (the figures are presented as a histogram and error bar, such as the Fig. 7 and Fig. 9). The validation of precipitation and soil properties is performed at one station against one cell of the gridded products. We have added detailed explanation in the legends of each figure when it comes to validation.

**Section 2.4: description of the eight RZSM products**

The description of the eight products should be improved in several parts. The information provided for each product is not always consistent. Some products are better described and with more details than others. E.g., for MERRA-2, the description focuses on the precipitation. Instead, ERA5 does not have any information about. NCEP CFSv2 description is very short. Who provided that? What are the main properties? Some characteristics provided should also be put more in relation to the focus of the paper. E.g., for ERA5, the data assimilation system is described in detail. Is that relevant for the purpose of the paper. If yes, it should be clarified. Overall, the main differences between the products relevant for the present study (e.g., soil map, precipitation, land use, irrigation etc.) should be better highlighted. Table 1 should be extended accordingly.

Response: We have revised the whole section 2.4 thoroughly and Table 1, and provide more focused and consistent description of the eight RZSM products in the revised manuscript.

Please note that some relevant information are discussed only later but in my opinion they should be moved to the method section. This would help understanding and strengthening the discussion of the results. E.g., L410 The soil properties data used in the eight RZSM products were all derived from the FAO/UNESCO soil map of World except for CLDAS, which used the soil data developed by Shangguan et al. (2013), and SMAP L4, which used the HWSD soil properties over China. L426. Global precipitation and air temperature forcing data are used in the production of all RZSM products except for SMOS L4. L452. The overestimation of RZSM by ERA5 (Fig. 3) could be a signature of irrigation because the in situ RZSM observations do not capture irrigation.

Response: L452 will be removed. The following text (section 3.3 RZSM products aggregation and validation strategies) will be added in chapter 3 Methods.

"In terms of the temporal resolution, except for the RZSM products (e.g., GLDAS_CLSM, SMOS L4) provided at daily time intervals, the other sub-daily RZSM datasets (hourly/3-hourly/6-hourly time steps, shown in Table 1) are aggregated to daily average values to match the daily sampling frequency of the in situ observations. In terms of spatial resolution, we didn't change the spatial resolution of any RZSM products and used the RZSM time series for each grid where the in situ stations are located. Two validation strategies were used in the study. The first is to compare the RZSM time series averaged over all in situ stations with the RZSM time series averaged over all model grids where the in situ stations are located (Fig.2 and 3 shown in this study). The second one is the point-grid validation, the RZSM measurements at each in situ station are compared directly with the RZSM values for the grid where the in situ station is located, if there is more than one in situ station within a grid, the RZSM measurements at each station are compared to the grid values separately. The point-grid validation is provided in the supplement (Fig. S1 and S2).

The global precipitation and air temperature forcing data are used in the production of model-based RZSM products except for SMOS L4, which are validated against the China daily gridded ground precipitation and air temperature dataset V2.0 described in section 2.2. The soil properties data used in the eight RZSM products were all derived from the FAO/UNESCO soil map of World except for CLDAS, which used the soil data developed by Shangguan et al. (2013), and SMAP L4, which used the HWSD V1.2 soil properties over China. The China soil

dataset developed by Shangguan et al. (2013) is used as a reference to evaluate the accuracy of FAO/UNESCO and HWSD V1.2 soil properties (clay and sand content, organic carbon content and bulk density)."

Figure quality

Figures are not always readable and meaningful. I suggest putting some more effort into evaluating how to present the results. E.g.,

Response: The figures have been revised thoroughly.

Fig 1. could also shows the pixel size of the products. This would help understanding spatial extend and intercomparisons.

Response: Fig. 1 has been revised and shows the grid size of 0.25 degree covering the in situ stations.

[Figure]

Fig. 1 Overview of the study area and distribution and land cover of in situ soil moisture stations (green pentagon). The squares in Fig.1b and c represent 0.25° grid.

Fig. 4 is not readable at all. Plots and texts are too small.

Response: Fig. 4 has been revised with improved plots and texts.

[Figure]

Fig. 4 Comparison of different RZSM products (volumetric water content, m3 m-3) with each other. The scatterplots and their corresponding statistics are located on opposite sides of each other, that is, the scatterplot of the data pair SMOS L4-ERA5 is in the top left-hand corner, while the respective statistical values are found in the bottom right-hand corner. Darker regions show a higher density of data point and the blue line in each subplot represents the fitted trend for the data points.

Fig. 5 can be improved by having only 8 histograms of the RZSM products and overlapping each histogram with the observation's histogram. This could help to visualize the differences.

Response: Fig. 5 has been revised.

[Figure]

Fig. 5 The histograms of normalised RZSM products (dashed and red lines) and in situ observations (black and solid lines).

Fig.6. The plots are hardly comparable. It could be evaluated to present one plot with all the cumulative precipitation, or histogram of the precipitation etc.

Response: We have presented one new plot with all the cumulative precipitation. Since most of the daily precipitation ranges from 1 to 10 mm day$^{-1}$, the histogram or probability density function of different precipitation datasets can't be well distinguished from each other. Therefore, it is not included in the plot.

[Figure]

Fig. 6 Comparison of cumulative precipitation events and cumulative precipitation amounts between model-derived precipitation and in situ precipitation observations averaged over all in situ stations.

Fig.7. It is not clear to me what is actually presented. This is in line to the general comment above about aggregation. Are you comparing spatial averaged precipitation? What does standard deviation refer to? If you compare each rain gauge to pixel wise, how have you aggregated?

Response: For Fig. 7, we compare each rain gauge to pixel wise where the rain gauge is located. The six statistical metrics in Fig.7 are calculated at each station, so there are 58 data points for each statistical metric. The histogram represents the median of 58 data points for each statistical metrics between modelled precipitation and observations. The stand deviation represents the variability in the statistical metrics. We will add more detailed descriptions to the legend of Fig. 7.

**Take-home-message**

I'm expecting to read the overall take-home-message. After performing this intercomparison, what can you conclude? Could we trust this products? Where and which conditions? How would you suggest further improving, studies etc.? This is missing throughout the manuscript but should be understandable from the abstract and more extended at the conclusions.

Response: We have reworded the abstract to provide more focused and important information (the following text) concluded in the study, especially the take-home-message. Furthermore, we have extended the key conclusions and provide some insights into how to improve the accuracy of modelled soil moisture.

"Overall, the GLDAS_CLSM outperforms the other RZSM products and shows the potential for drought monitoring and flood forecast in the Huaibei Plain. Model-based RZSM products could capture the temporal dynamics of in situ observations and response to precipitation events well. While SMOS L4 shows a much lower correlation with in situ observations than model-based RZSM products, which could be caused by the fact that the precipitation is not used in

the production of SMOS L4. So, the SMOS L4 RZSM has a weaker response to precipitation and low correlation with in situ observations. The model-based RZSM products generally perform better in the wet season than in the dry season due to the enhanced ability to capture of the temporal dynamics of in situ observations in the wet season and the inertia of remaining high soil moisture values even in the dry season. While SMOS L4 performs better in the dry season than in the wet season, the ground microwave radiation signal is more attenuated in the wet season due to a substantial increase in water vapor absorption and scattering than in the dry season, which is used to retrieve SMOS L3 SSM and is propagated to SMOS L4 RZSM. The comparison of frequency distribution between eight RZSM products and in situ observations indicates that MERRA-2, GLDAS_CLSM and SMAP L4 show better agreement with the in situ observations than the other RZSM products, although they slightly overestimate the frequency of wet soil moisture. However, they all don't capture the frequency peak and underestimate the frequency peak of normalized soil moisture ranging from 0.2 to 0.4. In contrast, the other RZSM products show significant overestimation of frequency of wet soil moisture, underestimation of dry soil moisture and of peak frequency. Therefore, the Richard's equation used to simulate the water content in different soil layers in LSMs should focus on producing less wet soil moisture and more dry soil moisture to obtain more accurate frequency distribution of modelled soil moisture.

**SPECIFIC COMMENTS IN ORDER OF APPEARANCE (L = LINE NUMBER)**

L13. I would not use the term "direct validation" but "assessment".

Response: Correction done.

L29. Th abstract focused on describing the actual results. This is fine but I would also expect at the end to read the take-home-message. E.g., what do we learn by this study? Can we trust, use, apply RZSM products? Where? In which conditions? What in our view and based on this study would further suggest to improve the performances?

Response: The abstract has been replaced by the following text.

"Root zone soil moisture (RZSM) is critical for water resource management, drought monitoring and sub-seasonal flood climate prediction. While RZSM is not directly observable from space, several RZSM products are available and widely used at global and continental scales. This study conducts a comprehensive and quantitative evaluation of eight RZSM products using observations from 58 *in situ* soil moisture stations over the Huai River Basin (HRB) in China. Attention is drawn to the potential factors that contribute to the uncertainties of model-based RZSM, including the errors in atmospheric forcing, vegetation parameterizations, soil properties, and spatial scale mismatch. The results show that the Global Land Data Assimilation System Catchment Land Surface Model (GLDAS_CLSM) outperforms the other RZSM products with the highest correlation coefficient ($R = 0.69$) and the lowest unbiased root mean square error (ubRMSE = 0.018 $m^3 m^{-3}$), and shows the potential for drought monitoring and flood forecast in Huaibei Plain. While SMOS Level 4 (L4) RZSM shows a much lower correlation with in situ observations than model-based RZSM products forced by precipitation, this could be due to the fact that SMOS L4 does not contain precipitation information and has a weaker response to precipitation. The model-based RZSM products generally perform better in the wet season than in the dry season due to the enhanced

ability to capture of the temporal dynamics of in situ observations in the wet season and the inertia of remaining high soil moisture values even in the dry season. While SMOS L4 performs better in the dry season than in the wet season, because the ground microwave radiation signal is more attenuated in the wet season due to a substantial increase in water vapor absorption and scattering than in the dry season, which is used to retrieve SMOS Level 3 (L3) SSM and is propagated to SMOS L4 RZSM. The underestimation of Surface Soil Moisture (SSM) in SMOS Level 3 (L3), caused by underestimated physical surface temperature and overestimated ERA interim soil moisture, may trigger the underestimation of RZSM in SMOS L4. The seven model-based RZSM products show an overestimation of in situ observations, which could be associated with the overestimation of precipitation amounts, the frequency of precipitation events (drizzle effects) and the underestimation of air temperature and the underestimated ratio of transpiration to the total terrestrial evapotranspiration. In addition, the biased soil properties (organic carbon, clay and sand fractions) and flawed vegetation parameterizations (e.g., canopy, root tissue and soil evaporation and transpiration model representation) affect the hydrothermal transport processes represented in different LSMs and lead to inaccurate soil moisture simulation. The scale mismatch between point and footprint also introduces representative errors. The comparison of frequency of normalized soil moisture between RZSM products and in situ observations indicates that the LSMs should focus on reducing the frequency of wet soil moisture, increasing the frequency of dry soil moisture and the ability to capture the frequency peak of soil moisture."

L108-109. Are these two lines really needed here? I would integrate this information later when you speak about comparison. E.g., L133 for the definition of RZSM.

Response: L108-109 was deleted. L133 was replaced by "Since the study aims to evaluate the accuracy of eight RZSM products (0-100 cm) which are summarized in Table 1, the in situ soil moisture measurements at the four depths are depth-weighted averaged to obtain the 0-100 cm soil moisture data."

L128. Table S1 shows the results of the assessment, and it does not provide additional information about the in situ stations. I would remove this cross reference here and rather add Figure 1 where the locations of the in situ stations are shown.

Response: "Table S1" has been replaced by "Fig. 1".

L263. I would extend a bit on the meaning and interpretation for PD, FAR and CSI.

Response: The following text has been added in section "3.1 Statistical metrics".

"POD is the proportion of real precipitation events simulated by AGCM relative to the actual precipitation events, reflecting the ability of AGCM to detect precipitation. FAR is the fraction of unreal precipitation events out of the total precipitation events simulated by AGCM. CSI is a more balanced score that combines the characteristics of false alarms and missed events, representing the probability of successful simulation of AGCM precipitation."

L280. I was expecting to read more about the description of the equation after L280. Any text missing?

Response: We will add the following description after L280.

"where $\theta_{RZSM}$ denotes 0-100 cm RZSM (m³ m⁻³), $\theta_{0-10}$ , $\theta_{10-40cm}$ and $\theta_{40-100cm}$ denote the soil moisture estimates at 0-10 cm, 10-40 cm and 40-100 cm, respectively."

L293. I think here is a good place to cite table 3 as well.

Response: L293-294 will be replaced by "Figure 2 shows scatterplots of RZSM products against the in situ measurements averaged across all in situ stations over the HRB, from 1 April 2015 to 31 March 2020. The statistical metrics are shown in Table 3."

L295. It is stated that SMOS-L4 underestimates and the other overestimated the observations. By looking at figure 2, I see the opposite. Please double check what you are plotting.

Response: Thank you for bringing up the mistake. We reversed the labels for the x and y axes of Figure 2.

[Figure]

Fig. 2 Scatterplots of RZSM products vs. in situ RZSM observations averaged across all in situ stations from 1 April 2015 to 31 March 31 2020. Scores are given in Table 3. Darker regions show a higher density of data point and the blue line in each subplot represents the fitted trend for the data points.

L306. Figure 3 shows spatial average of in situ soil moisture and its spatial variability. As far as I have understood (see general comment above on the spatial aggregation), the spatial average of the RZSM products is shown but, if possible, it could be interesting to show here also the spatial variability of the RZSM products.

Response: L306 was a typo and replaced by "Figure 3 shows the time series of observation- and model-based RZSM averaged over all in situ stations and the grids where the in situ stations are located". We add the following figure in the supplement to display the spatial distribution pattern of eight RZSM products averaged from 1 April 2015 to 31 March 2020.

[Figure]

Fig. S4 Spatial distribution pattern of eight RZSM products and in situ observations interpolated using Inverse Distance Weighting (IDW) averaged over from 1 April 2015 to 31 March 2020.

L317. Description of the comparison at each site should be reported as well in the method (section 3.1) See also general comment above on the spatial aggregation.

Response: See the response in **Spatial aggregation and comparison at each site should be clarified and improved (Page 27-28).**

L319. The method to calculate the anomalies is reported only in the supplement. I think should be moved to the method (section 3.1)

Response: The anomalies metrics are not very important and relevant to the results and discussion in the study, the related statements are removed in the revised manuscript.

L585. After summarizing the mani conclusions, I suggest summarizing the outlook of the study. See also general comments above.

Response: The conclusions have been revised.

(1) GLDAS_CLSM outperformed the other RZSM products over the HRB in terms of R, ubRMSE and mean bias, followed by MERRA-2, CLDAS, SMAP, ERA5, NCEP CFSv2, and GLDAS_NOAH. The SMOS L4 product presented the worst performance due to the fact that SMOS L4 does not contain precipitation information and has a weaker response to precipitation.

Seven model-based RZSM products overestimated the in situ observations with median bias values ranging from 0.033 $m^3$ $m^{-3}$ (SMAP L4) to 0.116 $m^3$ $m^{-3}$ (CLDAS). While SMOS L4 underestimated the RZSM with a median bias value of -0.050 $m^3$ $m^{-3}$.

(2) The intercomparison of RZSM products shows that the correlation coefficient R between any two of the seven model-based RZSM products varied from 0.68 (ERA5 vs. CLDAS) to 0.95 (SMAP L4 vs. MERRA-2). In contrast, SMOS L4 presented lower correlation with the other seven RZSM products with R ranging from 0.30 (MERRA-2) to 0.41 (GLDAS_NOAH) and with a negative bias ranging from -0.165 $m^3$ $m^{-3}$ (SMOS L4 minus ERA5) to -0.077 $m^3$ $m^{-3}$ (SMOS L4 minus SMAP L4). The comparison of the frequency distribution between eight RZSM products and in situ observations indicates that MERRA-2, GLDAS_CLSM and SMAP L4 are in better agreement with the in situ observations than the other RZSM products. All RZSM products overestimate the frequency of wet soil moisture and underestimate the frequency of dry soil moisture. Besides, the frequency peaks of the RZSM products show an obvious offset towards wet soil moisture and are underestimated compared to the in situ observations. Therefore, the Richard's equation used to simulate the water content in different soil layers in LSMs should focus on producing less wet soil moisture and more dry soil moisture.

(3) Except for CLDAS, the overestimation of in situ soil moisture observations by the model-based RZSM products could be associated with the overestimation of precipitation amounts, the frequency of precipitation events (excessive number of occurrences of drizzle events). The air temperature datasets used to drive the LSMs have a cold bias, which tends to reduce evapotranspiration and result in more soil moisture residuals. In addition, the underestimated ratio of transpiration to the total terrestrial evapotranspiration existing in most earth system models consumes less water in the root zone for transpiration and large RZSM. The underestimation of the SMOS L4 RZSM may be related to the underestimation of the SMOS L3 SSM.

(4) The model-based RZSM products generally perform better in the wet season than in the dry season due to the enhanced ability to capture of the temporal dynamics of in situ observations in the wet season and the inertia of remaining high soil moisture values even in the dry season. While SMOS L4 performs better in the dry season than in the wet season, because the ground microwave radiation signal is more attenuated in the wet season due to a substantial increase in water vapor absorption and scattering than in the dry season, which is used to retrieve SMOS L3 SSM and is propagated to SMOS L4 RZSM.

(5) The utilization of the HWSD soil property dataset instead of the FAO/UNESCO World Soil Map will contribute to improve the simulation of the hydrothermal transport processes represented in LSMs and thus to an improved land surface analysis.

(6) Spatial-average validation could reduce the spatial noise of in situ soil moisture

measured at different locations and improve the representativeness of soil moisture observations to model-based grid values.

Supplement. I think the supplement should have a title with the name of authors as well.

Response: According to the submission requirement of HESS journal: Supplements will receive a title page added during the publication process including title ("Supplement of"), authors, and the correspondence email. Therefore, please avoid providing this information in the supplement.

---

## Referee Report (RR1)

Review of the manuscript egusphere-2023-1597 Evaluation of root-zone soil moisture products over the Huai river basin by Liu et al.

**SUMMARY**

I believe that the Authors prepared a comprehensive rebuttal to the comments of the Reviewers and the manuscript has been improved accordingly. Still, I found some issues that merit in my opinion further attention. These issues are described in detail below followed by more specific edits. As soon as the Authors could also improve these parts, I think the manuscript should reach a good quality for possible publication.

**MAJOR ISSUES**

**Irrigation**: at L141 it is stated that the main cover types in the HRB are rainfed croplands, followed by irrigated croplands. Later (L143) it is stated that 76% of the cultivated area is irrigated. Finaly (L148) it is underlined that heavy irrigation in the HRB can explain the extra water available for evaporation. Based on these statements, I'm still confused by the role of irrigation and its effect on the present intercomparison. I understand that ground soil moisture observations are in rainfed areas. Thus, some LSM that do not consider irrigation can be compared accordingly. However, recently the use of remote sensing products to estimate irrigation has been widely promoted. So, a comparison of the ground measurements to remote sensing products or soil moisture products based on the assimilation of remote sensing into LSM is in my opinion misleading as soon as the area is irrigated and the remote sensing data capture to some extent this signal. I encourage the Authors to further clarify.

**Temporal resolution**: soil moisture observations are collected at 8:00 am (L154). So, the aggregation to daily average soil moisture products is not consistent (L351 - 352). It would have been better to address the temporal mismatch by, e.g., selecting the consistent hour of the soil moisture products when possible or resampling the ground soil moisture time series. Please at least clarify this issue in the methods and during the discussion.

**Homogenous area**. The bold statement in the rebuttal (e.g., response to comment 2 of Reviewer #1) about having a homogenous area is arguable also in the light of the data and analysis presented and discussed. E.g., fig 3 shows with gray shaded area the standard deviation of the soil moisture observation which is not negligible in my opinion. Fig 9 e fig 10 shows strong variability of soil properties. Fig S2 and S3 show strong variability of the performance when point-to-grid comparison is performed. In Section 5.5 it is discussed that (L669) results can be interpreted considering the high spatial variability resulting from different characteristics of the underlying surface and meteorological forcing. I encourage the Authors to further clarify. Please also consider moving the results (figure S2) and the discussion (section 5.5) into the results section.

**MINOR ISSUES**

L18-19. The manuscript does not test the capability of soil moisture products to support drought monitoring and flood forecasting. Thus, the statement "and shows the potential for drought monitoring and flood forecast" is not supported. I would remove it.

L24-27. The sentence is not clear. Please check English grammar.

L165. Average coverage is 38%. Do you mean that for 38% of the area you have at least 1 station per cell? Please clarify.

L332 – 333. Since you have added the section 3.3, I think you can now remove the sentence starting with "The time series…metrics".

Figures. Some figures (fig. 4-5-6-10-11) are cut in the lower part and labels are not always visible. Please provide new figures.

Fig.10: Please be more precise in the caption legend. what do you mean by "different stations"?, all?

L547. The fact that soil properties are time invariant depends on the spatial and temporal scale of the study. I agree that porosity is considered time-invariant in these LSM but I would consider other examples, e.g., texture.

L748. The 5th statement is not supported by the present study, i.e., you do not have results that support the fact that HWSD will contribute to improve the simulation. It is a hypothesis that should be tested but is also out of scope of the present study. Please reformulate.

---

## Author Response (AR2)

**Dear Editor and Reviewers,**

We would like to thank the Editor and three Reviewers for their efforts in handling the manuscript and for their valuable comments to improve the manuscript. We have revised our manuscript thoroughly according to the comments and provide a point-by-point response to the comments from three Reviewers below. The original comments from three Reviewers are in black font, and our responses are in blue font.

On behalf of all co-authors,
En Liu

**Response to Reviewer #1's comments on the manuscript egusphere-2023-1597**

**RC1: 'Comment on egusphere-2023-1597', Anonymous Referee #1, 02 Feb 2024**

While the information contained in the manuscript is quantitative, I ended up questioning the potential contribution of this paper to the readers. I have tried to address why I think While the revised manuscript has been improved in the sense that the objective can be better interpreted, I found the whole manuscript is still relatively complex to understand mainly due to 1) many awkward/incomplete sentences, 2) figures/tables are not well organized and indicated (e.g., labels, units, captions). I am afraid the readers might get confused due to missing information in the figures. Please check my specific comments below:

Response: The authors thank the Reviewer #1 for her/his constructive and insightful comments that help us improve the quality of the manuscript. The original comments from Reviewer #1 are in black font, and our responses are in blue font.

Title: Should the title include 'remote sensing RZSM'?

Response: Seven RZSM products evaluated in the study are derived from land surface models, they are numerical simulation products. And the SMOS L4 RZSM product is derived from the SMOS L3 3-day SSM product using a statistical exponential filter. These RZSM products are not closely related to remote sensing. Therefore, we did not use the term "remote sensing RZSM".

Figure (all): Please increase labels and units and add more explanation to the captions to help understand more correctly.

Response: We have revised all figures and captions.

Table 1: It is nice to have this organized table for the spatiotemporal resolution of each product. Can you add a plot showing how in-situ stations overlap with each RZSM product?

Response: Figure 3 shows how the time series of in-situ observations overlap with eight RZSM products.

Line 354: Didn't > did not (or change the sentence)

Response: Correction done.

Line 446-448: Richards' equation is just an equation; what would be the way for Richards's equation to produce different soil moisture dynamics? Do you mean retention curve? or changing boundary conditions?

Response: The LSMs generally produce higher soil moisture content than in situ observations due to the overestimated precipitation input. In addition, the inertia of remaining high soil moisture content after precipitation events could be weakened by modifying the initial soil moisture values, e.g. data assimilation of soil moisture observations or reducing the model default soil moisture values. Therefore, the Richard's equation could produce different soil moisture dynamics by modifying the soil water retention curve or changing the initial and boundary conditions.

Line 499: Are these soil properties data compared with observations? Is it overestimation (compared to soil properties in-situ) or just higher estimates?

Response: These soil properties data are compared with reference dataset developed by Shangguan et al. (2013), which integrates the physical and chemical properties of 8979 soil profiles along with the soil map of China. It shows overestimation compared to the soil property reference dataset.

Line 532: I do not expect significant spatial heterogeneity for precipitation in a watershed. And what do you mean by temporal heterogeneity? Accuracy for precipitation forcing has always been a major challenge.

Response: We agree, "spatiotemporal heterogeneity" was replaced by "spatial heterogeneity".

Line 536: Does not make any sense! "overestimation of in situ observations by LSM-based RZSM products" ???

Response: L536 was removed.

Fig. 10: So what is this figure for? Is this observation and what sites? Why didn't you plot this within Fig. 9?

Response: Fig 10 shows the soil stratification in Huai river basin. The soil stratification affects the water transfer from the surface layer to the root zone layer in land surface models.

Line 618-621: The sentence needs to be restructured.

Response: L618-621 was deleted.

Line 622: Why is line 618-621 the specific reason for the difficulty of vegetation parameterization? That is a very general theory and not appropriate to be selected as a specific reason as it does not stem from the model structure.

Response: L618-622 was deleted.

Line 666-670: The whole sentence needs to be rephrased.

Response: L666-670 is replaced by:

"The statistical scores for spatial-average validation are generally better than that for point-grid validation, which are shown in Tables 3 and S1, respectively. For the point-grid validation, the spatial representativeness of *in situ* soil moisture observations at the grid scale is insufficient due to the heterogeneity of the underlying surface and precipitation forcing."

While I didn't compile the whole list of awkward/unclear sentences, I do express my concern about the clarity of this paper due to the components that I tried to address here. I do encourage authors to review and revise the paper thoroughly to make sure the paper conveys what they want to convey.

Response: We have revised the manuscript thoroughly and make it more clear and concise.

**Response to Reviewer #2's comments on the manuscript egusphere-2023-1597**

**RC2: 'Comment on egusphere-2023-1597', Anonymous Referee #2, 12 Jan 2024**

Thanks for the considerable revisions made. I still have two small comments:

Response: The authors thank the Reviewer #2 for her/his constructive and insightful comments that help us improve the quality of the manuscript. The original comments from Reviewer #2 are in black font, and our responses are in blue font.

- L104: 2.84m should be 2.89m

Response: Correction done.

- Both the conclusion and the abstract would benefit from a sentence on the outlook/implications of this research as now it is mostly summarizing the results. Here the previous comment 'Lastly, how easily can we extrapolate these results to other regions?' could help.

Response: The following text has been added in the Abstract (L39-42):

"The study provides some insights into how to improve the ability of land surface models to simulate the land surface states and fluxes by taking into account the issues mentioned above. Finally, these results can be extrapolated to other regions located in the similar climate zone, as they share the similar precipitation patterns that dominate the terrestrial water cycle."

The following text has been added in the Conclusion (L720-723):

"The study could provide some insights into how to improve the ability of land surface models to perform the land surface analysis by addressing the above issues. Furthermore, these results can be extended to other regions to improve the numerical simulation capability of land surface models at global scale."

**Response to Reviewer #3's comments on the manuscript egusphere-2023-1597**

**RC3: 'Comment on egusphere-2023-1597', Anonymous Referee #3, 31 Jan 2024**

Review of the manuscript egusphere-2023-1597 Evaluation of root-zone soil moisture products over the Huai river basin by Liu et al.

SUMMARY

I believe that the Authors prepared a comprehensive rebuttal to the comments of the Reviewers and the manuscript has been improved accordingly. Still, I found some issues that merit in my opinion further attention. These issues are described in detail below followed by more specific edits. As soon as the Authors could also improve these parts, I think the manuscript should reach a good quality for possible publication.

The authors thank the Reviewer #3 for her/his constructive and insightful comments that help us improve the quality of the manuscript. The original comments from Reviewer #3 are in black font, and our responses are in blue font.

MAJOR ISSUES

Irrigation: at L141 it is stated that the main cover types in the HRB are rainfed croplands, followed by irrigated croplands. Later (L143) it is stated that 76% of the cultivated area is irrigated. Finally (L148) it is underlined that heavy irrigation in the HRB can explain the extra water available for evaporation. Based on these statements, I'm still confused by the role of irrigation and its effect on the present intercomparison. I understand that ground soil moisture observations are in rainfed areas. Thus, some LSM that do not consider irrigation can be compared accordingly. However, recently the use of remote sensing products to estimate irrigation has been widely promoted. So, a comparison of the ground measurements to remote sensing products or soil moisture products based on the assimilation of remote sensing into LSM is in my opinion misleading as soon as the area is irrigated and the remote sensing data capture to some extent this signal. I encourage the Authors to further clarify.

Response: Current land surface models (LSMs) traditionally don't take into account irrigation practices (Romaguera et al., 2012; Lievens et al., 2017; Brocca et al., 2018; Abolafia-Rosenzweig et al., 2019). Indeed, recent studies have shown that the remote sensing soil moisture products (e.g. SMAP, Sentinel-1, ASCAT and SMOS) can capture the irrigation signal to some extent. However, these remotely sensed SM retrievals alone are insufficient to assess spatiotemporally continuous estimates of irrigation and its effects on the water and energy cycles (Abolafia-Rosenzweig et al., 2019). The assimilation of remote sensing soil moisture into LSMs has been widely conducted and shows an overall superior performance than open loop in terms of soil moisture simulations. However, the assimilation of remote sensing data mainly improves the surface soil moisture simulations, and the impact on RZSM is less pronounced and mostly neutral (Lievens et al., 2017; Reichle et al., 2017). Besides, the satellite observations with coarse spatial resolution at kilometer level (e.g. SMAP and SMOS)

struggle to resolve the local irrigation practices at field scale and can't present accurate irrigation signal. For example, Abolafia-Rosenzweig et al. (2019) indicated that Sentinel-1 soil moisture observations (10 m spatial resolution) performs better than SMAP observations in resolving irrigation practices. Escorihuela andQuintana-Seguí (2016) compared SMOS L2 soil moisture and downscaled SMOScat at 1 km scale with SURFEX LSM simulations, respectively, the high resolution SMOScat product shows a lower correlation with modelled soil moisture compared to raw SMOS L2 soil moisture at a small heavily irrigated region. Among the eight RZSM products used in the study, SMAP L4 RZSM and SMOS L4 RZSM may contain irrigation signal, as SMAP L4 RZSM assimilates L1C brightness temperature which implicitly includes an irrigation signal, and SMOS L4 RZSM is derived from SMOS L3 surface soil moisture which can detect irrigation signal but with low skill. Though SMOS L4 RZSM may contain irrigation signal, it still underestimates the *in situ* soil moisture observations. Therefore, we think the irrigation signal captured by the remote sensing soil moisture has little impact on the intercomparison and the effect of irrigation signal contained in surface soil moisture on RZSM can be ignored.

Temporal resolution: soil moisture observations are collected at 8:00 am (L154). So, the aggregation to daily average soil moisture products is not consistent (L351 - 352). It would have been better to address the temporal mismatch by, e.g., selecting the consistent hour of the soil moisture products when possible or resampling the ground soil moisture time series. Please at least clarify this issue in the methods and during the discussion.

Response: We will clarify this issue in the methods and during the discussion.

In the methods (section 3.3), L353-358 are replaced by "In terms of the temporal resolution, GLDAS_CLSM and SMOS L4 products provide RZSM data at daily time intervals. NCEP CFSv2 and GLDAS_NOAH products provide RZSM data at 3-hourly and 6-hourly time interval, respectively, which don't have consistent hour of soil moisture data with *in situ* observations only available at 08:00 AM. To keep consistent, thus the other sub-daily RZSM datasets (hourly/3-hourly/6-hourly time steps, shown in Table 1) are aggregated to daily average values to match the daily sampling frequency of the *in situ* observations."

In the discussion (section 5.5), the following text are added in L689-692.

"Finally, the temporal mismatch between model-based RZSM values which are aggregated to daily average values and in situ observations available at 08:00 AM could also induce partial bias, but this type of bias is generally small due to the low variability of soil moisture during the day."

Homogenous area. The bold statement in the rebuttal (e.g., response to comment 2 of Reviewer #1) about having a homogenous area is arguable also in the light of the data and analysis presented and discussed. E.g., fig 3 shows with gray shaded area the standard deviation of the soil moisture observation which is not negligible in my opinion. Fig 9 e fig 10 shows strong variability of soil properties. Fig S2 and S3 show strong variability of the performance when point-to-grid comparison is performed. In Section 5.5 it is discussed that

(L669) results can be interpreted considering the high spatial variability resulting from different characteristics of the underlying surface and meteorological forcing. I encourage the Authors to further clarify. Please also consider moving the results (figure S2) and the discussion (section 5.5) into the results section.

Response: We agree. The soil moisture and soil properties show strong variability across different *in situ* stations. We average different datasets (e.g. soil moisture, soil properties) to weaken the spatial heterogeneity across *in situ* stations in this study. And investigate how different factors (forcing data, model parameters, soil properties, etc.) affect the soil moisture simulation from the perspective of physical process at regional scale rather than point scale. More importantly, the focus of this study is to investigate the sources of error of the different RZSM products, which could provide some insights about how to improve the ability of land surface models to simulate the land surface states and fluxes. Therefore, the heterogeneity across different stations is not the focus of the study.

Figure S2 and section 5.5 have been moved into the results section.

MINOR ISSUES

L18-19. The manuscript does not test the capability of soil moisture products to support drought monitoring and flood forecasting. Thus, the statement "and shows the potential for drought monitoring and flood forecast" is not supported. I would remove it.

Response: Correction done.

L24-27. The sentence is not clear. Please check English grammar.

Response: L24-27 is replaced by

"The ground microwave radiation signal captured by SMOS Level 3 (L3) SSM is more attenuated in the wet season due to a substantial increase in water vapor absorption and scattering than in the dry season, which is propagated to SMOS L4 RZSM. So SMOS L4 RZSM performs better in the dry season than in the wet season."

L165. Average coverage is 38%. Do you mean that for 38% of the area you have at least 1 station per cell? Please clarify.

Response: The number of meteorological stations divide the number of grid cells is 0.38. It means per cell has 0.38 station.

L332 – 333. Since you have added the section 3.3, I think you can now remove the sentence starting with "The time series…metrics".

Response: Correction done.

Figures. Some figures (fig. 4-5-6-10-11) are cut in the lower part and labels are not always visible. Please provide new figures.

Response: We have provided new figures.

Fig.10: Please be more precise in the caption legend. what do you mean by "different stations"?, all? L547. The fact that soil properties are time invariant depends on the spatial and temporal scale of the study. I agree that porosity is considered time-invariant in these LSM but I would consider other examples, e.g., texture.

Response: The caption legend of Fig. 10 is replaced by

"Fig. 11 Boxplot of soil properties for three soil layers at all *in situ* stations (Layer1 (0-16.6 cm): plough layer; Layer 2 (16.6-49.3 cm): black soil layer; Layer3 (49.3-138.3 cm): lime concretion layer)." The soil texture (clay and sand fractions) is shown in Fig 11a and b.

L748. The 5 th statement is not supported by the present study, i.e., you do not have results that support the fact that HWSD will contribute to improve the simulation. It is a hypothesis that should be tested but is also out of scope of the present study. Please reformulate.

Response: We have deleted the 5th statement in the conclusion.

Reference

Abolafia-Rosenzweig, R., Livneh, B., Small, E. E. and Kumar, S. V.: Soil Moisture Data Assimilation to Estimate Irrigation Water Use, J Adv Model Earth Syst, 11, 3670-3690, https://doi.org/10.1029/2019MS001797, 2019.

Brocca, L., Tarpanelli, A., Filippucci, P., Dorigo, W., Zaussinger, F., Gruber, A. and Fernández-Prieto, D.: How much water is used for irrigation? A new approach exploiting coarse resolution satellite soil moisture products, International Journal of Applied Earth Observation and Geoinformation, 73, 752-766, https://doi.org/10.1016/j.jag.2018.08.023, 2018.

Escorihuela, M. J. and Quintana-Seguí, P.: Comparison of remote sensing and simulated soil moisture datasets in Mediterranean landscapes, Remote Sensing of Environment, 180, 99-114, https://doi.org/10.1016/j.rse.2016.02.046, 2016.

Lievens, H., Reichle, R. H., Liu, Q., De Lannoy, G. J. M., Dunbar, R. S., Kim, S. B., Das, N. N., Cosh, M., Walker, J. P. and Wagner, W.: Joint Sentinel-1 and SMAP data assimilation to improve soil moisture estimates, Geophys Res Lett, 44, 6145-6153, https://doi.org/10.1002/2017GL073904, 2017.

Reichle, R. H., De Lannoy, M., G. J. and Liu, Q.: Assessment of the SMAP Level-4 Surface and Root-Zone Soil Moisture Product Using In Situ Measurements, J. Hydrometeorol., 18, 2621-2645, https://doi.org/10.1175/jhm-d-17-0063.1, 2017.

Romaguera, M., Krol, M. S., Salama, M. S., Hoekstra, A. Y. and Su, Z.: Determining irrigated areas and quantifying blue water use in Europe using remote sensing Meteosat Second Generation (MSG) products and Global Land Data Assimilation System (GLDAS) data, Photogrammetric Engineering and Remote Sensing, 78, 861–873, https://doi.org/10.14358/PERS.78.8.861, 2012.

---

## Author Response (AR3)

**Dear Editor and Reviewers,**

We would like to thank the Editor and three Reviewers for their efforts in handling the manuscript and for their valuable comments to improve the manuscript. We have revised our manuscript thoroughly according to the comments and provide a point-by-point response to the comments from three Reviewers below. The original comments from three Reviewers are in black font, and our responses are in blue font.

On behalf of all co-authors,

En Liu

**Response to Reviewer #2's comments on the manuscript egusphere-2023-1597**

**RC2: 'Comment on egusphere-2023-1597', Anonymous Referee #2, 12 Mar 2024**

I believe that the manuscript has been improved based on the reviewer comments. I still have difficulty in grasping the key messages and contributions from the paper. I think this can be overcome by critically removing text that does not directly contribute to the main conclusion. This would make the paper more readable. Besides, I think that strengthening (and shortening) the abstract and conclusion can definitely help to better convey the main message.

Response: The authors thank the Reviewer #2 for her/his constructive and insightful comments that help us improve the quality of the manuscript. The original comments from Reviewer #2 are in black font, and our responses are in blue font.

See below some specific comments:

- The abstract is relatively long and not entirely understandable by itself. For example L28: 'The seven model based products' raises questions by the reader; too many details on SMOS for the abstract; a lot of emphasis on the limitations. I believe that addressing these issues in the abstract will make it more clear and impactful.

Response: We have strengthened and shortened the abstract and conclusion to make it more concise and readable.

The abstract is replaced by

[revised manuscript text omitted]

- L28: we have eight products, but only seven model-based products? I assume SMOS is not considered as model based? Please clarify this throughout the text and in Table 1

Response: Yes, SMOS L4 RZSM product is not considered as model based. SMOS L4 RZSM product is obtained from SMOS L3 surface soil moisture combined with exponential filter method, which have been mentioned in abstract, Table 1 and section 2.4.8.

- L266 doesn't -> does not

Response: Correction done.

- Table 2: which correlation coefficient? Pearson?

Response: Correction done.

- Fig 4: Why are the outliers red, and not the same color as the corresponding box?

Response: The outliers are abnormal values, which is used to distinguish from the normal values.

- 5.3 title: capital W in 'What'

Response: Correction done.

-L740-743: this would fit better at the end of the conclusion

Response: Correction done.

- The authors have put a lot of effort in evaluating potential causes for mismatches between the different datasets, could the authors also put this into perspective in the conclusion? I mean: what is needed to overcome the major issues?

Response: Yes, the following text is added into the conclusion.

"Nevertheless, there are still some shortcomings to be overcome in this study. The land cover type affects the dynamics of soil moisture, future study should focus on the effect of different land cover types on soil moisture simulation."

- Can you put the potential causes for the mismatches into perspective while considering the spatial resolution of the models assessed here? For example, it is not likely that the models will ever cover all the details on the soil textures as presented in Sect. 5.2. The answer to the section title: 'are the soil properties correctly represented' is obviously no. Can you refer to literature that is trying to overcome this issue in the model world? - L104: 2.84m should be 2.89m

Response: Due to the influence of precipitation and underlying surface conditions, soil moisture shows great spatial heterogeneity. Therefore, the validation of point-scale site observations and grid-scale soil moisture simulation values results in spatial scale mismatch. How to address the issue has been illustrated in section 4.3.3 (Line 535-538).

" In addition, upscaling the sparse ground-based observations to the footprint-scale satellite soil moisture retrieval or model grid scale through the temporal stability concept, block kriging, field campaign data, or LSM, reduces the uncertainty of spatial resampling and further improves the reliability of soil moisture validation (Crow et al., 2012)."

2.84m have been replaced by 2.89m.

Reference

Crow, W. T., Berg, A. A., Cosh, M. H., Loew, A., Mohanty, B. P., Panciera, R., de Rosnay, P., Ryu, D. and Walker, J. P.: Upscaling sparse ground-based soil moisture observations for the validation of coarse-resolution satellite soil moisture products, Rev. Geophys., 50, RG2002, https://doi.org/10.1029/2011rg000372, 2012.